# A²SEARCH: AMBIGUITY-AWARE QUESTION ANSWERING WITH REINFORCEMENT LEARNING

**Fengji Zhang[1], Xinyao Niu[2], Chengyang Ying[3], Guancheng Lin[1], Zhongkai Hao[3],**
**Fan Zhou[2], Chengen Huang[2], Jacky Keung[1], Bei Chen[2], Junyang Lin[2]**
[1] City University of Hong Kong, [2] Alibaba Group, [3] Tsinghua University

## ABSTRACT

Recent advances in Large Language Models (LLMs) and Reinforcement Learning (RL) have led to strong performance in open-domain question answering (QA). However, existing models still struggle with questions that admit multiple valid answers. Standard QA benchmarks, which typically assume a single gold answer, overlook this reality and thus produce inappropriate training signals. Existing attempts to handle ambiguity often rely on costly manual annotation, which is difficult to scale to multi-hop datasets such as HotpotQA and MuSiQue. In this paper, we present A²SEARCH, an annotation-free, end-to-end training framework to recognize and handle ambiguity. At its core is an automated pipeline that detects ambiguous questions and gathers alternative answers via trajectory sampling and evidence verification. The model is then optimized with RL using a carefully designed AnsF1 reward, which naturally accommodates multiple answers. Experiments on eight open-domain QA benchmarks demonstrate that A²SEARCH achieves new state-of-the-art performance. With only a single rollout, A²SEARCH-7B yields an average AnsF1@1 score of 48.4% across four multi-hop benchmarks, outperforming all strong baselines, including the substantially larger ReSearch-32B (46.2%). Extensive analyses further show that A²SEARCH resolves ambiguity and generalizes across benchmarks, highlighting that embracing ambiguity is essential for building more reliable QA systems. Our code, data, and model weights can be found at https://github.com/zfj1998/A2Search.

## 1 INTRODUCTION

Open-domain Question Answering (QA) is a fundamental yet challenging task that requires both accurate reasoning and effective search (Rajpurkar et al., 2016; Reddy et al., 2019; Kwiatkowski et al., 2019). Recent advances in the ability of Large Language Models (LLMs) to use external tools (Yao et al., 2024), together with Reinforcement Learning (RL) techniques (Shao et al., 2024; Yu et al., 2025), have driven rapid progress in this area. Models such as Search-R1 (Jin et al., 2025), ReSearch (Chen et al., 2025), and AFM (Li et al., 2025a) achieve strong performance by learning strategies for multi-step reasoning, active tool using, and precise evidence integration.

Yet the field has largely overlooked a pervasive source of difficulty: ambiguity in the questions themselves. Most QA benchmarks assume each question has a single correct answer, but in reality both annotation and real-world questions inevitably leave room for multiple, equally valid responses. This is especially evident in multi-hop questions, where different reasoning chains can legitimately reach different conclusions. Figure 1 shows an example from the MuSiQue benchmark (Trivedi et al., 2022), produced by ReSearch-32B rollouts: distinct answers emerge, each well-supported by evidence and arguably correct. Nevertheless, the benchmark provides only one "gold" reference answer. Such cases are far from rare–our analysis finds that 27.6% of MuSiQue's training examples admit more than one valid answer (see Section 4), and similar patterns occur in other QA datasets. We call the annotated gold the *reference answer* and the others *alternative answers*. Current RL pipelines, which reward only the *reference* and implicitly penalize *alternatives*, deliver misleading reward signals and systematically understate true model capability.

To address this challenge, models must learn not only to recognize when a question is ambiguous but also to present all valid answers rather than commit to a single reasoning path. Evaluation protocols

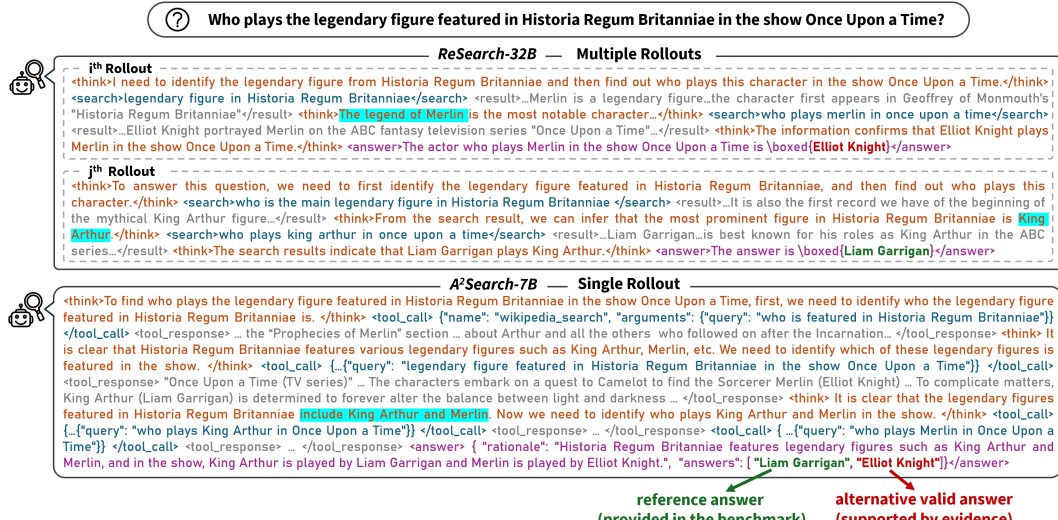

Figure 1: Rollout examples on an ambiguous question from MuSiQue. ReSearch yields different answers across rollouts, some diverging from the reference yet still evidence-supported, whereas A²SEARCH explicitly resolves ambiguity by retrieving multiple answers within a single rollout.

likewise need to evolve to assess performance in genuinely multi-answer settings. In this paper, we propose A²SEARCH, an annotation-free, end-to-end reinforcement learning framework for training ambiguity-aware QA models. At its core is an evidence-verification-based data generation pipeline that automatically detects ambiguous questions and gathers *alternative answers*. The model is then trained with Group Relative Policy Optimization (GRPO), where outcome rewards are based on answer-level F1 (AnsF1), a metric that naturally accommodates multiple answers. By combining multi-step reasoning with tool use, A²SEARCH follows an agentic training paradigm that enables models to sense ambiguity and produce multiple answers whenever the evidence warrants it.

We comprehensively evaluate A²SEARCH on eight open-domain QA benchmarks, achieving comparable or superior performance with only a single greedy decoding rollout, while prior methods typically require three sampled rollouts. On four multi-hop datasets, A²SEARCH-7B yields an average AnsF1@1 of 48.4% under *Exact Match* and 62.7% under *LMJudge* using just one rollout, substantially outperforming ReSearch-32B (46.2% / 60.7%) and far exceeding ReSearch-7B (39.3% / 53.6%). Even the smaller A²SEARCH-3B achieves competitive results (43.1% / 55.3%), demonstrating the efficiency gains of our training paradigm. On AmbigQA (Min et al., 2020), a human-annotated benchmark for ambiguous questions, A²SEARCH surpasses baseline models trained directly on the curated AmbigQA training set, illustrating the robustness and transferability of our ambiguity-aware approach. The main contributions of this work are threefold:

- We introduce a fully automated pipeline that identifies alternative answers for ambiguous questions via trajectory sampling and evidence-based verification.
- We establish a stronger RL baseline for open-domain QA by training A²SEARCH at 3B and 7B scales, achieving state-of-the-art results across eight benchmarks.
- Through comprehensive analyses, we validate both the data pipeline and the RL paradigm, and show that A²SEARCH learns to sense ambiguity and retrieve multiple answers.

## 2 RELATED WORK

Language models equipped with search tools have recently made rapid progress, enabling them to retrieve factual and real-time information through reasoning (Shen et al., 2023; Chang et al., 2024). Existing approaches can be broadly categorized into two main types: prompt-based and training-based methods. Prompt-based methods (Trivedi et al., 2023; Wang et al., 2024a; Yue et al., 2024; Li et al., 2025b; Alzubi et al., 2025) manually design prompts to guide LLMs in invoking search

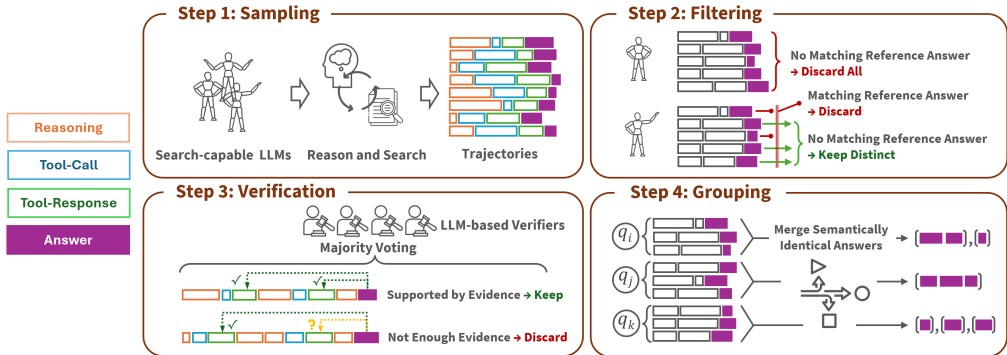

Figure 2: Our pipeline for automatically identifying alternative answers in ambiguous questions.

tools and to construct workflows for multi-turn tool usage. These approaches largely rely on the inherent agentic capabilities of LLMs. In contrast, training-based methods adopt supervised fine-tuning (Wang et al., 2024b) or reinforcement learning (Chen et al., 2025; Jin et al., 2025; Song et al., 2025; Sha et al., 2025; Fan et al., 2025) to improve search skills. In particular, reinforcement learning methods gradually enhance search performance by incorporating interactive feedback from the environment, which lays an important foundation for more complex search tasks in the future.

Multi-hop QA provides a natural environment for training models' search and reasoning capabilities. Representative benchmarks (Joshi et al., 2017; Yang et al., 2018; Kwiatkowski et al., 2019; Ho et al., 2020; Trivedi et al., 2022; Press et al., 2023; Shen et al., 2025) are constructed on Wikipedia, offering questions of sufficient difficulty to require multi-turn search, while the shared corpus ensures reproducibility and stable evaluation. Nevertheless, most of these benchmarks assume a single correct answer per question, thereby overlooking ambiguity and cases with alternative answers. Some efforts, such as AmbigQA (Min et al., 2020) and ASQA (Stelmakh et al., 2022), address ambiguity through manual re-annotation, but they rely heavily on human effort and are mainly limited to single-hop questions, making them difficult to scale or generalize to multi-hop training. Other studies focus on detecting ambiguity without providing alternative answers (Shi et al., 2025; Kim et al., 2024). In contrast, $A^2$SEARCH automatically recognizes ambiguous questions and generates multiple alternative answers, thereby benefiting end-to-end RL training.

## 3 METHODOLOGY

This section provides a formal description of our proposed $A^2$SEARCH. Section 3.1 outlines the automatic pipeline for alternative answer generation, which provides the training data, while Section 3.2 presents the training method based on reinforcement learning.

### 3.1 ALTERNATIVE ANSWER GENERATION

Instead of relying on manual annotation, we build an automatic pipeline that exploits ambiguous questions in existing datasets to produce verifiable and effective training data for reinforcement learning. Formally, given a question $q$ and its reference answer $ans^*$, our objective is to produce a set of alternative answers $\mathcal{A}_{\text{alt}}$ that are semantically distinct from one another, different from the reference answer, and can be independently verified. To achieve this, we carefully design a four-step process that is evidence-based and highly reliable, as illustrated in Figure 2. It first performs **Sampling** to collect multiple automatically generated trajectories, then applies **Filtering** and **Verification**, and finally conducts **Grouping** to obtain alternative answers. Details are provided below.

**Step 1: Sampling.** We employ a collection of $N$ search-capable language models, denoted $\mathcal{S} = \{S_n\}_{n=1}^{N}$, each trained on single-answer QA datasets and equipped with the ability to interact with a search tool. Given a question $q$, every model $S_n$ produces $M$ trajectories $\mathcal{T}_n = \{\tau_{nm}\}_{m=1}^{M}$. A trajectory is represented as $\tau = (a_1, o_1, \ldots, a_T)$, where $a_t$ denotes an action and $o_t$ the content returned by the tool. Actions are of three types: (i) *reasoning*, recording intermediate thinking steps; (ii) *tool-call*, which issues a search query and receives a corresponding *tool-response*; or (iii) *answer*, which outputs a final answer. Only the type *tool-call* yields returned content. At the end of

this step, all trajectories for question $q$ are aggregated into $\mathcal{T}^{(1)} = \bigcup_{n=1}^{N} \mathcal{T}_n$, where each $\tau \in \mathcal{T}^{(1)}$ corresponds to a candidate answer.

**Step 2: Filtering.** Not all the $N \times M$ candidate answers obtained from the trajectories in $\mathcal{T}^{(1)}$ are useful. A *useful* trajectory should provide an *alternative answer* that is valid but different from the *reference answer* $ans^*$. We therefore perform a coarse filtering using three intuitive rules. First, trajectories with answers judged by the LLM to be semantically equivalent to $ans^*$ are removed. Second, suppose all answers in $\mathcal{T}_n$ differ from $ans^*$. In that case, we drop all trajectories from $\mathcal{T}_n$, as this indicates that model $S_n$ is unable to produce the reference answer even with multiple rollouts, suggesting a lack of capability to solve the question. Third, for trajectories that produce exactly identical answers, we keep only one representative trajectory. The resulting filtered set is denoted by $\mathcal{T}^{(2)}$, where each $\tau \in \mathcal{T}^{(2)}$ provides a distinct candidate answer.

**Step 3: Verification.** For the filtered set $\mathcal{T}^{(2)}$, we perform a fine-grained verification to determine whether each trajectory $\tau \in \mathcal{T}^{(2)}$ provides sufficient evidence to support its candidate answer $a\hat{n}s$. We employ $K$ number of LLM-based verifiers, denoted as $\mathcal{V} = \{V_k\}_{k=1}^{K}$, where each verifier $V_k$ takes $(q, \tau, a\hat{n}s)$ as input and outputs a binary judgment $z_k \in \{0, 1\}$. A value of $z_k = 1$ indicates that the trajectory contains sufficient evidence to support the candidate answer as a valid alternative answer; otherwise, $z_k = 0$. We aggregate the results using majority voting:

$$\text{Verify}(q, \tau, a\hat{n}s) = \begin{cases} 1, & \text{if } \frac{1}{K} \sum_{k=1}^{K} z_k \geq \eta, \\ 0, & \text{otherwise,} \end{cases}$$

where $\eta$ is the voting threshold. For each $\tau \in \mathcal{T}^{(2)}$, we perform verification and retain those with $\text{Verify}(q, \tau, a\hat{n}s) = 1$, resulting in the verified trajectory set $\mathcal{T}^{(3)}$.

**Step 4: Grouping.** Finally, we apply a clustering procedure to the verified trajectory set $\mathcal{T}^{(3)}$, using an LLM to merge semantically equivalent answers into groups. The final alternative answer set is then given by $\mathcal{A}_{\text{alt}} = \text{Group}(\mathcal{T}^{(3)})$, where $\text{Group}(\cdot)$ denotes a semantic clustering operator that groups semantically equivalent candidates and selects one representative alternative answer per cluster, retaining the others as aliases. Typical cases of semantic equivalence include abbreviation versus full name (e.g., ["NDZ", "Nkosazana Dlamini-Zuma"]), different numeric representations (e.g., ["five", "5"]), and variations in word order (e.g., ["2001 fiscal year", "fiscal year 2001"]).

## 3.2 REINFORCEMENT LEARNING FRAMEWORK.

Through the above pipeline, we obtain the training data by extending the reference answer set with mined alternative answers, denoted as $\mathcal{A} = \{ans^*, \mathcal{A}_{\text{alt}}\}$. This extension allows some questions to have multiple reference answers. We then design a reinforcement learning algorithm that uses an answer-level F1 reward, which is suitable for scenarios involving multiple reference answers.

**Training Objective.** We adopt Group Relative Policy Optimization (GRPO) (Shao et al., 2024) as the reinforcement learning algorithm. Unlike Proximal Policy Optimization (Schulman et al., 2017), which relies on a separately trained critic network to provide a baseline, GRPO estimates the baseline directly from a group of sampled rollouts. Concretely, given an existing policy $\pi_{\text{old}}$, we generate $G$ rollouts $\{y_i\}_{i=1}^{G} \sim \pi_{\text{old}}(\cdot|x)$ for each input $x \sim \mathcal{D}$. Following He et al. (2025) and Yu et al. (2025), we discard the KL penalty term. The optimization objective is then to update the policy $\pi_\theta$ by maximizing

$$\mathcal{J}(\theta) = \mathbb{E}_{x \sim \mathcal{D}, \{y_i\}_{i=1}^{G} \sim \pi_{\theta_{\text{old}}}(\cdot|x)} \frac{1}{G} \sum_{i=1}^{G} \left[ \min \left( \frac{\pi_\theta(y_i|x)}{\pi_{\theta_{\text{old}}}(y_i|x)} A_i, \text{clip} \left( \frac{\pi_\theta(y_i|x)}{\pi_{\theta_{\text{old}}}(y_i|x)}, 1 - \epsilon, 1 + \epsilon \right) A_i \right) \right],$$

where $A_i = \left( r_i - \text{mean}(\{r_j\}_{j=1}^{G}) \right) / \text{std}(\{r_j\}_{j=1}^{G})$ is the normalized advantage of the $i$-th rollout in the group, $r_i$ is the reward, and $\epsilon$ is the clipping ratio.

**Rollout with Search Tool.** We formulate rollout generation as an iterative interaction between the policy and a search tool. Given a question $q$, the model constructs a trajectory $\tau = (a_1, o_1, \ldots, a_T)$ consisting of alternating actions $a_t$ and tool responses $o_t$ (with $o_t$ empty unless $a_t$ is a tool call). At each step $t$, the state $s_t$ is the accumulated prompt containing the question, all past actions, and any returned responses. The policy $\pi_\theta$ samples an action $a_t \sim \pi_\theta(\cdot|s_t)$ from three types: *reasoning,*

*tool-call* (which returns a *tool-response*); and *answer*, as described in Section 3.1. The rollout terminates once an end-of-sequence token is generated or a maximum length is reached. Each trajectory, therefore, encapsulates the model's reasoning, search interactions, and final prediction, and serves as the basic unit for estimating the rewards used in reinforcement learning. During training, tokens in *tool-response blocks* are excluded from the policy loss via masking, since they are generated by the external search tool rather than by the policy model itself.

**Reward Design.** We employ an outcome-only reward design, which has been proven successful in recent studies (Guo et al., 2025; Yu et al., 2025). The reward combines a format check with an answer-matching score. An output is considered *format-valid* if it satisfies all of the following: (i) containing at least one successful tool call, (ii) including intermediate reasoning blocks, and (iii) terminating with an end-of-sequence token after exactly one answer block whose content can be correctly parsed. Outputs failing any of these criteria receive a reward 0. If the format is valid but none of the reference answers are matched, we assign a small constant reward of 0.1. Otherwise, we compute an $\mathrm{AnsF1}$ (Answer-level F1) score based on exact match. $\mathrm{AnsF1}$ rewards coverage of valid answers while penalizing over-generation, balancing precision and recall, and remaining comparable across questions with varying levels of ambiguity.

For answer matching, we define three quantities: $\mathrm{preds}$ denotes the total number of answers produced by the model rollout; $\mathrm{hits}$ denotes the number of *reference answers* exactly matched by the predictions; and $\mathrm{refs}$ denotes the total number of *reference answers*. We define precision as $\mathrm{Precision} = \mathrm{hits}/\mathrm{preds}$ and recall as $\mathrm{Recall} = \mathrm{hits}/\mathrm{refs}$, with $\mathrm{AnsF1} = 2 \cdot \mathrm{Precision} \cdot \mathrm{Recall}/\left(\mathrm{Precision} + \mathrm{Recall}\right)$. The final reward is then defined as

$$R(q, a\hat{n}s) = \begin{cases} 0, & \text{if format invalid,} \\ 0.1, & \text{if format valid and } \mathrm{hits} = 0, \\ 1 - \alpha\left(1 - \mathrm{AnsF1}\right), & \text{if format valid and } \mathrm{hits} > 0, \end{cases}$$

where $\alpha \in [0, 1]$ controls the relative margin between format-valid but fully incorrect predictions and partially correct ones, and $a\hat{n}s$ denotes the predicted answer set of a trajectory, which may contain multiple predicted answers. This design ensures that the model is encouraged to follow the required format, produce valid alternative answers, and cover as many reference answers as possible.

## 4 TRAINING DATA CONSTRUCTION

In this section, we present the implementation details of training data construction, based on the automatic pipeline described in Section 3.1. As a result, we identified alternative answers for 19.0% of the 49,938 questions through 19,529 trajectories.

**Step 1: Sampling.** To encourage diversity in the sampled trajectories, we employ five distinct search models: ReSearch-7B/32B and Search-R1-7B/14B/32B. These models achieve state-of-the-art performance on open-domain QA benchmarks and, importantly, are able to produce search trajectories, which we later leverage for evidence-based verification. For each question, we generate 16 trajectories from each model using a sampling temperature of 0.8. We utilize the same search tool as introduced in Search-R1, where the 2018 Wikipedia dump is partitioned into 100-word chunks, embedded with E5 (Wang et al., 2022), and indexed using FAISS (Douze et al., 2024). At query time, the retriever returns the top-5 passages ranked by embedding similarity. The source questions are drawn from the full training splits of two multi-hop QA datasets, MuSiQue with 19,938 questions and 2Wiki (Ho et al., 2020) with 15,000 questions, together with 15,000 randomly sampled questions from the single-hop dataset NQ (Kwiatkowski et al., 2019). This setup yields around 3.99 million trajectories across 49,938 questions. Despite the scale, the sampling process remains efficient and computationally practical. Additional details on runtime and computational cost are provided in Appendix B.1.

**Step 2: Filtering.** We employ Qwen2.5-32B-Instruct (Yang et al., 2024) as an automatic judge, using the prompt provided in Appendix G.1 to determine whether a trajectory's predicted answer is semantically equivalent to the reference answer. The evaluation shows that 86.8% of the questions contain at least one trajectory matching the reference answer, demonstrating the high quality of the sampled trajectories. After applying the filtering rules defined in Section 3.1 and performing answer-level deduplication, 208,829 trajectories are retained, accounting for 5.2% of the original

3.99 million. These trajectories span $33,997$ questions, covering $68.1\%$ of the $49,938$ source questions. On average, $6.1$ distinct trajectories remain per question, which constitute the input to the verification stage. Further details and statistics of the filtering step are provided in Appendix B.2.

**Step 3: Verification.** To determine whether candidate answers are supported by evidence, we use four proprietary LLMs as verifiers (Claude 3.5 Sonnet, Claude 3.7 Sonnet, OpenAI o3, and OpenAI o4-mini) with the prompt provided in Appendix G.2. Each verifier assigns one of three labels to a trajectory: *supported*, *partially supported*, or *not supported*. The intermediate category prevents borderline cases from being overly judged as *supported*, thereby improving robustness. For aggregation, we retain only the *supported* trajectories and apply majority voting with threshold $\eta$. We further study how the choice of $\eta$ affects reliability by conducting an ablation with human evaluation: for each threshold, we randomly sample 100 positively voted answers and measure the human agreement rate. As expected, stricter thresholds improve agreement but substantially reduce the number of remaining trajectories. We therefore adopt $\eta = 3$, which achieves $96\%$ human agreement while maintaining adequate coverage, leaving $19,529$ trajectories ($9.4\%$ of those from the previous step). Further details and statistics are provided in Appendix B.3, and Appendix B.4 presents an additional study showing that a fully open-weight verifier ensemble can closely approximate the judgments of the proprietary verifiers.

**Step 4: Grouping.** Model-generated answers often differ lexically while being semantically identical. To consolidate such variants, we apply Claude 3.7 Sonnet with a prompt in Appendix G.3, which groups semantically equivalent answers into clusters and assigns each cluster a canonical form with aliases. Overall, $28.6\%$ of candidate answers are grouped in this way.

We then construct the final training dataset. Figure 3 reports the distribution of answer multiplicity across datasets. While most questions remain associated with a single reference answer, a substantial portion contains multiple alternative answers. MuSiQue shows the highest ambiguity ratio, with $5,498$ questions ($27.6\%$) containing alternative answers, compared to $1,076$ questions ($7.2\%$) in 2Wiki and $2,899$ questions ($19.3\%$) in NQ.

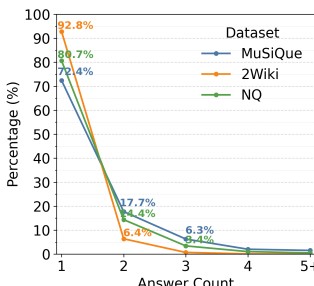

Figure 3: Answer count distribution in the final dataset.

## 5 EXPERIMENTS

Following the procedure in Section 4, we construct a training dataset containing $49,938$ questions, among which approximately $19\%$ have alternative answers. On top of this dataset, we employ the RL framework described in Section 3.2 to perform end-to-end training and obtain our ambiguity-aware model, $A^2$SEARCH. We then validate its effectiveness through systematic comparisons with a diverse set of baselines. The experimental setup is presented in Section 5.1, while results and analyses are provided in Section 5.2. Overall, the experiments demonstrate that our data construction pipeline combined with RL training is highly effective: $A^2$SEARCH can resolve ambiguous questions by identifying multiple answers and outperforms baseline methods.

### 5.1 EXPERIMENTAL SETUP

**Benchmarks.** We evaluate our ambiguity-aware training on eight open-domain QA benchmarks. The multi-hop setting is represented by MuSiQue ($2,417$ questions), HotpotQA ($7,405$) (Yang et al., 2018), 2Wiki ($12,576$), and Bamboogle ($125$) (Press et al., 2023). For general open-domain QA, we use NQ ($8,757$), TriviaQA ($8,837$) (Joshi et al., 2017), and PopQA ($14,267$) (Mallen et al., 2023). Finally, we include AmbigQA ($2,002$) (Min et al., 2020), a variant of NQ augmented with human-annotated alternative answers for ambiguous questions, averaging $2.1$ answers per question.

**Baselines.** We compare our method against four categories of baselines. *(1) Prompt-based methods.* These methods involve no model training: *DirectGen* simply prompts the model to answer the question. *Naive-RAG* retrieves passages in a single round, concatenates them with the question, and asks the model to generate an answer. *Iter-RetGen* (Shao et al., 2023) extends this by performing three fixed rounds of retrieval and generation, where each retrieval step can use previously generated content. *IRCoT* (Trivedi et al., 2023) further integrates retrieval into chain-of-thought reasoning, allowing arbitrary iterations. *(2) RL-trained search models.* We include Search-R1 (Jin et al., 2025),

ReSearch (Chen et al., 2025), and AFM-MHQ (Li et al., 2025a), all trained with reinforcement learning in an agentic fashion. *(3) $SinSearch$.* This baseline is trained on the same set of questions as A$^2$SEARCH, but it relies solely on the original *single* reference answer provided for each question in the datasets. The training prompt for this model is provided in Appendix G.4. *(4) AbgSearch.* This baseline is trained on the AmbigQA dataset (10,036 questions) using the same training setup as A$^2$SEARCH. Because AmbigQA primarily contains simpler single-hop questions, this comparison underscores the importance of constructing multi-answer datasets for the more challenging multi-hop setting. The prompt for this model is in Appendix G.5.

**Evaluation Metrics.** Two primary metrics are used: AnsF1(@1) and Recall(@1). As described in Section 3.2, they are computed from hits, preds, and refs. To obtain hits, we adopt two complementary schemes: *Exact Match*, which checks whether a prediction exactly matches a reference answer or one of its aliases, and *LMJudge*, implemented with Qwen2.5-32B-Instruct using the prompt in Appendix G.1. We additionally report AnsF1@$k$ and Recall@$k$, where $k$ denotes the number of sampled trajectories. To reduce the effect of sampling randomness, we approximate the expected value of @$k$ ($k > 1$) by repeatedly subsampling $k$ items from $k'$ ($k' > k$) sampled trajectories, with the detailed algorithm provided in Appendix D.1.

**Hyperparameters.** For a fair comparison with prior work, we use the Qwen2.5 model family (Yang et al., 2024), experimenting with the 3B, 7B, *Base*, and *Instruct* variants to examine the generalization capability of our training framework. Training prompts are provided in Appendix G.5 and G.6. We train with a batch size of 256, learning rate $1e-6$, maximum context length 8,192, rollout size 16, and 4 epochs. The parameter $\alpha$ in reward design is set to 0.4. We hold out 512 randomly sampled questions from the MuSiQue development set for hyperparameter tuning and checkpoint selection. Ablation studies on $\alpha$ and rollout size are reported in Appendix E.1 and E.2. For evaluation, AnsF1@1 is computed with greedy decoding, and AnsF1@3 is estimated by generating 6 rollouts for single-answer search models using a sampling temperature of 0.6. Further analysis of sampling temperature effects is provided in Appendix E.3.

## 5.2 EXPERIMENTAL RESULTS

**Main Evaluation Results.** Table 1 and 2 report the overall performance on eight QA benchmarks under the *Exact Match* metric. Additional results evaluated with the *LMJudge* are provided in Appendix E.4. Both evaluation metrics yield consistent findings, which we summarize as follows:

(1) Agentic search models consistently outperform prompt-based and RAG-based baselines, with especially clear gains on multi-hop benchmarks. In addition, their Recall@3 is substantially higher than Recall@1 across all model sizes, indicating notable headroom unlocked by modest sampling. (2) Even with a single greedy decoding rollout, A$^2$SEARCH reaches recall levels comparable to or surpassing the @3 performance of baselines, and even outperforms larger 32B models on several multi-hop benchmarks. While $Sin$Search, trained under the same setup but without alternative answers, performs competitively against other baselines, it still falls short of A$^2$SEARCH. Moreover, A$^2$SEARCH achieves the best AnsF1 on most benchmarks, striking a strong precision–recall balance. On average, A$^2$SEARCH-7B generates 1.51 answers per question and A$^2$SEARCH-3B generates 1.23, with detailed per-benchmark statistics provided in Appendix C.1. (3) $Abg$Search performs well on AmbigQA but fails to generalize to other datasets. In contrast, A$^2$SEARCH is trained without AmbigQA data and still surpasses it on the same benchmark. This result highlights the effectiveness of our evidence-based data generation pipeline.

**Training Dynamics and Stability.** Throughout the four training epochs, both the 3B and 7B models exhibit stable and consistent improvements. As shown in Figure 4, AnsF1 and Recall steadily increase without signs of collapse or instability, indicating reliable training dynamics. Moreover, the entropy curves indicate that the models do not exhibit premature entropy collapse. The 3B model maintains a lower entropy than the 7B model, fluctuating around 0.2, while the 7B model remains within a healthy range of variation, suggesting that it preserves sufficient diversity in training.

**Additional Analyses.** We conduct several complementary analyses, with full details provided in the appendix. (1) Our framework generalizes successfully to Qwen2.5-Base models, where it achieves strong recall at the @3 level with a single rollout (Appendix A). (2) Our framework also generalizes to Llama-based models, where it exhibits similar performance trends and demonstrates

| Model | HotpotQA | | 2Wiki | | MuSiQue | | Bamboogle | | Macro-Avg | |
|---|---|---|---|---|---|---|---|---|---|---|
| AnsF1/Recall@k | @1 | @3 | @1 | @3 | @1 | @3 | @1 | @3 | @1 | @3 |
| **Models with 3B Parameters** | | | | | | | | | | |
| DirectGen-3B | 15.9 | 16.2 /17.8 | 24.8 | 25.5 /27.8 | 2.3 | 2.4 /3.1 | 2.4 | 3.1 /3.8 | 11.3 | 11.8 /13.1 |
| Naive-RAG-3B | 28.2 | 28.3 /29.3 | 24.7 | 25.2 /26.7 | 5.7 | 5.7 /5.1 | 9.6 | 9.7 /10.4 | 17.1 | 17.2 /17.9 |
| Iter-RetGen-3B | 30.1 | 30.8 /32.7 | 26.0 | 26.9 /29.3 | 7.0 | 7.3 /8.1 | 11.2 | 12.4 /13.8 | 18.6 | 19.4 /21.0 |
| IRCoT-3B | 27.6 | 29.2 /36.7 | 21.4 | 24.8 /34.8 | 6.9 | 7.7 /10.6 | 20.8 | 22.6 /31.4 | 19.2 | 21.1 /28.4 |
| Search-R1-3B | 37.0 | 38.2 /41.4 | 39.7 | 42.3 /47.7 | 15.1 | 16.3 /18.9 | 36.8 | 35.7 /38.0 | 32.2 | 33.1 /36.5 |
| AFM-MHQ-3B | 41.5 | **43.0** /51.6 | 43.6 | 46.9 /**59.0** | 17.5 | 18.9 /25.4 | 39.2 | 40.6 /50.3 | 35.5 | 37.4 /**46.6** |
| $Sin$Search-3B | 37.9 | 41.1 /47.1 | 47.3 | 50.8 /58.2 | 19.5 | 20.5 /25.6 | 38.4 | 38.2 /41.8 | 35.8 | 37.7 /43.2 |
| $Abg$Search-3B | | 28.3 /31.4 | | 28.7 /34.8 | | 8.94 /9.85 | | 20.9 /21.6 | | 21.7 /24.4 |
| A²SEARCH-3B | | 42.8 /44.4 | | **56.2** /58.9 | | **24.2** /25.9 | | **49.3** /50.4 | | **43.1** /44.9 |
| **Models with 7 ∼ 32B Parameters** | | | | | | | | | | |
| DirectGen-7B | 19.3 | 19.5 /21.1 | 25.5 | 26.8 /30.2 | 3.8 | 4.1 /4.9 | 10.4 | 10.9 /12.0 | 14.8 | 15.3 /17.1 |
| Naive-RAG-7B | 31.9 | 32.1 /33.2 | 25.9 | 26.0 /27.1 | 6.3 | 6.4 /6.7 | 20.8 | 20.7 /22.9 | 21.2 | 21.3 /22.5 |
| Iter-RetGen-7B | 34.3 | 35.0 /36.8 | 28.0 | 28.9 /31.2 | 8.8 | 9.2 /10.4 | 21.6 | 21.6 /23.1 | 23.2 | 23.7 /25.4 |
| IRCoT-7B | 30.3 | 32.9 /39.8 | 21.7 | 24.3 /33.1 | 7.4 | 7.9 /10.8 | 24.0 | 24.3 /30.2 | 20.9 | 22.4 /28.5 |
| ReSearch-7B | 43.2 | 45.6 /51.9 | 47.2 | 51.4 /61.4 | 22.6 | 24.9 /31.1 | 44.0 | 46.5 /53.5 | 39.3 | 42.1 /49.5 |
| Search-R1-7B | 43.2 | 44.5 /47.5 | 39.8 | 42.3 /47.9 | 20.0 | 20.7 /23.6 | 42.4 | 41.9 /45.8 | 36.4 | 37.4 /41.2 |
| AFM-MHQ-7B | 46.1 | 47.6 /53.9 | 46.2 | 48.9 /58.0 | 20.5 | 21.5 /27.4 | 43.2 | 46.3 /53.5 | 39.0 | 41.1 /48.2 |
| Search-R1-14B | 47.5 | 47.9 /51.6 | 48.1 | 50.0 /55.5 | 25.4 | 26.7 /31.0 | 51.2 | 51.2 /53.4 | 43.1 | 44.0 /47.9 |
| Search-R1-32B | 46.2 | 47.8 /51.9 | 51.0 | 53.5 /61.1 | 25.1 | 26.3 /31.4 | 53.6 | 55.8 /61.0 | 44.0 | 45.9 /51.4 |
| ReSearch-32B | 46.6 | 49.4 /**54.1** | 53.0 | 57.9 /**66.7** | 26.0 | 28.6 /34.3 | **59.2** | 59.1 /**64.4** | 46.2 | **48.8** /**54.9** |
| $Sin$Search-7B | 45.6 | 46.9 /50.3 | 57.6 | 59.5 /64.1 | 25.4 | 27.0 /30.9 | 48.8 | 50.6 /53.8 | 44.4 | 46.0 /49.8 |
| $Abg$Search-7B | | 39.2 /43.7 | | 35.5 /41.8 | | 15.9 /19.0 | | 33.7 /35.2 | | 31.1 /34.9 |
| A²SEARCH-7B | | **49.5** /52.1 | | **62.3** /64.4 | | **30.1** /34.8 | | 51.7 /53.6 | | 48.4 /51.2 |

Table 1: Main results on four multi-hop QA benchmarks under the *Exact Match* metric. We report AnsF1/Recall@$k$ with $k$ rollouts. For $Abg$Search and A²SEARCH, only @1 is reported, reflecting their ability to produce multiple answers within a single rollout. For the remaining baselines, where each rollout generates only one answer and thus AnsF1@1 = Recall@1, we additionally include AnsF1/Recall@3 to evaluate their performance when more rollouts are available. The best result in each comparison group is shown in **bold**, and the second best is underlined.

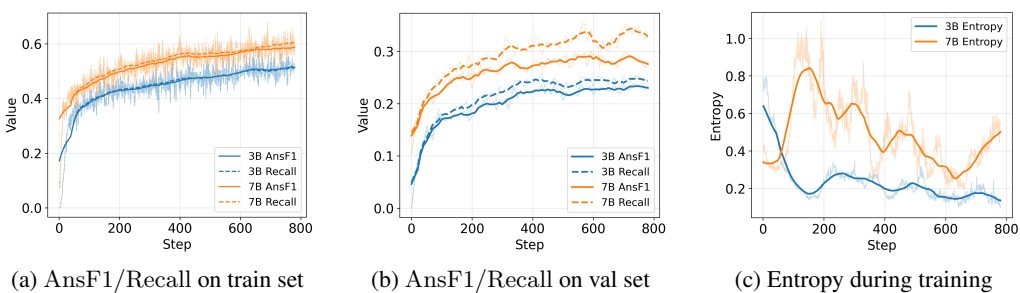

(a) AnsF1/Recall on train set     (b) AnsF1/Recall on val set     (c) Entropy during training

Figure 4: Training dynamics of A²SEARCH. Curves are smoothed for readability.

that the ambiguity-aware RL training is backbone-agnostic and broadly applicable across model families (Appendix E.5). (3) We analyze rollout efficiency by measuring both the number of tool calls and the corresponding recall. We observe that A²SEARCH reaches the @3 recall level with fewer tool calls on average (2.16 for 3B and 4.14 for 7B), demonstrating more effective utilization of reasoning steps (Appendix C.2). (4) To illustrate the prevalence of ambiguity in evaluation benchmarks, we apply the verification steps from Section 4 to trajectories sampled from baseline models, and find that a non-trivial portion of questions admit alternative answers (e.g., 19.7% in MuSiQue, 11.1% in HotpotQA, 8.6% in 2Wiki, and 8% in Bamboogle; More details in Appendix C.3). (5) We further train three additional ablation models to disentangle the contributions of ambiguity-aware data construction and the AnsF1 reward. The results show that neither component alone is sufficient. Full details are provided in Appendix E.6.

| Model
AnsF1/Recall@$k$ | NQ
@1 | @3 | TriviaQA
@1 | @3 | PopQA
@1 | @3 | AmbigQA
@1 | @3 | Macro-Avg
@1 | @3 |
|---|---|---|---|---|---|---|---|---|---|---|
| **Models with 3B Parameters** | | | | | | | | | | |
| DirectGen-3B | 11.4 | 11.7/$_{13.6}$ | 32.9 | 33.3/$_{37.0}$ | 13.0 | 13.1/$_{14.7}$ | 10.8/$_{9.3}$ | 11.6/$_{11.2}$ | 17.0/$_{16.7}$ | 17.4/$_{19.1}$ |
| Naive-RAG-3B | 38.2 | 38.8/$_{40.3}$ | 57.0 | 57.3/$_{58.7}$ | 41.4 | 41.7/$_{42.8}$ | 36.6/$_{31.6}$ | 37.3/$_{32.8}$ | 43.3/$_{42.1}$ | 43.8/$_{43.7}$ |
| Iter-RetGen-3B | 39.2 | 39.9/$_{42.0}$ | 58.8 | 59.3/$_{61.1}$ | 43.9 | 44.3/$_{45.9}$ | 38.4/$_{33.1}$ | 39.3/$_{35.1}$ | 45.1/$_{33.8}$ | 45.7/$_{46.0}$ |
| IRCoT-3B | 25.3 | 27.6/$_{34.9}$ | 45.2 | 48.0/$_{56.3}$ | 33.8 | 36.6/$_{42.5}$ | 27.6/$_{24.3}$ | 31.3/$_{32.1}$ | 33.0/$_{32.1}$ | 35.9/$_{41.5}$ |
| Search-R1-3B | 43.9 | 44.6/$_{46.8}$ | 60.1 | 60.8/$_{63.0}$ | 46.5 | 47.0/$_{48.4}$ | 38.7/$_{33.4}$ | 39.8/$_{35.5}$ | 47.3/$_{46.0}$ | 48.0/$_{48.4}$ |
| AFM-MHQ-3B | 38.3 | 38.9/$_{48.2}$ | 58.1 | 59.0/**$_{67.6}$** | 37.8 | 39.2/$_{47.2}$ | 36.4/$_{31.6}$ | 39.0/**$_{39.4}$** | 42.6/$_{41.5}$ | 44.0/**$_{50.6}$** |
| *Sin*Search-3B | 40.9 | 43.3/$_{48.2}$ | 58.0 | 59.9/$_{64.9}$ | 43.7 | 45.0/$_{49.3}$ | 38.2/$_{32.8}$ | 41.2/$_{38.6}$ | 45.2/$_{43.9}$ | 47.4/$_{50.2}$ |
| *Abg*Search-3B | 41.3/$_{45.3}$ | | 54.9/$_{57.0}$ | | 39.3/$_{42.6}$ | | 40.4/$_{36.6}$ | | 44.0/$_{45.4}$ | |
| A²SEARCH-3B | **47.3**/$_{49.7}$ | | **60.9**/$_{62.5}$ | | **48.2**/$_{50.5}$ | | **43.1**/$_{38.2}$ | | **49.9**/$_{50.2}$ | |
| **Models with 7 ∼ 32B Parameters** | | | | | | | | | | |
| DirectGen-7B | 14.3 | 15.0/$_{16.7}$ | 44.3 | 44.7/$_{47.5}$ | 15.2 | 15.5/$_{17.1}$ | 14.1/$_{12.2}$ | 14.9/$_{14.0}$ | 22.0/$_{21.5}$ | 22.5/$_{23.8}$ |
| Naive-RAG-7B | 38.7 | 38.7/$_{39.9}$ | 61.0 | 61.4/$_{62.5}$ | 40.1 | 40.2/$_{41.1}$ | 37.8/$_{33.1}$ | 37.8/$_{33.9}$ | 44.4/$_{43.2}$ | 44.5/$_{44.3}$ |
| Iter-RetGen-7B | 40.4 | 40.6/$_{42.5}$ | 62.6 | 63.3/$_{64.7}$ | 42.8 | 43.4/$_{45.2}$ | 39.5/$_{34.5}$ | 40.3/$_{36.2}$ | 46.3/$_{45.1}$ | 46.3/$_{47.1}$ |
| IRCoT-7B | 25.9 | 27.5/$_{34.1}$ | 53.7 | 55.0/$_{60.6}$ | 34.3 | 36.0/$_{41.2}$ | 27.8/$_{24.4}$ | 30.0/$_{30.9}$ | 35.4/$_{34.6}$ | 37.1/$_{41.7}$ |
| ReSearch-7B | 42.4 | 44.5/$_{50.9}$ | 63.1 | 65.3/$_{70.4}$ | 44.7 | 46.7/$_{52.2}$ | 40.8/$_{35.4}$ | 45.3/$_{42.8}$ | 47.8/$_{46.4}$ | 50.5/$_{54.1}$ |
| Search-R1-7B | 47.7 | 48.4/$_{50.2}$ | 64.0 | 64.6/$_{66.5}$ | 46.1 | 46.9/$_{48.4}$ | 41.8/$_{36.0}$ | 43.1/$_{38.2}$ | 49.9/$_{48.5}$ | 50.8/$_{50.8}$ |
| AFM-MHQ-7B | 44.4 | 46.5/$_{53.7}$ | 64.4 | 66.2/$_{71.3}$ | 43.3 | 45.8/$_{52.0}$ | 41.1/$_{35.5}$ | 44.5/$_{42.1}$ | 48.3/$_{46.9}$ | 50.8/$_{54.8}$ |
| Search-R1-14B | 50.1 | 50.4/$_{52.9}$ | 67.0 | 67.7/$_{71.4}$ | 49.6 | 50.3/$_{53.6}$ | 43.8/$_{37.8}$ | 45.2/$_{40.5}$ | 52.6/$_{51.1}$ | 53.4/$_{54.6}$ |
| Search-R1-32B | 49.1 | 50.3/$_{53.7}$ | 70.0 | **71.1**/$_{74.0}$ | 49.6 | 51.0/$_{54.8}$ | 44.3/$_{38.3}$ | 46.4/$_{42.0}$ | 53.2/$_{51.8}$ | 54.7/$_{56.1}$ |
| ReSearch-32B | 43.0 | 46.8/$_{51.7}$ | 67.7 | 70.4/**$_{74.3}$** | 48.2 | 51.3/**$_{56.0}$** | 44.1/$_{38.2}$ | 47.8/**$_{44.1}$** | 50.8/$_{49.3}$ | 54.1/**$_{56.5}$** |
| *Sin*Search-7B | 49.3 | 49.8/$_{51.3}$ | 66.2 | 67.0/$_{69.2}$ | 50.5 | 51.4/$_{53.5}$ | 44.6/$_{38.4}$ | 45.1/$_{39.8}$ | 52.6/$_{51.1}$ | 53.3/$_{53.5}$ |
| *Abg*Search-7B | 47.6/$_{53.7}$ | | 64.8/$_{68.2}$ | | 48.0/$_{53.2}$ | | 47.5/$_{43.9}$ | | 52.0/$_{54.8}$ | |
| A²SEARCH-7B | **51.4**/**$_{54.7}$** | | 67.8/$_{69.6}$ | | **52.5**/$_{55.6}$ | | **48.1**/$_{43.2}$ | | **55.0**/$_{55.8}$ | |

Table 2: Main results with the *Exact Match* metric on four general QA benchmarks, using the same notations as Table 1. For AmbigQA, where questions may have multiple reference answers, AnsF1@1 and Recall@1 are not equivalent in this setting, and both are therefore reported.

| Ambiguity Type | Definition |
|---|---|
| Under-Constrained | Question lacks critical contextual constraints, allowing multiple distinct entities to satisfy the query. |
| Granularity Ambiguity | Specificity level (e.g., spatial, temporal) is not stated, yielding answers valid at different granularities. |
| Time Sensitivity | Correct answer depends on an implicit temporal reference point. |
| Evidence Conflict | Retrieved passages provide contradictory factual claims about the same entity. |
| Multi-Item Response | Question expects a single answer but the true answer is a non-singleton set without a selection criterion. |
| Open-Ended | Question is subjective or interpretive, supporting multiple qualitatively valid responses. |
| Alias Variance | Different surface forms refer to the same underlying real-world entity. |

Table 3: Ambiguity taxonomy and definitions. Examples of these types are listed in Table 12.

## 5.3 Understanding the Sources of Ambiguity

To analyze the underlying causes of ambiguity in QA datasets and understand why multiple valid answers emerge, we conduct a systematic analysis of the underlying ambiguity types present in our training data. Through manual inspection of the multi-answer portion of our constructed dataset, we first identified seven recurring categories of ambiguity that account for the vast majority of cases. The resulting taxonomy and its definitions are shown in Table 3. We then use a strong LLM judge (OpenAI o3, full prompt in Appendix G.8) to automatically assign an ambiguity type to each ambiguous question in the dataset. For every question that contains multiple valid answers, the judge is provided with the question as well as multiple evidence-supported execution trajectories that lead to different valid answers. The judge is asked to compare these trajectories, infer the source of ambiguity, and select the primary ambiguity category.

In terms of prevalence, *Under-Constrained* questions dominate the ambiguous portion of the dataset (52%), followed by *Granularity Ambiguity* (34%). *Time Sensitivity* (5%) and *Evidence Conflict* (4%) occur far less frequently, while *Multi-Item Response* accounts for 3%. Truly *Open-Ended* questions and *Alias Variance* each constitute only about 1% of cases, indicating that most ambiguity arises from meaningful semantic underspecification rather than superficial variation. To more intuitively illustrate how A²SEARCH identifies and resolves genuine ambiguity, we provide addi-

tional representative ambiguous questions together with the full reasoning-and-search trajectories in Appendix H.

## 6  CONCLUSION

In this work, we revisited the challenge of ambiguity in open-domain QA, a pervasive yet underexplored issue. We proposed $A^2$SEARCH, an annotation-free RL framework that automatically identifies ambiguous questions, discovers alternative answers through trajectory sampling and evidence verification, and optimizes models with an answer-level F1 reward that naturally accommodates multiple references. Our experiments confirm that $A^2$SEARCH attains state-of-the-art results with a single rollout and generalizes effectively to AmbigQA, demonstrating both scalability and robustness. Analyses further reveal that the model acquires the ability to detect ambiguity and generate multiple valid answers. These findings suggest that real progress in QA requires explicitly embracing ambiguity. By treating alternative answers as first-class signals, models become more reliable and better aligned with human expectations. We expect this perspective to inform future evaluation protocols and extend naturally to broader domains.

## ETHICS STATEMENT

This work adheres to the ICLR Code of Ethics. Our research aims to advance open-domain QA while contributing positively to scientific excellence and societal well-being. We emphasize transparency by addressing the overlooked issue of ambiguity in QA benchmarks, which, if ignored, can lead to misleading evaluations and the underestimation of model capabilities. Our methodology requires no additional human annotation, thereby avoiding ethical risks associated with large-scale data collection. All experiments were conducted responsibly, with careful consideration of reproducibility, fairness, and inclusiveness. By explicitly modeling ambiguity, our approach seeks to reduce harmful biases introduced by incomplete ground-truth annotations and to promote more trustworthy, reliable QA systems that respect diverse perspectives.

## REPRODUCIBILITY STATEMENT

To facilitate reproducibility, we have open-sourced the source code, constructed dataset, and model weights at `https://github.com/zfj1998/A2Search`. Detailed experimental setups are described in Section 5.1, while additional implementation and hyperparameter settings are given in Appendix D–E. The exact language model prompts used in our experiments are included in Appendix G.

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

APPENDIX

# A  GENERALIZATION TO BASE LLMS

## A.1  TRAINING SETUP

To assess the generalization of our ambiguity-aware training beyond instruction-tuned models, we also experiment with Qwen2.5-Base. The training configuration is kept consistent with $A^2$SEARCH and $Sin$Search, except that we adopt prompts tailored for base models (Appendix G.6 and G.7).

In practice, base models are likewise able to generate multiple answers for ambiguous questions. However, unlike their instruct-tuned counterparts, they tend to undergo early entropy collapse, which restricts exploration and results in lower validation performance. To mitigate this issue, we introduce an entropy regularization term with adaptive entropy control (He et al., 2025), as detailed in the next subsection.

## A.2  ENTROPY CONTROL

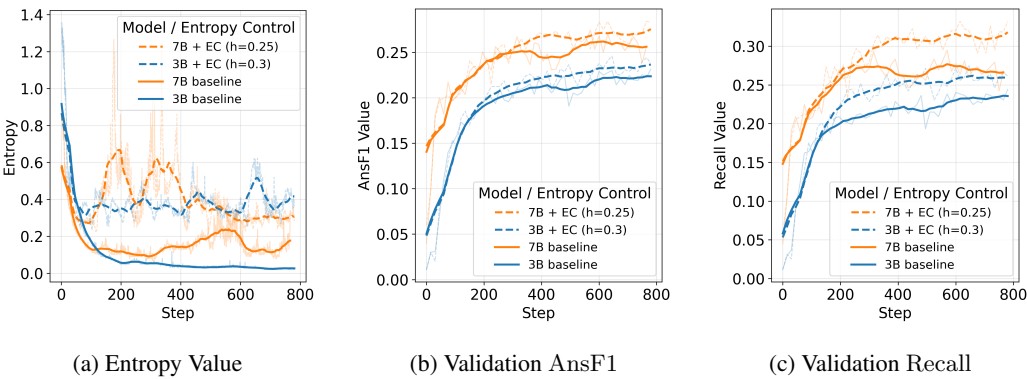

|                                       |                                  |                                 |
|:-------------------------------------:|:--------------------------------:|:-------------------------------:|
| (a) Entropy Value | (b) Validation AnsF1 | (c) Validation Recall |

Figure 5: Effect of entropy control on validation performance and the model's entropy value.

For RL training on base models, we extend the optimization objective by adding an entropy regularization term. Specifically, the policy $\pi_\theta$ is updated by maximizing

$$\mathcal{J}(\theta) = \mathbb{E}_{x \sim \mathcal{D}, \{y_i\}_{i=1}^G \sim \pi_{\theta_{\text{old}}}(\cdot|x)}$$

$$\frac{1}{G} \sum_{i=1}^G \left[ \min\left( \frac{\pi_\theta(y_i|x)}{\pi_{\theta_{\text{old}}}(y_i|x)} A_i, \text{clip}\left( \frac{\pi_\theta(y_i|x)}{\pi_{\theta_{\text{old}}}(y_i|x)}, 1-\epsilon, 1+\epsilon \right) A_i \right) + \lambda \, \mathcal{H}_\theta(x, y_i) \right],$$

where $A_i = \left( r_i - \text{mean}(\{r_j\}_{j=1}^G) \right) / \text{std}(\{r_j\}_{j=1}^G)$ is the normalized advantage of the $i$-th rollout in the group, $r_i$ is the reward, $\epsilon$ is the clipping ratio, and $\lambda$ is the entropy weight. The entropy term $\mathcal{H}_\theta(x, y_i)$ is computed as the average token-level entropy along the rollout $y_i$:

$$\mathcal{H}_\theta(x, y_i) = \frac{1}{|y_i|} \sum_{t=1}^{|y_i|} H\big(\pi_\theta(\cdot \mid x, y_{i,<t})\big), \quad H(p) = -\sum_a p(a) \log p(a).$$

This entropy term encourages exploration and mitigates over-confident predictions. As noted in prior work (He et al., 2025; Yu et al., 2025), keeping entropy within a moderate range is critical: too low leads to collapse, while too high causes unstable learning. To address the early entropy collapse we observed with base models, we adopt the adaptive entropy control method of Skywork-OR1 (He et al., 2025). This method sets a target entropy $h$ and dynamically adjusts $\lambda$: when entropy falls below $h$, $\lambda$ is increased by a small step $\delta$ (up to a maximum $\lambda_{max}$); when it rises above $h$, $\lambda$ is decreased symmetrically. In our experiments, we set $\lambda_{max} = 1\text{e}{-2}$, $\delta = 2\text{e}{-3}$, and target entropy values of 0.3 for 3B-Base and 0.25 for 7B-Base.

As shown in Figure 5, entropy control effectively stabilizes training by maintaining higher entropy levels. Models trained with entropy control achieve substantially better validation AnsF1 and Recall, confirming that controlled exploration enables faster and more effective learning.

## A.3 Results

| Multi-hop QA | HotpotQA | | 2Wiki | | MuSiQue | | Bamboogle | | Macro-Avg | |
|---|---|---|---|---|---|---|---|---|---|---|
| AnsF1/Recall@$k$ | @1 | @3 | @1 | @3 | @1 | @3 | @1 | @3 | @1 | @3 |
| A$^2$SEARCH-3B | 42.8 /$_{44.4}$ | | 56.2 /$_{58.9}$ | | 24.2 /$_{25.9}$ | | 49.3 /$_{50.4}$ | | 43.1 /$_{44.9}$ | |
| $Sin$Search-3B | 37.9 | 41.1 /$_{47.1}$ | 47.3 | 50.8 /$_{58.2}$ | 19.5 | 20.5 /$_{25.6}$ | 38.4 | 38.2 /$_{41.8}$ | 35.8 | 37.7 /$_{43.2}$ |
| A$^2$SEARCH-3B-Base | 41.7 /$_{44.5}$ | | 55.2 /$_{59.0}$ | | 25.2 /$_{28.3}$ | | 42.4 /$_{45.6}$ | | 41.1 /$_{44.4}$ | |
| $Sin$Search-3B-Base | 38.3 | 40.7 /$_{47.1}$ | 45.0 | 48.1 /$_{55.4}$ | 20.1 | 22.1 /$_{27.1}$ | 37.6 | 39.1 /$_{44.0}$ | 35.2 | 37.5 /$_{43.4}$ |
| A$^2$SEARCH-7B | 49.5 /$_{52.1}$ | | 62.3 /$_{64.4}$ | | 30.1 /$_{34.8}$ | | 51.7 /$_{53.6}$ | | 48.4 /$_{51.2}$ | |
| $Sin$Search-7B | 45.6 | 46.9 /$_{50.3}$ | 57.6 | 59.5 /$_{64.1}$ | 25.4 | 27.0 /$_{30.9}$ | 48.8 | 50.6 /$_{53.8}$ | 44.4 | 46.0 /$_{49.8}$ |
| A$^2$SEARCH-7B-Base | 47.4 /$_{50.0}$ | | 59.3 /$_{61.3}$ | | 27.0 /$_{30.8}$ | | 49.5 /$_{52.0}$ | | 45.8 /$_{48.5}$ | |
| $Sin$Search-7B-Base | 42.1 | 43.6 /$_{49.8}$ | 52.0 | 54.6 /$_{62.9}$ | 21.4 | 23.1 /$_{28.7}$ | 45.6 | 43.9 /$_{49.0}$ | 40.3 | 41.3 /$_{47.6}$ |

| General QA | NQ | | TriviaQA | | PopQA | | AmbigQA | | Macro-Avg | |
|---|---|---|---|---|---|---|---|---|---|---|
| AnsF1/Recall@$k$ | @1 | @3 | @1 | @3 | @1 | @3 | @1 | @3 | @1 | @3 |
| A$^2$SEARCH-3B | 47.3 /$_{49.7}$ | | 60.9 /$_{62.5}$ | | 48.2 /$_{50.5}$ | | 43.1 /$_{38.2}$ | | 49.9 /$_{50.2}$ | |
| $Sin$Search-3B | 40.9 | 43.3 /$_{48.2}$ | 58.0 | 59.9 /$_{64.9}$ | 43.7 | 45.0 /$_{49.3}$ | 38.2 /$_{32.8}$ | 41.2 /$_{38.6}$ | 45.2 /$_{43.9}$ | 47.4 /$_{50.2}$ |
| A$^2$SEARCH-3B-Base | 47.2 /$_{50.6}$ | | 62.5 /$_{65.2}$ | | 50.0 /$_{53.0}$ | | 44.1 /$_{40.1}$ | | 51.0 /$_{52.2}$ | |
| $Sin$Search-3B-Base | 45.4 | 46.8 /$_{50.4}$ | 59.6 | 61.6 /$_{66.2}$ | 47.4 | 49.0 /$_{53.9}$ | 41.5 /$_{35.7}$ | 43.7 /$_{40.1}$ | 48.5 /$_{47.0}$ | 50.3 /$_{52.6}$ |
| A$^2$SEARCH-7B | 51.4 /$_{54.7}$ | | 67.8 /$_{69.6}$ | | 52.5 /$_{55.6}$ | | 48.1 /$_{43.2}$ | | 55.0 /$_{55.8}$ | |
| $Sin$Search-7B | 49.3 | 49.8 /$_{51.3}$ | 66.2 | 67.0 /$_{69.2}$ | 50.5 | 51.4 /$_{53.5}$ | 44.6 /$_{38.4}$ | 45.1 /$_{39.8}$ | 52.6 /$_{51.1}$ | 53.3 /$_{53.5}$ |
| A$^2$SEARCH-7B-Base | 50.2 /$_{54.2}$ | | 65.9 /$_{67.8}$ | | 49.0 /$_{52.0}$ | | 47.3 /$_{42.8}$ | | 53.1 /$_{54.2}$ | |
| $Sin$Search-7B-Base | 46.8 | 48.3 /$_{53.2}$ | 63.8 | 64.6 /$_{69.2}$ | 47.7 | 49.2 /$_{53.9}$ | 43.0 /$_{37.1}$ | 45.5 /$_{41.8}$ | 50.3 /$_{48.8}$ | 51.9 /$_{54.5}$ |

Table 4: Evaluation results with the *Exact Match* metric on eight open-domain QA benchmarks. We report AnsF1/Recall@$k$, where $k$ denotes the number of rollouts. A$^2$SEARCH and A$^2$SEARCH-Base use a single rollout since they can generate multiple answers per attempt.

Table 4 reports the performance of A$^2$SEARCH-Base and $Sin$Search-Base. Several conclusions can be drawn. First, base models trained with our framework learn to recognize ambiguity and retrieve multiple answers, consistently outperforming their $Sin$Search counterparts across both sizes (3B and 7B) and all QA benchmarks. Second, the performance of A$^2$SEARCH-Base is broadly comparable to that of A$^2$SEARCH, with only minor degradation. We attribute this gap to the additional post-training of instruct models on tool-use data, which enhances their agentic behavior, whereas base models have not been exposed to such data.

# B TRAINING DATA CONSTRUCTION

## B.1 SAMPLING RUNTIME AND COMPUTATIONAL COST

| Model | Avg. Tool Calls | Duration |
|---|---|---|
| ReSearch-32B | 3.52 | 38.6 h |
| ReSearch-7B | 3.00 | 16.0 h |
| Search-R1-32B | 1.82 | 7.3 h |
| Search-R1-14B | 1.68 | 5.2 h |
| Search-R1-7B | 2.80 | 7.8 h |
| **Total** | — | **74.9 h** |

Table 5: Runtime for generating the sampled trajectories. "Avg. Tool Calls" denotes the average number of retrieval-tool invocations per sampled trajectory.

In our data construction pipeline, we generate a large number of trajectories in order to maximize sampling diversity and expose latent ambiguity. Concretely, we use five search models (ReSearch-7B/32B and Search-R1-7B/14B/32B), each generating 16 sampled trajectories per question over the full set of 49, 938 training questions, for a total of 3.99M trajectories. This is a one-time offline data construction step and is not required at deployment time.

All sampling experiments are conducted on a single node equipped with $4\times$H100 GPUs, using vLLM-accelerated inference (Kwon et al., 2023) and a locally served Wikipedia search index following the Search-R1 setup (Jin et al., 2025). The end-to-end runtime for evaluating all five models across the training set is summarized in Table 5. This cost is amortized over the entire training corpus and is substantially lower than manual annotation at a comparable scale. The pipeline is also highly parallelizable, and both inference throughput and retrieval latency can be further optimized in practical deployments.

## B.2 DETAILED DESCRIPTION OF THE FILTERING STEP

| ReSearch | | Search-R1 | | | *All 5 Models* |
|---|---|---|---|---|---|
| 32B | 7B | 32B | 14B | 7B | Recall@$k$ |
| 0.785 | 0.758 | 0.729 | 0.720 | 0.635 | **0.868** |

Table 6: Performance of individual models and the five-model ensemble during the trajectory generation stage. Results are reported as $\mathrm{Recall@16}$ for each model and $\mathrm{Recall@80}$ for the ensemble. The ensemble achieves higher recall, indicating greater trajectory diversity and broader coverage of reference answers.

We begin by evaluating the correctness of sampled trajectories against reference answers. We use $\mathrm{Recall@}k$ as the evaluation metric, which measures whether at least one of the $k$ sampled trajectories for a given question yields an answer semantically equivalent to the reference answer. Semantic equivalence is automatically judged by Qwen2.5-32B-Instruct (Yang et al., 2024), using the prompt described in Appendix G.1. As reported in Table 6, both individual models and the five-model ensemble achieve high $\mathrm{Recall@}k$. The ensemble further improves coverage by producing more diverse trajectories, confirming that our sampling stage provides a sufficiently rich candidate pool for subsequent processing.

The objective of the filtering stage is to discard trajectories that do not contribute novel candidate answers. Based on the criteria described in Section 3.1, each model–question pair falls into one of three categories:

- Case 1 (34.9%): All rollouts for a given question are semantically equivalent to the reference answer. These trajectories provide no novel candidates and are therefore removed.

- Case 2 (27.4%): None of the rollouts for a given question match the reference answer. This indicates that the model fails to solve the question, and the entire trajectory set is discarded.

- Case 3 (37.7%): The rollouts for a given question include both canonical and non-canonical answers. In this case, the non-canonical rollouts may represent plausible alternative answers and are retained for further verification.

To further reduce redundancy, we perform de-duplication at the answer level. Candidate answers are normalized through lower-casing, punctuation removal, and whitespace trimming, after which only one trajectory is kept for each unique normalized answer.

After filtering and de-duplication, the dataset contains $208,829$ trajectories across $33,997$ questions. This corresponds to $5.2\%$ of the original $3.99$ million trajectories and covers $68.1\%$ of the $49,938$ source questions. On average, $6.1$ distinct trajectories are retained per question. This intermediate dataset constitutes the input to the verification stage, where the validity of alternative answers is further assessed.

## B.3 DETAILED DESCRIPTION OF THE VERIFICATION STEP

This step determines whether each candidate answer in a trajectory is sufficiently supported by the retrieved evidence. We employ four proprietary LLMs as verifiers: Claude 3.5 Sonnet, Claude 3.7 Sonnet, OpenAI o3, and OpenAI o4-mini. Each verifier processes a trajectory using the prompt in Appendix G.2 and assigns one of three labels: *supported*, *partially supported*, or *not supported*. The intermediate label *partially supported* is introduced to prevent borderline cases from being

| Model | supported | partially | not supported |
|-------|-----------|-----------|---------------|
| Claude 3.5 Sonnet | 13.9 | 18.8 | 67.3 |
| Claude 3.7 Sonnet | 8.5 | 21.4 | 70.1 |
| OpenAI o3 | 13.7 | 14.9 | 71.4 |
| OpenAI o4-mini | 20.8 | 6.8 | 72.5 |

Table 7: Label distribution (%) assigned by each verifier model.

| Voting Threshold | $\eta = 1$ | $\eta = 2$ | $\eta = 3$ | $\eta = 4$ |
|------------------|------------|------------|------------|------------|
| Trajectory count | $56,986$ | $36,096$ | $19,529$ | $9,655$ |
| Percentage | $27.3\%$ | $17.3\%$ | $9.4\%$ | $4.6\%$ |
| Human Agreement | $64.0\%$ | $79.0\%$ | $96.0\%$ | $99.0\%$ |

Table 8: Number, proportion, and human agreement of trajectories labeled as *supported* under different voting thresholds $\eta$. Percentages are computed relative to the 208,829 trajectories obtained after the **Filtering** step.

over-classified as *supported*, thereby improving robustness. The distribution of labels varies systematically across verifiers. As shown in Table 7, Claude 3.7 Sonnet tends to produce more *partially supported* judgments, while o4-mini more frequently labels trajectories as *supported*. These complementary behaviors are reconciled in the subsequent majority-voting stage.

Following the notation in Section 3.1, we aggregate verifier outputs using majority voting with threshold $\eta$. Table 8 reports the number and proportion of trajectories classified as *supported* under different thresholds. To assess reliability, we additionally conduct a manual evaluation: For each $\eta$, 100 positively voted answers are randomly sampled, and two co-authors of this paper serve as annotators to judge whether they represent valid alternative answers. As expected, higher thresholds yield stricter criteria, thereby improving human agreement but reducing coverage. For example, $\eta = 4$ achieves 99% agreement but retains only 4.6% of the data. Balancing agreement with coverage, we adopt $\eta = 3$ as the default setting, which achieves a 96% agreement rate while preserving 9.4% of the trajectories (19,529 in total).

## B.4 RELIABILITY OF AN ALL-OPEN VERIFICATION PIPELINE

| Threshold | Precision | Recall | Human Agreement |
|-----------|-----------|--------|-----------------|
| $\geq 1/4$ | 0.30 | 0.99 | 0.56 |
| $\geq 2/4$ | 0.50 | 0.95 | 0.78 |
| $\geq 3/4$ | **0.80** | **0.85** | **0.96** |
| $\geq 4/4$ | 0.87 | 0.67 | 0.98 |

Table 9: Performance of open-weight verifiers under different voting thresholds, measured against the proprietary verifier ensemble and human agreement.

In our main experiments, we employ strong proprietary LLMs (Claude 3.5/3.7 and OpenAI o3/o4-mini) as verifiers in order to maximize verification reliability. However, our framework does not rely on proprietary models, and it is desirable to assess whether an entirely open verifier stack can achieve comparable agreement and coverage. To this end, we conduct an additional study using four moderate-sized open-weight LLMs that fit on a single H100 GPU: *GPT-OSS-20B*, *GPT-OSS-120B*, *Qwen3-30A3B*, and *QwQ-32B*.

We sample $25,000$ trajectories from the data construction pipeline. These trajectories may contain alternative valid answers to the same question. Each verifier assigns one of three labels to a trajectory: supported, partially supported, or not supported. In our evaluation, only the supported label is treated as a positive judgment. Using the proprietary verifier ensemble (four models with a voting threshold of $\geq 3/4$), a total of $4,651$ trajectories are marked as supported. We

then apply the open-weight verifier ensemble to the same set and compare its decisions against the proprietary ensemble.

We report two metrics: (1) *precision*, defined as the fraction of trajectories approved by the open ensemble that are also approved by the proprietary ensemble, and (2) *recall*, defined as the fraction of proprietary-approved trajectories that are also approved by the open ensemble. For each voting threshold, we additionally sample 100 open-approved trajectories and manually verify whether the retrieved evidence fully supports the answer; the proportion of such cases is reported as *Human Agreement*.

The results in Table 9 show that an all-open verifier ensemble with a $\geq 3/4$ voting threshold achieves a strong balance between precision (80%) and recall (85%) relative to the proprietary ensemble. Human evaluation further confirms that this threshold yields high-quality labels, with 96% of accepted trajectories being fully supported by the retrieved evidence. We observe that most disagreements arise from how open-weight verifiers categorize borderline cases between `supported` and `partially supported`. Some answers marked as `partially supported` are in fact fully supported, reflecting a conservative tendency that increases precision at the cost of discarding some valid but borderline cases. Overall, this analysis demonstrates that a fully open verifier stack can reliably approximate the judgments of proprietary models, enabling a fully reproducible pipeline.

## C  ADDITIONAL STATISTICS

### C.1  ANSWER COUNT DISTRIBUTION

| Answer Count | MSQ | HPQ | 2Wiki | BBG | PQ | NQ | TQ | AQ | Overall |
|---|---|---|---|---|---|---|---|---|---|
| A$^2$SEARCH-7B | | | | | | | | | |
| 1 | 52.9% | 72.2% | 78.8% | 86.4% | 75.0% | 74.2% | 84.6% | 77.1% | 75.1% |
| 2 | 25.4% | 17.9% | 17.0% | 8.0% | 16.6% | 15.3% | 8.9% | 13.6% | 15.4% |
| 3 | 14.8% | 4.5% | 1.9% | 2.4% | 4.8% | 5.2% | 3.3% | 4.3% | 5.1% |
| > 3 | 6.9% | 5.4% | 2.4% | 3.2% | 3.7% | 5.3% | 3.2% | 5.0% | 4.4% |
| Avg. | 2.26 | 1.53 | 1.33 | 1.31 | 1.42 | 1.50 | 1.31 | 1.45 | 1.51 |
| A$^2$SEARCH-3B | | | | | | | | | |
| 1 | 69.8% | 81.2% | 88.1% | 88.8% | 79.8% | 77.6% | 86.0% | 80.8% | 81.5% |
| 2 | 20.9% | 14.3% | 10.3% | 8.8% | 15.2% | 15.6% | 9.8% | 14.2% | 13.6% |
| 3 | 9.4% | 4.5% | 1.6% | 2.4% | 5.0% | 6.8% | 4.2% | 4.9% | 4.8% |
| > 3 | 0.0% | 0.0% | 0.0% | 0.0% | 0.0% | 0.1% | 0.0% | 0.1% | 0.0% |
| Avg. | 1.40 | 1.23 | 1.14 | 1.14 | 1.25 | 1.29 | 1.18 | 1.24 | 1.23 |

Table 10: Answer count distribution across benchmarks for A$^2$SEARCH.

Table 10 presents the distribution of answer counts produced by A$^2$SEARCH across individual benchmarks. We report results separately for the 7B and 3B models. The benchmark abbreviations are: MSQ (MuSiQue), HPQ (HotpotQA), BBG (Bamboogle), PQ (PopQA), TQ (TriviaQA), and AQ (AmbigQA).

For A$^2$SEARCH-7B, the model most frequently outputs a single answer (about 75.1% overall), but also produces two answers in 15.4% of cases and three or more answers in roughly 9.5% of cases combined. The higher frequency of multiple answers is especially evident on MuSiQue, where nearly half of the questions elicit more than one answer on average (Avg.2.26). In contrast, datasets such as Bamboogle (1.31) and 2Wiki (1.33) show relatively few multi-answer cases, reflecting their lower inherent ambiguity.

For A$^2$SEARCH-3B, the tendency to produce multiple answers is weaker, with 81.5% of questions receiving exactly one answer and only 13.6% receiving two answers. The overall average is 1.23 answers per question, notably lower than the 1.51 average of the 7B model. Still, the 3B model demonstrates sensitivity to dataset-specific ambiguity: for example, MuSiQue again yields the highest average count (1.40).

In summary, $A^2$SEARCH adapts to varying levels of dataset ambiguity, with the larger 7B model generating more multi-answer outputs, particularly on MuSiQue, while still maintaining a reasonable precision–recall balance. This validates that our ambiguity-aware training enables models not only to capture multiple answers when necessary, but also to refrain from over-generating when questions admit only a single correct response.

## C.2 ROLLOUT EFFICIENCY

| Model | ReSearch | | Search-R1 | | | AFM-MHQA | | $A^2$SEARCH | |
|---|---|---|---|---|---|---|---|---|---|
| Size | 7B | 32B | 3B | 7B | 32B | 3B | 7B | 3B | 7B |
| Single-rollout (Temperature=0) | | | | | | | | | |
| Tool calls | 3.17 | 3.54 | 3.19 | 3.03 | 1.97 | 2.96 | 3.31 | 2.16 | 4.14 |
| Recall@1/$rptc$ | 39.2 /0.12 | 46.2 /0.13 | 32.2 /0.10 | 36.4 /0.12 | 44.0 /0.22 | 35.5 /0.12 | 39.0 /0.12 | 44.7 /0.21 | 51.2 /0.12 |
| Multi-rollout (Temperature=0.6) | | | | | | | | | |
| Tool calls | 3.11 | 3.31 | 3.19 | 3.03 | 1.98 | 2.72 | 2.96 | - | - |
| Recall@2/$rptc$ | 45.9 /0.07 | 52.0 /0.08 | 34.9 /0.05 | 39.4 /0.06 | 48.8 /0.12 | 42.0 /0.08 | 44.7 /0.08 | - | - |
| Recall@3/$rptc$ | 49.5 /0.05 | 54.9 /0.06 | 36.5 /0.04 | 41.2 /0.05 | 51.4 /0.09 | 46.6 /0.06 | 48.2 /0.05 | - | - |

Table 11: Rollout efficiency with average tool calls, $\text{Recall}@k$, and $rptc$.

To further assess the efficiency of different methods, we report in Table 11 the average number of tool calls, together with recall at different rollout depths and the derived metric $rptc$ (recall per tool call), across four multi-hop QA benchmarks. Since each tool call directly incurs inference cost, this metric measures how effectively a model converts reasoning steps into recall gain.

Overall, we observe that multi-hop QA typically requires around three tool calls on average, but a larger number of calls does not necessarily yield better recall. For example, in the single-rollout (temperature $= 0$) setting, $A^2$SEARCH-3B requires only 2.16 calls on average while already achieving a $\text{Recall}@1$ of 44.7%, which corresponds to an $rptc$ of 0.21. This efficiency is on par with or even higher than the much larger Search-R1-32B (0.22), despite the significant gap in model scale. Similarly, $A^2$SEARCH-7B performs the highest number of calls (4.14), but this is offset by its substantially higher recall (51.2% at $k = 1$), leading to a strong $rptc$ of 0.12 that surpasses the other 7B baselines. These results indicate that $A^2$SEARCH leverages additional calls in a productive manner, rather than wasting them on uninformative exploration.

In contrast, baseline methods such as ReSearch, Search-R1, and AFM-MHQA often rely on multiple rollouts with stochastic decoding (temperature $= 0.6$) to approach the same level of recall. For instance, ReSearch-7B requires two or three rollouts to increase $\text{Recall}@k$ to the range of 45–50%, whereas $A^2$SEARCH-7B achieves over 51% recall with a single rollout. Taken together, these observations demonstrate that $A^2$SEARCH achieves a favorable trade-off between recall and tool call efficiency, scaling effectively across different model sizes.

## C.3 AMBIGUITY RATIO OF QA BENCHMARKS

To obtain a coarse estimation of the ambiguity level in existing QA benchmarks, we apply the **Filtering**, **Verification**, and **Grouping** steps described in Section 4 to trajectories generated by five baseline models (ReSearch-7B/32B and Search-R1-7B/14B/32B). For each question, we have six sampled rollouts at temperature 0.6 and automatically annotate whether the predicted answers have enough evidence to support. The resulting statistics are as follows: 19.7% of questions in MuSiQue, 8.6% in 2Wiki, 11.1% in HotpotQA, 8.0% in Bamboogle, 12.7% in NQ, 14.7% in PopQA, and 7.0% in TriviaQA exhibit valid alternative answers.

These numbers suggest that ambiguity is non-trivial across benchmarks. Among multi-hop datasets, MuSiQue and HotpotQA display the highest ambiguity rates, reflecting the inherently open-ended reasoning process required. In the general QA setting, PopQA shows the highest proportion of ambiguous cases, while TriviaQA remains relatively less ambiguous.

| Ambiguity Type | Question Example | Alternative Answers | Explanation |
|---|---|---|---|
| Under-Constrained | What was the record label of the performer of *There Goes Rhymin' Simon*? | Columbia Records; Warner Bros. | Artist was signed to different labels at different career stages. |
| Granularity Ambiguity | What is the place of birth of the director of the film *The Outsider* (2018)? | Denmark; Fredericia | Country vs. city specificity. |
| Time Sensitivity | Where is the next FIFA World Cup going to take place? | Qatar; United States, Canada, Mexico | Depends on pre-/post-2022 interpretation. |
| Evidence Conflict | How many fish species live in the river system containing the Jari River? | Over 5,600; 2,200 | Conflicting retrieved sources. |
| Multi-Item Response | Who is the child of the performer of "You Can Call Me Al"? | Lulu; Harper Simon | Performer has multiple children. |
| Open-Ended | In what ways did Kanye draw inspiration from U2, Led Zeppelin, and the performer of "Mother's Little Helper"? | Various interpretations | Subjective, multiple valid stylistic explanations. |
| Alias Variance | Who gets Blair pregnant in season 5 of the series with the episode "The Ex Files"? | Louis Grimaldi; Prince of Monaco | Different surface forms of the same entity. |

Table 12: Representative ambiguous questions and valid alternative answers.

## C.4 AMBIGUITY ANALYSIS

To better characterize the kinds of ambiguity present in our data, we first manually inspected a subset of questions with multiple valid answers and derived a taxonomy of seven recurring ambiguity types: *Under-Constrained*, *Granularity Ambiguity*, *Time Sensitivity*, *Evidence Conflict*, *Multi-Item Response*, *Open-Ended*, and *Alias Variance*. The definitions of these categories are summarized in Table 3. We also provide representative ambiguous question examples in Table 12.

# D EXPERIMENTAL SETUP

## D.1 AnsF1@$k$ ESTIMATION ALGORITHM

In practice, evaluating AnsF1@$k$ requires averaging over all possible subsets of $k$ trajectories drawn from a larger pool of $k'$ ($k' > k$) sampled trajectories. Simply reporting the best or worst case among $k'$ samples would give a biased picture of model performance. Our estimation procedure therefore computes the expected precision, recall, and F1 under uniform subsampling without replacement, which provides a more faithful measure of a model's ability to generate diverse and valid answers.

It is also worth noting that large language models are inherently stochastic. Their randomness can be amplified by sampling at a higher temperature. In our experiments, we set the temperature to 0.6, which encourages stronger diversity in rollouts and thereby increases the chance of capturing multiple answers.

---

**Algorithm 1** AnsF1@$k$ Estimation via Subsampling

---

**Require:** hits: a list of length $k'$ ($k' > k$), where each entry is either the identifier of the reference answer matched by a predicted answer, or $\perp$ if no match
**Require:** $g$: total number of reference answers
**Require:** $k$: number of trajectories to subsample
**Ensure:** Expected Precision@$k$, Recall@$k$, and F1@k
  1: For each reference answer $a$, compute its multiplicity $m_a$ in hits
  2: denom $\leftarrow \binom{k'}{k}$                   ▷ number of size-$k$ subsets (uniform sampling without replacement)
  3: $\text{sum}_p \leftarrow 0$, $\text{sum}_r \leftarrow 0$, $\text{sum}_{f1} \leftarrow 0$
  4: **for all** size-$k$ subsets $S \subset$ hits **do**
  5:     $s \leftarrow$ number of positive predictions in $S$ (i.e., $|\{x \in S \mid x \neq \perp\}|$)
  6:     $u \leftarrow$ number of unique reference answers covered in $S$ (i.e., $|\{x \in S \mid x \neq \perp\}|_{\text{unique}}|$)
  7:     $p \leftarrow s/k$                                                                  ▷ precision
  8:     $r \leftarrow u/g$                                                                  ▷ recall
  9:     **if** $p > 0$ and $r > 0$ **then**
 10:         $f1 \leftarrow \frac{2pr}{p+r}$                           ▷ equivalently $f1 = \frac{2su}{gs+ku}$
 11:     **else**
 12:         $f1 \leftarrow 0$
 13:     **end if**
 14:     $\text{sum}_p \leftarrow \text{sum}_p + p$
 15:     $\text{sum}_r \leftarrow \text{sum}_r + r$
 16:     $\text{sum}_{f1} \leftarrow \text{sum}_{f1} + f1$
 17: **end for**
 18: $\mathbb{E}[\text{Precision@}k] \leftarrow \text{sum}_p/\text{denom}$
 19: $\mathbb{E}[\text{Recall@}k] \leftarrow \text{sum}_r/\text{denom}$
 20: $\mathbb{E}[\text{F1@}k] \leftarrow \text{sum}_{f1}/\text{denom}$
 21: **return** $\left(\mathbb{E}[\text{Precision@}k],\ \mathbb{E}[\text{Recall@}k],\ \mathbb{E}[\text{F1@}k]\right)$

---

Algorithm 1 summarizes the procedure. Given a list of $k'$ predicted answers hits, where each element indicates either a matched reference answer or $\perp$, the algorithm enumerates all size-$k$ subsets, computes precision, recall, and F1 for each, and then averages them with equal weight. This yields unbiased estimates of AnsF1@$k$ and Recall@$k$ that account for both prediction correctness and coverage of distinct reference answers.

# E  ABLATION EXPERIMENTS

## E.1  THE ROLE OF $\alpha$ IN REWARD DESIGN

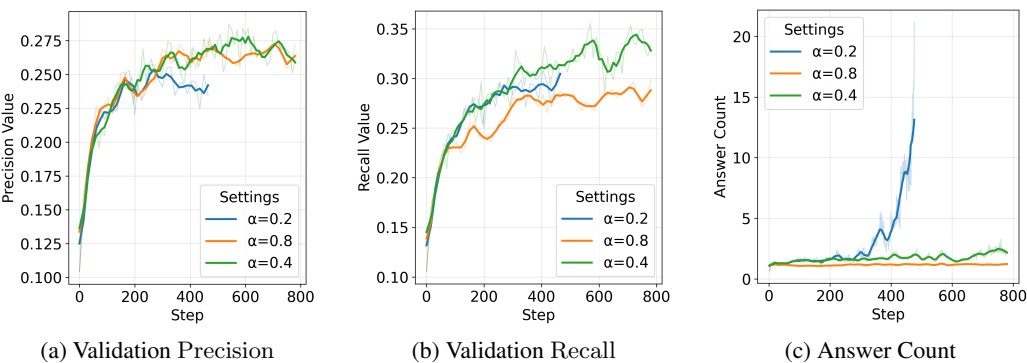

|                          |                       |                     |
| ------------------------ | --------------------- | ------------------- |
| (a) Validation Precision | (b) Validation Recall | (c) Answer Count    |

Figure 6: Effect of $\alpha$ in reward design on validation performance and the model's answer count.

Our reward is defined as

$$R(q, a\hat{n}s) = \begin{cases} 0, & \text{if format invalid,} \\ 0.1, & \text{if format valid and hits} = 0, \\ 1 - \alpha\,(1 - \text{AnsF1}), & \text{if format valid and hits} > 0, \end{cases}$$

with $\alpha \in [0, 1]$ so that $R \in [1 - \alpha, 1]$ when hits $> 0$. The parameter $\alpha$ governs the extent to which imperfect AnsF1 is penalized, thereby mediating the balance between precision and recall.

When $\alpha$ is set to a relatively large value, such as $0.8$, the reward approximates $R = 0.2 + 0.8 \cdot \text{AnsF1}$ and becomes tightly coupled to AnsF1. Since AnsF1 is particularly sensitive to precision, the model is strongly discouraged from producing many answers: additional candidates reduce precision, lower AnsF1, and thus incur substantial penalty. This tendency is further amplified by the data distribution itself, as more than half of the training examples contain only a single valid answer. Under such conditions, optimizing for reward with large $\alpha$ naturally aligns with producing highly precise, single-answer outputs. Empirically, this effect is evident in Figure 6c, where we train A$^2$SEARCH-7B with different $\alpha$ settings. We can find that $\alpha = 0.8$ leads the model to converge to nearly one answer throughout training.

In contrast, when $\alpha$ is small, such as $0.2$, the reward is bounded below by $0.8$ regardless of precision, yielding $R = 0.8 + 0.2 \cdot \text{AnsF1}$. In this regime, the incentive to maintain precision nearly vanishes, and the model quickly learns to enumerate many answers to ensure at least one match. This behavior inflates recall and results in rapidly increasing answer counts, as shown in Figure 6c.

For intermediate values, such as $\alpha = 0.4$, the penalty balances the two extremes: too few outputs reduce recall, while too many reduce precision, and the model stabilizes by producing a moderate number of answers with a controlled trade-off. This behavior is reflected in the validation trends in Figures 6a–6c, where $\alpha = 0.4$ achieves both stability and a reasonable precision–recall balance.

## E.2  ROLLOUT SIZE

One factor that may influence training effectiveness is the choice of rollout size. Intuitively, increasing the number of rollouts per step can enhance the diversity of sampled trajectories, thereby improving the likelihood of discovering high-quality reasoning paths and providing denser reward signals. To determine an appropriate rollout size for training, we conduct an ablation study on A$^2$SEARCH-7B, training under three different rollout settings (8, 16, and 32) for two epochs.

As shown in Figure 7, larger rollout sizes could lead to better validation performance in terms of Precision, Recall, and AnsF1. However, the performance gap between rollout sizes 16 and 32 is relatively small. Considering the trade-off between performance gains and training efficiency, we adopt rollout size 16 as the default configuration for the experiments reported in the main paper.

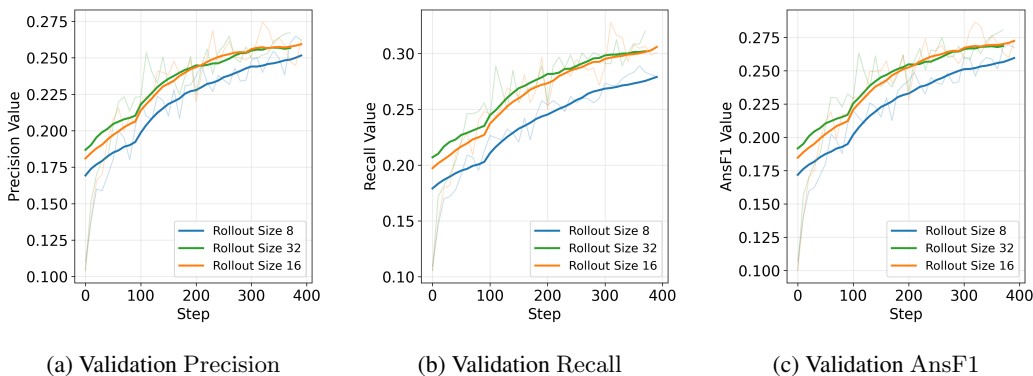

(a) Validation Precision        (b) Validation Recall        (c) Validation AnsF1

Figure 7: Effect of rollout size on validation performance.

### E.3 SAMPLING TEMPERATURE

| Temperature | 0.2 | | 0.4 | | 0.6 | | 0.8 | | 1.0 | |
|---|---|---|---|---|---|---|---|---|---|---|
| Metric@3 | AnsF1 | Recall | AnsF1 | Recall | AnsF1 | Recall | AnsF1 | Recall | AnsF1 | Recall |
| ReSearch-7B | 24.5 | 29.5 | 25.1 | 30.5 | 24.9 | 31.1 | 24.3 | 30.8 | 23.2 | 30.2 |
| Search-R1-7B | 20.3 | 21.7 | 20.8 | 23.0 | 20.7 | 23.6 | 20.8 | 23.9 | 20.8 | 24.6 |
| AFM-MHQA-7B | 22.9 | 28.3 | 22.8 | 28.4 | 21.5 | 27.4 | 19.5 | 25.5 | 17.4 | 23.7 |
| Average | 22.6 | 26.5 | 22.9 | 27.3 | 22.4 | **27.4** | 21.5 | 26.7 | 20.5 | 26.2 |

Table 13: Ablation study of sampling temperature on the MuSiQue benchmark. We report AnsF1@3 and Recall@3 for three baseline models across different temperatures.

Since baseline models such as ReSearch, Search-R1, and AFM are trained to produce a single answer per question, we compute their multi-answer scores by sampling multiple rollouts. This introduces an additional variable, the sampling temperature, which directly influences randomness. A higher temperature typically increases diversity, which may improve recall but often at the cost of precision. The degree of sensitivity to temperature also varies across models, depending on their training objectives. To ensure fair comparison, we therefore conduct an ablation study to identify the most suitable temperature setting for evaluation.

Concretely, we evaluate three baseline models (ReSearch-7B, Search-R1-7B, and AFM-MHQA-7B) on the MuSiQue benchmark. For each model, we sample six rollouts at temperatures ranging from 0.2 to 1.0 and compute both AnsF1@3 and Recall@3. The results are summarized in Table 13. From the table, we observe that Search-R1-7B exhibits relatively stable performance across temperatures, with recall steadily improving as the temperature increases, while F1 remains largely unchanged. In contrast, AFM-MHQA-7B shows a sharper decline in both F1 and recall as temperature rises, suggesting that it is more sensitive to randomness. ReSearch-7B achieves its best balance around 0.4–0.6, where recall is highest (31.1 at $T = 0.6$) without significant loss in F1. Averaged across all three models, recall peaks at $T = 0.6$, while F1 remains competitive compared to lower temperatures. Based on these findings, we select $T = 0.6$ as the default sampling temperature in our experiments, as it provides a fair trade-off between recall and F1 while allowing each baseline model to perform near its best.

### E.4 EVALUATING WITH LMJUDGE

While our main evaluation relies on the *Exact Match* (EM) metric, it only measures lexical overlap and cannot capture semantic similarity. This raises a potential risk: since natural language often admits multiple surface forms, A$^2$SEARCH could appear to achieve higher recall simply by generating paraphrased variants of reference answers, thereby "hacking" EM. By contrast, baseline models, especially those trained to produce a single answer, would need multiple samples to generate such

| Model | HotpotQA | | 2Wiki | | MuSiQue | | Bamboogle | | Macro-Avg | |
|---|---|---|---|---|---|---|---|---|---|---|
| AnsF1/Recall@$k$ | @1 | @3 | @1 | @3 | @1 | @3 | @1 | @3 | @1 | @3 |
| **Models with 3B Parameters** | | | | | | | | | | |
| DirectGen-3B | 25.3 | 26.1 /$_{29.6}$ | 27.5 | 28.3 /$_{31.4}$ | 6.6 | 7.2 /$_{9.7}$ | 7.2 | 8.7 /$_{11.4}$ | 16.6 | 17.6 /$_{20.5}$ |
| Naive-RAG-3B | 43.2 | 43.1 /$_{45.5}$ | 29.3 | 29.9 /$_{31.7}$ | 11.4 | 11.9 /$_{13.3}$ | 23.2 | 22.8 /$_{24.4}$ | 26.8 | 26.9 /$_{28.7}$ |
| Iter-RetGen-3B | 45.8 | 47.0 /$_{49.7}$ | 30.6 | 31.7 /$_{34.6}$ | 13.2 | 14.0 /$_{16.2}$ | 22.4 | 23.7 /$_{27.2}$ | 28.0 | 29.1 /$_{31.9}$ |
| IRCoT-3B | 52.6 | 56.5 /$_{67.3}$ | 39.0 | 43.2 /$_{57.6}$ | 15.9 | 18.5 /$_{25.5}$ | 37.6 | 41.9 /$_{52.4}$ | 36.3 | 40.0 /$_{50.7}$ |
| Search-R1-3B | 54.6 | 56.5 /$_{60.8}$ | 45.3 | 47.9 /$_{53.8}$ | 22.2 | 24.2 /$_{28.7}$ | 47.2 | 47.2 /$_{50.5}$ | 42.3 | 43.9 /$_{48.5}$ |
| AFM-MHQ-3B | 41.5 | **63.2** /**$_{72.6}$** | 52.3 | 55.0 /**$_{67.6}$** | 28.2 | 31.5 /**$_{41.8}$** | 56.8 | 56.5 /**$_{65.3}$** | 44.7 | 51.6 /**$_{61.8}$** |
| *Sin*Search | 56.1 | 59.9 /$_{67.7}$ | 51.9 | 55.5 /$_{63.0}$ | 28.6 | 31.1 /$_{38.6}$ | 52.0 | 38.2 /$_{41.8}$ | 47.2 | 46.2 /$_{53.2}$ |
| *Abg*Search | 43.8 /$_{48.4}$ | | 33.5 /$_{40.0}$ | | 15.6 /$_{17.3}$ | | 33.7 /$_{34.4}$ | | 31.5 /$_{35.0}$ | |
| A$^2$SEARCH-3B | 62.4 /$_{64.9}$ | | **60.9** /$_{64.0}$ | | **36.3** /$_{39.4}$ | | **61.4** /$_{62.4}$ | | **55.3** /$_{57.2}$ | |
| **Models with 7 $\sim$ 32B Parameters** | | | | | | | | | | |
| DirectGen-7B | 31.8 | 32.5 /$_{35.9}$ | 27.8 | 29.3 /$_{33.5}$ | 12.4 | 13.2 /$_{16.4}$ | 24.0 | 22.8 /$_{25.9}$ | 24.0 | 24.5 /$_{27.9}$ |
| Naive-RAG-7B | 53.7 | 54.7 /$_{57.2}$ | 35.7 | 36.5 /$_{38.5}$ | 15.7 | 16.5 /$_{18.7}$ | 39.2 | 40.0 /$_{42.9}$ | 36.1 | 36.9 /$_{39.3}$ |
| Iter-RetGen-7B | 55.3 | 57.3 /$_{60.8}$ | 37.0 | 38.8 /$_{42.5}$ | 19.0 | 20.1 /$_{23.5}$ | 36.8 | 39.0 /$_{41.9}$ | 37.0 | 38.8 /$_{42.2}$ |
| IRCoT-7B | 62.0 | 66.3 /$_{76.7}$ | 44.2 | 48.4 /$_{60.0}$ | 18.0 | 20.6 /$_{27.8}$ | 51.2 | 53.0 /$_{64.2}$ | 43.9 | 47.1 /$_{57.2}$ |
| ReSearch-7B | 64.4 | 66.8 /$_{74.3}$ | 55.5 | 58.8 /$_{68.8}$ | 35.3 | 39.2 /$_{47.9}$ | 59.2 | 59.8 /$_{66.2}$ | 53.6 | 56.2 /$_{64.3}$ |
| Search-R1-7B | 62.3 | 63.5 /$_{66.9}$ | 46.3 | 48.9 /$_{54.6}$ | 30.0 | 30.9 /$_{35.5}$ | 56.0 | 53.8 /$_{56.7}$ | 48.7 | 49.3 /$_{53.4}$ |
| AFM-MHQ-7B | 67.8 | 68.3 /$_{75.5}$ | 55.1 | 56.6 /$_{66.6}$ | 33.9 | 34.8 /$_{43.1}$ | 60.8 | 63.5 /$_{72.9}$ | 54.4 | 55.8 /$_{64.3}$ |
| Search-R1-14B | 67.4 | 68.4 /$_{73.1}$ | 55.5 | 57.6 /$_{63.4}$ | 36.9 | 39.2 /$_{45.5}$ | 64.0 | 67.0 /$_{69.9}$ | 55.9 | 58.1 /$_{62.9}$ |
| Search-R1-32B | 66.3 | 68.2 /$_{73.2}$ | 57.5 | 59.9 /$_{67.4}$ | 35.1 | 37.5 /$_{44.5}$ | 63.2 | 67.8 /$_{72.4}$ | 55.9 | 58.4 /$_{64.4}$ |
| ReSearch-32B | 70.9 | **72.2** /**$_{77.9}$** | 60.3 | 63.8 /**$_{72.3}$** | 39.8 | 43.5 /$_{51.4}$ | **72.0** | 71.2 /**$_{75.7}$** | 60.7 | **62.7** /**$_{69.3}$** |
| *Sin*Search | 65.9 | 67.4 /$_{71.6}$ | 62.8 | 64.7 /$_{69.7}$ | 37.9 | 40.2 /$_{46.3}$ | 63.2 | 63.8 /$_{66.9}$ | 57.5 | 59.0 /$_{63.6}$ |
| *Abg*Search | 58.7 /$_{63.7}$ | | 43.8 /$_{50.2}$ | | 26.0 /$_{30.2}$ | | 51.4 /$_{53.6}$ | | 44.9 /$_{49.4}$ | |
| A$^2$SEARCH-7B | 71.2 /$_{74.9}$ | | **67.8** /$_{69.8}$ | | **44.7** /$_{51.8}$ | | 67.2 /$_{68.8}$ | | **62.7** /$_{66.3}$ | |

Table 14: Main results on four multi-hop QA benchmarks under the *LMJudge* metric. We report AnsF1/Recall@$k$ with $k$ rollouts. For *Abg*Search and A$^2$SEARCH, only @1 is reported, reflecting their ability to produce multiple answers within a single rollout. For the remaining baselines, where each rollout generates only one answer and thus AnsF1@1 = Recall@1, we additionally include AnsF1/Recall@3 to evaluate their performance when more rollouts are available. The best result in each comparison group is shown in **bold**, and the second best is underlined.

variants, which could exaggerate the apparent recall advantage of A$^2$SEARCH. In such cases, EM may not fully reflect whether models actually resolve true ambiguity.

It is worth noting, however, that this issue does not affect training. When computing the hits for AnsF1, we only count distinct reference answers. If the model outputs multiple variants that all match the same reference answer or its aliases, the hits count remains one, and precision is penalized accordingly (e.g., predicting two synonyms for one reference yields only one hit and $50\%$ precision). This design prevents the model from exploiting lexical overlap during training.

To address the evaluation limitation, we complement EM-based scores with an LMJudge method, which measures semantic equivalence. Specifically, we prompt Qwen2.5-32B-Instruct (Appendix G.1) to judge whether each predicted answer semantically matches a reference answer. Based on these judgments, we recompute AnsF1 and Recall.

The results are reported in Table 14 (multi-hop QA benchmarks) and Table 15 (general QA benchmarks, including AmbigQA). We find the conclusions are highly consistent with those obtained under EM:

(1) Agentic search models consistently outperform prompting- and RAG-based baselines, with especially clear gains on multi-hop datasets.

(2) Even with a single greedy decoding rollout, A$^2$SEARCH matches or surpasses the Recall@3 of baselines and even outperforms larger 32B models on several benchmarks.

(3) A$^2$SEARCH achieves the best AnsF1 on multi-hop QA benchmarks, striking a strong balance between precision and recall.

| Model | NQ | | TriviaQA | | PopQA | | AmbigQA | | Macro-Avg | |
|---|---|---|---|---|---|---|---|---|---|---|
| AnsF1/Recall@$k$ | @1 | @3 | @1 | @3 | @1 | @3 | @1 | @3 | @1 | @3 |
| **Models with 3B Parameters** | | | | | | | | | | |
| DirectGen-3B | 25.9 | 26.8 /31.3 | 41.7 | 42.4 /46.9 | 18.7 | 18.7 /15.6 | 18.7 /15.6 | 20.7 /19.5 | 26.3 /25.5 | 27.2 /28.3 |
| Naive-RAG-3B | 59.2 | 59.8 /61.7 | 71.4 | 71.7 /73.0 | 49.1 | 49.5 /50.6 | 46.3 /39.4 | 47.3 /41.1 | 56.5 /54.8 | 57.1 /56.6 |
| Iter-RetGen-3B | 60.5 | 61.7 /64.2 | 73.1 | 73.8 /75.6 | 50.3 | 50.8 /52.5 | 48.2 /41.1 | 49.3 /43.2 | 58.0 /56.3 | 58.9 /58.9 |
| IRCoT-3B | 61.1 | 64.1 /72.8 | 70.2 | 72.8 /79.1 | 49.3 | 51.8 /58.2 | 46.7 /39.9 | 51.3 /48.7 | 56.8 /55.1 | 60.0 /64.7 |
| Search-R1-3B | 61.8 | 62.6 /65.2 | 72.4 | 73.0 /75.6 | 50.6 | 51.2 /52.7 | 47.2 /40.3 | 48.4 /42.8 | 58.0 /56.3 | 58.8 /59.1 |
| AFM-MHQ-3B | 60.9 | 62.4 /71.9 | 72.2 | 74.1 /82.1 | 47.7 | 47.7 /55.6 | 47.7 /40.8 | 51.8 /49.6 | 57.1 /55.4 | 59.0 /64.8 |
| $Sin$Search | 59.6 | 61.9 /67.5 | 70.0 | 72.3 /77.7 | 48.4 | 49.8 /54.4 | 46.1 /39.3 | 49.6 /45.5 | 56.0 /54.3 | 58.4 /61.3 |
| $Abg$Search | 59.1 /64.6 | | 67.9 /70.1 | | 43.6 /47.2 | | 48.6 /43.5 | | 54.8 /56.4 | |
| A$^2$SEARCH-3B | 65.1 /67.9 | | 73.3 /75.1 | | 51.5 /53.9 | | 51.4 /45.0 | | 60.3 /60.5 | |
| **Models with 7 ∼ 32B Parameters** | | | | | | | | | | |
| DirectGen-7B | 35.9 | 37.3 /41.7 | 55.7 | 56.6 /59.9 | 20.9 | 21.5 /24.2 | 26.3 /21.9 | 28.7 /25.9 | 34.7 /33.6 | 36.0 /37.9 |
| Naive-RAG-7B | 66.9 | 67.7 /69.9 | 76.6 | 77.1 /78.2 | 52.6 | 53.3 /55.2 | 51.2 /43.8 | 52.7 /46.1 | 61.8 /59.9 | 62.7 /62.3 |
| Iter-RetGen-7B | 67.4 | 68.6 /71.9 | 78.4 | 79.0 /80.4 | 52.8 | 53.7 /56.1 | 52.3 /44.7 | 54.1 /47.7 | 62.7 /60.8 | 63.8 /64.0 |
| IRCoT-7B | 65.5 | 68.8 /76.2 | 74.2 | 76.2 /80.8 | 51.4 | 54.1 /60.2 | 49.9 /42.7 | 54.6 /50.5 | 60.3 /58.5 | 63.4 /66.9 |
| ReSearch-7B | 65.8 | 67.9 /74.6 | 77.2 | 79.6 /84.7 | 50.9 | 52.6 /58.2 | 50.3 /42.9 | 55.2 /50.7 | 61.1 /59.2 | 63.8 /67.1 |
| Search-R1-7B | 65.6 | 66.4 /68.5 | 77.1 | 77.7 /79.7 | 50.8 | 51.7 /53.2 | 50.4 /43.0 | 51.9 /45.4 | 60.9 /59.1 | 61.9 /61.7 |
| AFM-MHQ-7B | 66.2 | 68.3 /75.3 | 77.8 | 79.8 /84.7 | 50.9 | 52.6 /59.0 | 51.4 /43.9 | 55.4 /51.2 | 61.6 /59.7 | 64.0 /67.6 |
| Search-R1-14B | 68.4 | 69.1 /72.0 | 79.2 | 80.1 /84.2 | 54.6 | 55.6 /59.2 | 52.8 /45.2 | 54.8 /48.6 | 63.8 /61.8 | 64.9 /66.0 |
| Search-R1-32B | 68.5 | 69.9 /73.5 | 82.3 | 83.5 /86.3 | 54.1 | 55.7 /59.5 | 52.7 /45.2 | 55.5 /49.6 | 64.4 /62.5 | 66.2 /67.2 |
| ReSearch-32B | 69.1 | 71.4 /76.6 | 81.8 | 84.8 /88.4 | 53.8 | 55.7 /60.4 | 55.1 /47.0 | 57.8 /52.1 | 64.9 /62.8 | 67.4 /69.4 |
| $Sin$Search | 67.5 | 68.1 /69.8 | 79.0 | 79.8 /82.1 | 55.5 | 56.5 /58.6 | 52.4 /44.7 | 53.4 /46.5 | 63.6 /61.6 | 64.5 /64.3 |
| $Abg$Search | 66.4 /72.8 | | 78.4 /81.5 | | 52.9 /58.2 | | 55.7 /50.4 | | 63.4 /65.7 | |
| A$^2$SEARCH-7B | 70.4 /74.0 | | 81.3 /83.4 | | 57.3 /60.6 | | 56.6 /50.3 | | 66.4 /67.1 | |

Table 15: Main results with the *LMJudge* metric on four general QA benchmarks, using the same notations as Table 14. For AmbigQA, where questions may have multiple reference answers, AnsF1@1 and Recall@1 are not equivalent in this setting, and both are therefore reported.

(4) $Abg$Search performs well on AmbigQA but fails to generalize, whereas A$^2$SEARCH, trained without AmbigQA data, surpasses it even on AmbigQA.

Overall, these results demonstrate that our findings are not artifacts of lexical metrics. Instead, they confirm that A$^2$SEARCH genuinely learns to resolve ambiguity, thereby validating both the effectiveness of our training approach and the robustness of our experimental conclusions.

## E.5 ADDITIONAL RESULTS ON LLAMA-BASED MODELS

To examine whether our ambiguity-aware RL framework generalizes beyond the Qwen backbone, we additionally trained A$^2$Search-LMA-3B using *Llama 3.2 Instruct 3B* under exactly the same training setup as used for A$^2$Search on Qwen models. For comparison, we also trained a matched single-answer baseline, denoted $Sin$Search-LMA-3B, using the same single-answer supervision and Exact Match reward applied in our Qwen-based baselines.

The evaluation results are listed in Table 16. Overall, the Llama-based results exhibit trends consistent with the Qwen-based experiments. A$^2$Search-LMA-3B achieves competitive performance relative to Qwen models of similar size, demonstrating that the ambiguity-aware RL objective transfers well across backbone architectures. Furthermore, despite using only a single rollout, A$^2$Search-LMA-3B typically achieves a high recall that approaches or exceeds that of $Sin$Search-LMA-3B, which is computed with three rollouts. These observations support the conclusion that the proposed framework is backbone-agnostic and broadly applicable across model families.

| Multi-hop QA | HotpotQA | | 2Wiki | | MuSiQue | | Bamboogle | | Macro-Avg | |
|---|---|---|---|---|---|---|---|---|---|---|
| AnsF1/Recall@$k$ | @1 | @3 | @1 | @3 | @1 | @3 | @1 | @3 | @1 | @3 |
| A$^2$SEARCH-3B | 42.8/44.4 | | 56.2/58.9 | | 24.2/25.9 | | 49.3/50.4 | | 43.1/44.9 | |
| SinSearch-3B | 37.9 | 41.1/47.1 | 47.3 | 50.8/58.2 | 19.5 | 20.5/25.6 | 38.4 | 38.2/41.8 | 35.8 | 37.7/43.2 |
| A$^2$SEARCH-7B | 49.5/52.1 | | 62.3/64.4 | | 30.1/34.8 | | 51.7/53.6 | | 48.4/51.2 | |
| SinSearch-7B | 45.6 | 46.9/50.3 | 57.6 | 59.5/64.1 | 25.4 | 27.0/30.9 | 48.8 | 50.6/53.8 | 44.4 | 46.0/49.8 |
| A$^2$SEARCH-LMA-3B | 42.2/43.6 | | 53.4/55.3 | | 22.6/24.8 | | 51.1/52.0 | | 42.3/43.9 | |
| SinSearch-LMA-3B | 36.2 | 38.7/45.7 | 48.2 | 49.9/55.1 | 16.7 | 18.7/23.8 | 39.2 | 43.4/53.1 | 35.1 | 37.7/44.4 |

| General QA | NQ | | TriviaQA | | PopQA | | AmbigQA | | Macro-Avg | |
|---|---|---|---|---|---|---|---|---|---|---|
| AnsF1/Recall@$k$ | @1 | @3 | @1 | @3 | @1 | @3 | @1 | @3 | @1 | @3 |
| A$^2$SEARCH-3B | 47.3/49.7 | | 60.9/62.5 | | 48.2/50.5 | | 43.1/38.2 | | 49.9/50.2 | |
| SinSearch-3B | 40.9 | 43.3/48.2 | 58.0 | 59.9/64.9 | 43.7 | 45.0/49.3 | 38.2/32.8 | 41.2/38.6 | 45.2/43.9 | 47.4/50.2 |
| A$^2$SEARCH-7B | 51.4/54.7 | | 67.8/69.6 | | 52.5/55.6 | | 48.1/43.2 | | 55.0/55.8 | |
| SinSearch-7B | 49.3 | 49.8/51.3 | 66.2 | 67.0/69.2 | 50.5 | 51.4/53.5 | 44.6/38.4 | 45.1/39.8 | 52.6/51.1 | 53.3/53.5 |
| A$^2$SEARCH-LMA-3B | 46.8/49.6 | | 62.0/63.2 | | 47.7/50.7 | | 43.4/38.7 | | 50.0/50.6 | |
| SinSearch-LMA-3B | 41.8 | 43.2/49.4 | 55.9 | 58.6/65.7 | 46.2 | 48.5/51.6 | 38.3/33.0 | 42.2/39.9 | 45.6/44.2 | 48.1/51.7 |

Table 16: Evaluation results with the *Exact Match* metric on eight open-domain QA benchmarks. We report AnsF1/Recall@$k$, where $k$ denotes the number of rollouts. A$^2$SEARCH uses a single rollout since they can generate multiple answers per attempt. A$^2$SEARCH-LMA uses *LLaMa-3.2-Instruct* as the backbone LLM for RL training.

## E.6 DISENTANGLING THE SOURCES OF A$^2$SEARCH'S GAINS

To better understand which components of A$^2$SEARCH are responsible for its improvements, we train several ablated variants that isolate the effects of (i) ambiguity-aware training data, (ii) the AnsF1 reward, and (iii) the ability to output multiple answers. All variants use the same backbone (Qwen2.5 7B Instruct) and follow the identical training setup unless otherwise stated.

**Ablation Variants.** We evaluate three additional models:

- **RECALLSearch-7B**: trained on single-answer data with a recall-oriented EM-Recall reward (reward = 1 if *any* generated answer matches the gold answer). The model is instructed to output up to three answers. This variant isolates the effect of "outputting multiple guesses" without ambiguity-aware data or the AnsF1 reward.

- **SIN*Search-7B**: trained on ambiguity-aware multi-answer data but with a single-answer EM reward and forced single-answer decoding. This variant tests whether access to multi-answer supervision alone (without multi-answer output or AnsF1 reward) can improve ambiguity handling.

- **A$^2$SINSearch-7B**: trained with the AnsF1 reward but on single-answer data, producing an arbitrary number of answers. This variant isolates whether AnsF1 alone, in the absence of multi-answer data, can induce ambiguity-aware behavior.

**Answer Count Analysis.** As shown in Table 17, across all datasets, SIN*Search-7B always outputs exactly one answer, confirming that multi-answer supervision alone does not induce ambiguity-aware behavior if the model is forced into single-answer decoding. A$^2$SINSearch-7B outputs slightly more than one answer on average, but the increase is marginal, indicating that AnsF1 alone does not meaningfully encourage the exploration of alternative interpretations. RECALLSearch-7B always outputs exactly three answers by construction, amplifying recall mechanically but without regard for evidence or ambiguity. Only A$^2$SEARCH-7B adapts its answer count to the dataset: it outputs more answers on datasets with higher inherent ambiguity and fewer answers where a single interpretation is sufficient. This adaptive behavior is a key indicator that the model is detecting underlying ambiguity rather than emitting guesses.

| Model | MSQ | HPQ | 2Wiki | BBG | PQ | NQ | TQ | AQ |
|---|---|---|---|---|---|---|---|---|
| Answer Count Distribution Across Models | | | | | | | | |
| SIN*Search-7B | 1.00 | 1.00 | 1.00 | 1.00 | 1.00 | 1.00 | 1.00 | 1.00 |
| $A^2$SINSearch-7B | 1.13 | 1.07 | 1.03 | 1.01 | 1.10 | 1.09 | 1.06 | 1.11 |
| RECALLSearch-7B | 3.00 | 3.00 | 3.00 | 3.00 | 3.00 | 3.00 | 3.00 | 3.00 |
| $A^2$SEARCH-7B | 2.26 | 1.53 | 1.33 | 1.31 | 1.42 | 1.50 | 1.31 | 1.45 |

Table 17: Answer count distribution across eight benchmarks for four ablation variants and $A^2$SEARCH-7B.

| Multi-hop QA | HotpotQA | 2Wiki | MuSiQue | Bamboogle |
|---|---|---|---|---|
| SIN*Search-7B | 47.2 | 58.4 | 26.5 | 50.4 |
| RECALLSearch-7B | 29.6 / 20.6 / 55.6 | 39.3 / 28.6 / 68.2 | 18.9 / 13.5 / 34.3 | 32.2 / 22.1 / 61.6 |
| $A^2$SINSearch-7B | 46.9 / 46.7 / 47.2 | 57.6 / 57.3 / 58.2 | 25.5 / 25.3 / 25.8 | 49.6 / 48.8 / 49.6 |
| $A^2$SEARCH-7B | 49.5 / 48.6 / 52.1 | 62.3 / 61.5 / 64.4 | 30.1 / 28.5 / 34.8 | 51.7 / 50.9 / 53.6 |
| **General QA** | **NQ** | **TriviaQA** | **PopQA** | **AmbigQA** |
| SIN*Search-7B | 49.8 | 67.6 | 50.5 | 45.0 / 64.8 / 38.8 |
| RECALLSearch-7B | 35.1 / 25.1 / 63.2 | 49.5 / 39.2 / 75.0 | 37.3 / 28.2 / 61.1 | 38.3 / 36.4 / 49.8 |
| $A^2$SINSearch-7B | 49.0 / 48.7 / 49.7 | 65.1 / 65.0 / 65.4 | 49.0 / 48.6 / 49.9 | 44.6 / 62.8 / 38.8 |
| $A^2$SEARCH-7B | 51.4 / 50.1 / 54.7 | 67.8 / 67.1 / 69.6 | 52.5 / 51.1 / 55.6 | 48.1 / 65.2 / 43.2 |

Table 18: Ablation study results on eight QA benchmarks. We report *AnsF1@1* for single-answer models and *AnsF1 / Precision / Recall* at @1 for the models that produce multiple answers.

**Performance Across Eight Benchmarks.** From the results reported in Table 18, we can conclude many findings. When evaluated on standard multi-hop and open-domain QA benchmarks, SIN*Search-7B underperforms due to its single-answer restriction. These datasets contain latent ambiguity, and a single deterministic prediction is often insufficient. RECALLSearch-7B achieves artificially high recall but suffers from extremely low precision, as many of its additional answers are unsupported or speculative. This behavior is consistent with reward hacking: the model increases recall by emitting paraphrases or guesses, rather than identifying genuine alternative answers.

$A^2$SINSearch-7B performs similarly to single-answer baselines and does not show meaningful gains in recall, further confirming that $\mathrm{AnsF1}$ without ambiguity-aware supervision is insufficient for uncovering alternative valid answers.

In contrast, $A^2$SEARCH-7B improves both precision and recall simultaneously across all benchmarks. This joint improvement is not achievable by any of the ablated variants, indicating that the full combination of (i) ambiguity-aware training data, (ii) $\mathrm{AnsF1}$ reward, and (iii) flexible multi-answer decoding is necessary to capture genuine ambiguity.

**Analysis of RECALLSearch-7B.** To understand why RECALLSearch-7B achieves high recall but low precision, we classify the sources of its multi-answer outputs using the ambiguity taxonomy introduced in Appendix C.4. Only a small portion of its predictions correspond to genuine ambiguity types (e.g., under-constrained or multi-granularity cases). A disproportionately large fraction stems from alias-level variation (35%) or unsupported guesses (21%). This confirms that the EM-Recall reward encourages broad coverage rather than true ambiguity resolution.

**Conclusion.** Taken together, these ablations demonstrate that each component of $A^2$SEARCH plays an indispensable role. The ambiguity-aware data enables exposure to multiple valid interpretations; the $\mathrm{AnsF1}$ reward encourages evidence-grounded reasoning toward all correct answers; and multi-answer generation allows the model to express these interpretations. None of the components alone is sufficient. The complete system is required to achieve robust improvements in both precision and recall, reflecting genuine advances in ambiguity resolution rather than output-format artifacts or metric gaming.

## F    THE USE OF LARGE LANGUAGE MODELS

In the preparation of this manuscript, LLMs were used solely as auxiliary tools for paraphrasing and polishing the writing to improve readability. No LLM was involved in formulating research ideas, proposing methods, designing experiments, or drawing conclusions. All scientific contributions and substantive content presented in this paper are the original work of the authors.

## G    PROMPT TEMPLATES

### G.1    PROMPT FOR LMJUDGE

---
**Prompt for LMJudge**

```
You will be given a question and its ground truth answer list where each item can be a ground truth answer.
↪   Provided a pred_answer, you need to judge if the pred_answer correctly answers the question based on the
↪   ground truth answer list. You should first give your rationale for the judgement, and then give your
↪   judgement result (i.e., correct or incorrect).

Here is the criteria for the judgement:
1. The pred_answer doesn't need to be exactly the same as any of the ground truth answers, but should be
↪   semantically same for the question.
2. Each item in the ground truth answer list can be viewed as a ground truth answer for the question, and the
↪   pred_answer should be semantically same to at least one of them.

question: {question}
ground truth answers: {gt_answer}
pred_answer: {pred_answer}

The output should in the following json format:
```json
{{
    "rationale": "your rationale for the judgement, as a text",
    "judgement": "your judgement result, can only be 'correct' or 'incorrect'"
}}
```

Your output:
```
---

### G.2    PROMPT FOR EVIDENCE-BASED VERIFICATION

---
**Prompt for Evidence-based Verification**

```
You are an Evidence-Consistency Judge.

[CRITICAL SCOPE]
- Do NOT assess the correctness of the Question; ambiguity is normal and expected.
- The Final Answer need not be comprehensive; concise is acceptable. Your job is only to judge whether it is a
↪   valid, defensible resolution supported by retrieved evidence.

[You will receive]
- Question
- The agent's rollout:
  - Thinking (ignore as evidence)
  - Search Queries
  - Tool Results (titles + snippets)
  - Final Answer

[Evaluation Principles]
1) Only Tool Results count as evidence. Ignore Thinking and outside knowledge.
2) For each atomic claim in the Final Answer, check support against evidence:
   - SUPPORTED: explicit match (paraphrase OK).
   - PARTIALLY_SUPPORTED: some support but with gaps/inference.
   - NOT_SUPPORTED: absent or contradicted.
   Concise answers choosing one reasonable reading are acceptable.
3) Be faithful: no details beyond evidence; numbers/dates must match.
4) If conflicting, prefer more specific/recent/relevant evidence; otherwise mark partial/not supported.
5) Closed-world: if insufficient, label PARTIALLY_SUPPORTED or NOT_SUPPORTED. Do not guess.
6) Scope: do not grade breadth, style, or completeness. Only check evidence support.

[Output Requirements]
- Output JSON only.
- Cite evidence with stable IDs (e.g., T1/T2).

[JSON Schema]
```
---

```
[JSON Schema]
{{
  "verdict": "SUPPORTED | PARTIALLY_SUPPORTED | NOT_SUPPORTED",
  "claims_analysis": [
    {{
      "claim": "atomic claim text",
      "status": "SUPPORTED | PARTIALLY_SUPPORTED | NOT_SUPPORTED",
      "evidence": ["title of the evidence", ...]
    }}
  ]
}}

[Verdict Labels]
- SUPPORTED: All claims clearly backed, no major gaps/conflicts.
- PARTIALLY_SUPPORTED: Some backing but with gaps/inference.
- NOT_SUPPORTED: Mostly unsupported or contradicted.

[Input Begin]
Question:
{question}

Rollout:
{rollout_full_text}
[Input End]
```

## G.3 PROMPT FOR GROUPING SEMANTICALLY IDENTICAL ANSWERS

**Prompt for Grouping Semantically Identical Answers**

```
You are provided with a list of textual answers. Your task is to organize these answers into groups of
↪  semantically equivalent or closely related responses.

### Requirements:
- Compare the **intended meaning** of each answer, rather than relying solely on surface wording.
- Answers that convey the **same or highly similar idea** should be placed in the same group, even if
↪  expressed differently.
- Answers with distinct meanings must remain in separate groups.
- Preserve the **original text** of each answer without modification.
- The output must follow the structure of a **JSON 2D array**, where each inner array contains one group of
↪  equivalent answers.

### Output Format:
```json
[
  ["Answer A1", "Answer A2", "Answer A3"],
  ["Answer B1", "Answer B2"],
  ...
]
```
```

## G.4 PROMPT FOR TRAINING $Sin$SEARCH

**Prompt for Training $Sin$Search**

```
You are a helpful assistant that solves the given question step by step using the wikipedia_search tool.

# Your Task
Use the wikipedia_search tool to gather comprehensive information and answer the user's question through a
↪  structured exploration process.

# Workflow
1. **Locate sources**
   - Use the wikipedia_search tool to find the most relevant Wikipedia pages related to the query.

2. **Branch out**
   - **Depth** - Explore each key page in detail to fully understand the topic.
   - **Breadth** - If there are multiple interpretations or entities, investigate each as a separate branch.

3. **Synthesize the answer**
   - Based on the evidence gathered, synthesize one well-supported answer.
   - If the question is ambiguous or has multiple valid interpretations, present and justify each possibility.

4. **Return in structured format**
   - Wrap your reasoning in `<think>` tags.
   - Always include at least one `<tool_call>`.
   - Present the final answer inside `<answer>` tags using the specified JSON structure below.
```

```
# Output Format
You must always follow this structure:

1. Start each step with your reasoning inside `<think> </think>` tags.
2. You must always make at least one wikipedia_search function call, even if you think you already know the
↪   answer.
    - Use `<tool_call> </tool_call>` tags to specify the function name and parameters, in the format shown
      ↪   below.
3. The user will provide the tool output inside `<tool_response> </tool_response>` tags. Never generate this
↪   output yourself.
4. Repeat the pattern of `<think>`, `<tool_call>` as needed to deepen or expand your search.
5. When ready to answer, present it inside:

<answer>
```json
{
    "rationale": "Concise reasoning: key search paths and evidence supporting the answer.",
    "answer": "your answer here"
}
```
</answer>

# Tools

You may call one or more functions to assist with the user query.

You are provided with function signatures within <tools></tools> XML tags:
<tools>
{
  "type": "function",
  "function": {
    "name": "wikipedia_search",
    "description": "Search Wikipedia for information about a specific query. Returns a list of summaries of
    ↪   the related articles.",
    "parameters": {
      "type": "object",
      "properties": {
        "query": {
          "type": "string",
          "description": "The specific query term to search on Wikipedia."
        }
      },
      "required": ["query"]
    }
  }
}
</tools>

For each function call, return a json object with function name and arguments within <tool_call></tool_call>
↪   XML tags:
<tool_call>
{"name": <function-name>, "arguments": <args-json-object>}
</tool_call>
```

## G.5  PROMPT FOR TRAINING A$^2$SEARCH AND $Abg$SEARCH

**Prompt for Training A$^2$SEARCH and $Abg$Search**

```
You are a helpful assistant that solves the given question step by step using the wikipedia_search tool.

# Your Task
Use the wikipedia_search tool to gather comprehensive information and answer the user's question through a
↪   structured exploration process.

# Workflow
1. **Locate sources**
    - Use the wikipedia_search tool to find the most relevant Wikipedia pages related to the query.

2. **Branch out**
    - **Depth** - Explore each key page in detail to fully understand the topic.
    - **Breadth** - If there are multiple interpretations or entities, investigate each as a separate branch.

3. **Extract answers**
    - Based on the evidence gathered, synthesize one or more well-supported answers (at most three different
    ↪   answers).
    - If the question is ambiguous or has multiple valid interpretations, present and justify each possibility.

4. **Return in structured format**
    - Wrap your reasoning in `<think>` tags.
```

```
    - Always include at least one `<tool_call>`.
    - Present the final answer inside `<answer>` tags using the specified JSON structure below.

# Output Format
You must always follow this structure:

1. Start each step with your reasoning inside `<think> </think>` tags.
2. You must always make at least one wikipedia_search function call, even if you think you already know the
↪  answer.
    - Use `<tool_call> </tool_call>` tags to specify the function name and parameters, in the format shown
    ↪  below.
3. The user will provide the tool output inside `<tool_response> </tool_response>` tags. Never generate this
↪  output yourself.
4. Repeat the pattern of `<think>`, `<tool_call>` as needed to deepen or expand your search.
5. When ready to answer, present it inside:

<answer>
```json
{
    "rationale": "Concise reasoning: key search paths and evidence supporting each answer.",
    "answers": [
        "Answer 1",
        "Answer 2",
        "Answer 3",
        ...
    ]
}
```
</answer>

# Tools

You may call one or more functions to assist with the user query.

You are provided with function signatures within <tools></tools> XML tags:
<tools>
{
  "type": "function",
  "function": {
    "name": "wikipedia_search",
    "description": "Search Wikipedia for information about a specific query. Returns a list of summaries of
    ↪  the related articles.",
    "parameters": {
      "type": "object",
      "properties": {
        "query": {
          "type": "string",
          "description": "The specific query term to search on Wikipedia."
        }
      },
      "required": ["query"]
    }
  }
}
</tools>

For each function call, return a json object with function name and arguments within <tool_call></tool_call>
↪  XML tags:
<tool_call>
{"name": <function-name>, "arguments": <args-json-object>}
</tool_call>
```

## G.6 Prompt for Training A²Search-Base

---

**Prompt for Training A²SEARCH-Base**

A conversation between User and Assistant. The user asks a question, and the assistant solves it step by step.
↪   The assistant first thinks about the reasoning process in the mind and then provides the user with the
↪   answer. During thinking, the assistant can invoke the wikipedia search tool to search for fact information
↪   about specific topics if needed. The reasoning process is enclosed within <think> </think> tags. When the
↪   assistant wants to search, the search query is enclosed in <search> </search> tags, and the user will
↪   provide the search result in <result> </result> tags. The assistant can repeat this pattern multiple times
↪   to explore different search paths or expand the reasoning. The assistant begins by locating relevant
↪   sources through the wikipedia search tool. Then, it explores each source in depth and also considers
↪   alternative interpretations in breadth. After gathering enough information, it extracts and organizes the
↪   possible answers. Finally, the assistant returns the answer in the required structured format. For
↪   example, <think> This is the reasoning process. </think> <search> search query here </search> <result>
↪   search result here </result> <think> This is the reasoning process. </think> <answer> The final answer is
↪   \[ \boxed{{answer1; answer2; answer3}} \] </answer>. In the last part of the answer, the final exact
↪   answer is enclosed within \boxed{{}} with latex format. If there are multiple possible answers, they
↪   should be separated by semicolons. User: {question}. Assistant:

---

## G.7 Prompt for Training $Sin$Search-Base

---

**Prompt for training $Sin$Search-Base**

A conversation between User and Assistant. The user asks a question, and the assistant solves it. The
↪   assistant first thinks about the reasoning process in the mind and then provides the user with the answer.
↪   During thinking, the assistant can invoke the wikipedia search tool to search for fact information about
↪   specific topics if needed. The reasoning process and answer are enclosed within <think> </think> and
↪   <answer> </answer> tags respectively, and the search query and result are enclosed within <search>
↪   </search> and <result> </result> tags respectively. For example, <think> This is the reasoning process.
↪   </think> <search> search query here </search> <result> search result here </result> <think> This is the
↪   reasoning process. </think> <answer> The final answer is \[ \boxed{{answer here}} \] </answer>. In the
↪   last part of the answer, the final exact answer is enclosed within \boxed{{}} with latex format. User:
↪   {question}. Assistant:

---

## G.8 Prompt for Ambiguity Type Classification

---

**Prompt for Ambiguity Type Classification**

You are a **QA Analyst Model**. You are given:

1.  A **Question**.
2.  Multiple **Search-Agent Trajectories** (queries, reasoning processes, and final answers).

**Premise:**
All provided final answers are considered **factually valid**. However, they are **substantively different**
↪   (they contain different strings, facts, numbers, or entities).

**Your Task:**
Determine the **specific root cause** of the divergence. Why did valid reasoning lead to different outcomes?
Select **one or more specific labels** from the list below and provide a concise explanation.

-----

# **Label Definitions & Decision Logic**

## **1. ALIAS_VARIANCE**

The divergence is superficial. The answers refer to the **exact same unique real-world entity** but use
↪   different names, abbreviations, spellings, or translation variations.

## **2. TIME_SENSITIVITY**

The divergence is caused because the answer depends on **when** the question is asked or the timeframe of the
↪   referenced documents. The question implies a "current" status or fails to lock a specific date.

## **3. EVIDENCE_CONFLICT**

The divergence is strictly caused by **contradictory information in the retrieved documents**. Different
↪   source passages provide conflicting facts about the same specific entity.

## **4. GRANULARITY_AMBIGUITY**

The divergence is caused by unspecified requirements regarding **precision of date/location, units of
↪   measurement, geographic scope, and others**. The agents agree on the fact but format it differently.

---

```
## **5. OPEN_ENDED**

The question is inherently qualitative or requests a list/example where **no single unique entity** is the
↪  exclusive truth. The question asks "Why", "How", or for "Benefits/Examples".

## **6. UNDER_CONSTRAINED**

The question appears to seek a specific unique entity but lacks sufficient conditions (constraints) to narrow
↪  the search down to one. This differs from "Open-Ended" because it usually feels like a "missing detail" or
↪  defect in the question, or involves homonyms (same name, different people/works).

## **7. MULTI_ITEM_RESPONSE**

The divergence is caused because the question's answer is inherently a list or set of items, not a single
↪  unique value. Even if there is only one obvious search path / interpretation of the question, there are
↪  multiple valid items that satisfy the criterion, and the question does not clearly require a complete or
↪  canonical list

-----

# **Output Format**

Return a JSON object:

```json
{
   "labels": [],
   "reason": "Concise explanation focusing on the specific differentiator."
}
```
```

# H  ROLLOUT CASES OF A$^2$SEARCH

| | |
|---|---|
| **Question** | **Who said that the most influential figure in Islamic philosophy was one of the greatest thinkers?** (from MuSiQue) |
| Cause of Ambiguity | Multiple historical figures could plausibly satisfy the given constraints. |
| Reference Answer | George Sarton (describes Avicenna as one of the greatest thinkers) |
| Alternative Answer | Oliver Leaman (describes Mulla Sadra as the most important thinker) |
| **Question** | **Who is the owner of the record label of the performer of What Kind of Love?** (from MuSiQue) |
| Cause of Ambiguity | The performer, Rodney Crowell, has released works under multiple record labels. |
| Reference Answer | Warner Music Group (parent of Warner Bros. Records) |
| Alternative Answer | Sony Music Entertainment (parent of Columbia Records) |
| **Question** | **Which country Prince Nikolaus Wilhelm Of Nassau's mother is from?** (from 2Wiki) |
| Cause of Ambiguity | The historical distinction between Württemberg and Germany is not explicitly considered. |
| Reference Answer | Germany (Württemberg later merged into Germany) |
| Alternative Answer | Württemberg (Princess Pauline of Württemberg was a member of the House of Württemberg) |
| **Question** | **Where was the place of death of Hayranidil Kadın's husband?** (from 2Wiki) |
| Cause of Ambiguity | Ambiguity arises from the granularity of the geographical specification. |
| Reference Answer | Constantinople (capital of the Ottoman Empire at the time) |
| Alternative Answer | Çırağan Palace (a palace located within Constantinople) |
| **Question** | **Where does the director of film Wine Of Morning work at?** (from HotpotQA) |
| Cause of Ambiguity | Ambiguity arises from different levels of institutional affiliation. |
| Reference Answer | Bob Jones University |
| Alternative Answer | Unusual Films (the university's production company) |
| **Question** | **How are Ceephax Acid Crew and Squarepusher's music similar?** (from HotpotQA) |
| Cause of Ambiguity | The question lacks a clearly defined dimension of comparison. |
| Reference Answer | Drum and bass electronic music |
| Alternative Answer | Acid house (another electronic subgenre frequently associated with their style) |
| **Question** | **What is the primary male hormone derived from?** (from Bamboogle) |
| Cause of Ambiguity | Testosterone can be traced either to its metabolic substrate or to its immediate biochemical precursor. |
| Reference Answer | Cholesterol (the fundamental metabolic source of steroid hormones) |
| Alternative Answer | Androstenedione (the direct biochemical precursor of testosterone) |

Table 19: Representative ambiguous questions drawn from multi-hop QA benchmarks. Each example includes the reference answer provided in the benchmark and the evidence-supported alternative answers identified by A$^2$SEARCH.

In this section, we present representative real examples of ambiguous questions together with the corresponding valid alternative answers identified by A$^2$SEARCH in Table 19. We additionally provide the complete reason-and-search trajectories produced by A$^2$SEARCH for the questions in Table 19. For clarity and to ensure correct LaTeX rendering, certain special Unicode characters have been omitted, which may result in minor character loss in the reproduced text.

***Who said that the most influential figure in Islamic philosophy was one of the greatest thinkers?*** *(from MuSiQue)*
***Reference Answer:*** *George Sarton*

```
<|im_start|>assistant
<think>To find who said that the most influential figure in Islamic philosophy was one of the greatest
↪  thinkers, first, we need to identify who the most influential figure in Islamic philosophy is
↪  considered to be, and then find who said that this figure was one of the greatest thinkers.</think>

<tool_call>
{"name": "wikipedia_search", "arguments": {"query": "who is the most influential figure in Islamic
↪  philosophy"}}
</tool_call><|im_end|>
<|im_start|>user
<tool_response>"Mulla Sadra"
Mulla Sadra adr ad-Dn Muammad Shrz, also called Mulla Sadr (; also spelled Molla Sadra, Mollasadra or
↪  Sadr-ol-Mote'allehin; ) (c. 1571/2  1640), was an Iranian Twelver Shi'a Islamic philosopher,
↪  theologian and lim who led the Iranian cultural renaissance in the 17th century. According to Oliver
↪  Leaman, Mulla Sadra is arguably the single most important and influential philosopher in the Muslim
↪  world in the last four hundred years. Though not its founder, he is considered the master of the
↪  Illuminationist (or, Ishraghi or Ishraqi) school of Philosophy, a seminal figure who synthesized the
↪  many tracts of the Islamic Golden Age

"Islamic philosophy"
after the golden ages] of Suhrawardi's ""Hikmat al-Ishraq"" (""Illumination Philosophy"") or Mulla
↪  Sadra's ""Hikmat-e-Mota'aliye"" (""Transcendent"" [or ""Exalted""] ""Philosophy""). Another figure is
↪  Allama Muhammad Iqbal, who reshaped and revitalized Islamic philosophy among the Muslims of the
↪  Indian sub-continent in the early 20th century. His ""The Reconstruction of Religious Thought in
↪  Islam"" is a milestone in the modern political philosophy of Islam. In contemporary Islamic regions,
↪  the teaching of ""hikmat"" or ""hikmah"" has continued to flourish. Philosophy has not been without
↪  criticism amongst Muslims, both contemporary and past. The imam Ahmad ibn Hanbal, for whom the
↪  Hanbali school of thought is named,

Al-Tafsir
Al-Tafsir Tafasir Al Quran (Persian:  ) are collections of books as commentaries on Quran written by
↪  Mulla Sadra. adr ad-Dn Muammad Shrz, also called Mulla Sadr (; also spelled Molla Sadra, Mollasadra
↪  or Sadr-ol-Mote'allehin; ) (c. 1571/2  1640), was an Iranian Shia Islamic philosopher, theologian and
↪  lim who led the Iranian cultural renaissance in the 17th century. According to Oliver Leaman, Mulla
↪  Sadra is arguably the single most important and influential philosopher in the Muslim world in the
↪  last four hundred years. Mulla Sadra brought ""a new philosophical insight in dealing with the nature
↪  of reality"" and created

"Islamic philosophy"
""reason"" (""'aql"") is sometimes recognised as a source of Islamic law, this may have a totally
↪  different meaning from ""reason"" in philosophy. The historiography of Islamic philosophy is marked
↪  by disputes as to how the subject should be properly interpreted. Some of the key issues involve the
↪  comparative importance of eastern intellectuals such as Ibn Sina (Avicenna) and of western thinkers
↪  such as Ibn Rushd, and also whether Islamic philosophy can be read at face value or should be
↪  interpreted in an esoteric fashion. Supporters of the latter thesis, like Leo Strauss, maintain that
↪  Islamic philosophers wrote so as to

"Mulla Sadra"
Mulla Sadra. In fact, the metalanguage approach could be considered in latter not the former. Mulla Sadra
↪  adr ad-Dn Muammad Shrz, also called Mulla Sadr (; also spelled Molla Sadra, Mollasadra or
↪  Sadr-ol-Mote'allehin; ) (c. 1571/2  1640), was an Iranian Twelver Shi'a Islamic philosopher,
↪  theologian and lim who led the Iranian cultural renaissance in the 17th century. According to Oliver
↪  Leaman, Mulla Sadra is arguably the single most important and influential philosopher in the Muslim
↪  world in the last four hundred years. Though not its founder, he is considered the master of the
↪  Illuminationist (or, Ishraghi or Ishraqi) school</tool_response><|im_end|>
<|im_start|>assistant
<think>From the provided responses, it is clear that the most influential figure in Islamic philosophy is
↪  considered to be Mulla Sadra or, alternatively, figures like Ibn Sina (Avicenna) or Ibn Rushd, but
↪  Oliver Leaman specifically mentions Mulla Sadra as the single most important and influential
↪  philosopher in the Muslim world in the last four hundred years. So, the most influential figure in
↪  Islamic philosophy is Mulla Sadra or Avicenna or others like them.

Next, we need to identify who said that this most influential figure was one of the greatest
↪  thinkers.</think>
<tool_call>
{"name": "wikipedia_search", "arguments": {"query": "who said that Mulla Sadra was one of the greatest
↪  thinkers"}}
</tool_call><|im_end|>
<|im_start|>user
<tool_response>"Mulla Sadra"
```

```
Mulla Sadra adr ad-Dn Muammad Shrz, also called Mulla Sadr (; also spelled Molla Sadra, Mollasadra or
↪   Sadr-ol-Mote'allehin; ) (c. 1571/2  1640), was an Iranian Twelver Shi'a Islamic philosopher,
↪   theologian and lim who led the Iranian cultural renaissance in the 17th century. According to Oliver
↪   Leaman, Mulla Sadra is arguably the single most important and influential philosopher in the Muslim
↪   world in the last four hundred years. Though not its founder, he is considered the master of the
↪   Illuminationist (or, Ishraghi or Ishraqi) school of Philosophy, a seminal figure who synthesized the
↪   many tracts of the Islamic Golden Age

"Mulla Sadra"
Mulla Sadra. In fact, the metalanguage approach could be considered in latter not the former. Mulla Sadra
↪   adr ad-Dn Muammad Shrz, also called Mulla Sadr (; also spelled Molla Sadra, Mollasadra or
↪   Sadr-ol-Mote'allehin; ) (c. 1571/2  1640), was an Iranian Twelver Shi'a Islamic philosopher,
↪   theologian and lim who led the Iranian cultural renaissance in the 17th century. According to Oliver
↪   Leaman, Mulla Sadra is arguably the single most important and influential philosopher in the Muslim
↪   world in the last four hundred years. Though not its founder, he is considered the master of the
↪   Illuminationist (or, Ishraghi or Ishraqi) school

Al-Tafsir
Al-Tafsir Tafasir Al Quran (Persian:  ) are collections of books as commentaries on Quran written by
↪   Mulla Sadra. adr ad-Dn Muammad Shrz, also called Mulla Sadr (; also spelled Molla Sadra, Mollasadra
↪   or Sadr-ol-Mote'allehin; ) (c. 1571/2  1640), was an Iranian Shia Islamic philosopher, theologian and
↪   lim who led the Iranian cultural renaissance in the 17th century. According to Oliver Leaman, Mulla
↪   Sadra is arguably the single most important and influential philosopher in the Muslim world in the
↪   last four hundred years. Mulla Sadra brought ""a new philosophical insight in dealing with the nature
↪   of reality"" and created

"Mulla Sadra"
of the Ash'ari school and Twelvers. His main work is ""The Transcendent Theosophy in the Four Journeys of
↪   the Intellect"", or simply ""Four Journeys"". Born in Shiraz, in what is now Iran, to a notable
↪   family of court officials in 1571 or 1572, In Mulla Sadra's time, the Safavid dynasty governed on
↪   Iran. Safavid kings granted independence to Fars Province which was ruled by the king's brother.
↪   Mulla Sadara's father, khwajah Ibrahim Qavami, was a knowledgeable and extremely faithful politician.
↪   His father was a rich man and held a high position, but had no children. However, after a lot of

"Mir Fendereski"
Mir Fendereski Mir Fendereski or Mir Findiriski (Persian: ) (15621640) was a Persian philosopher, poet
↪   and mystic of the Safavid era. His full name is given as Sayyed Mir Abulqasim Astarabadi (Persian:
↪   ), and he is famously known as Fendereski. He lived for a while in Isfahan at the same time as Mir
↪   Damad spent a great part of his life in India among yogis and Zoroastrians, and learnt certain things
↪   from them. He was patronized by both the Safavid and Mughal courts. The famous Persian philosopher
↪   Mulla Sadra also studied under him. Mir Fendereski remains a mysterious</tool_response><|im_end|>
<|im_start|>assistant
<tool_call>
{"name": "wikipedia_search", "arguments": {"query": "who said that Avicenna was one of the greatest
↪   thinkers"}}
</tool_call><|im_end|>
<|im_start|>user
<tool_response>Avicenna
Avicenna Avicenna (; also Ibn Sn or Abu Ali Sina; ;  June 1037) was a Persian polymath who is regarded as
↪   one of the most significant physicians, astronomers, thinkers and writers of the Islamic Golden Age.
↪   He has been described as the father of early modern medicine. Of the 450 works he is known to have
↪   written, around 240 have survived, including 150 on philosophy and 40 on medicine. His most famous
↪   works are ""The Book of Healing"", a philosophical and scientific encyclopedia, and ""The Canon of
↪   Medicine"", a medical encyclopedia which became a standard medical text at many

Avicenna
non-Christian thinkers in his ""Divine Comedy"" such as Virgil, Averroes, Homer, Horace, Ovid, Lucan,
↪   Socrates, Plato, and Saladin, Avicenna has been recognized by both East and West, as one of the great
↪   figures in intellectual history. George Sarton, the author of ""The History of Science"", described
↪   Ibn Sn as ""one of the greatest thinkers and medical scholars in history"" and called him ""the most
↪   famous scientist of Islam and one of the most famous of all races, places, and times."" He was one of
↪   the Islamic world's leading writers in the field of medicine. Along with Rhazes, Abulcasis, Ibn
↪   al-Nafis,

Avicenna
has been translated into English by Parwiz Morewedge in 1977. The book is also important in respect to
↪   Persian scientific works. ""Andar Danesh-e Rag"" (, ""On the Science of the Pulse"") contains nine
↪   chapters on the science of the pulse and is a condensed synopsis. Persian poetry from Ibn Sina is
↪   recorded in various manuscripts and later anthologies such as ""Nozhat al-Majales"". Avicenna
↪   Avicenna (; also Ibn Sn or Abu Ali Sina; ;  June 1037) was a Persian polymath who is regarded as one
↪   of the most significant physicians, astronomers, thinkers and writers of the Islamic Golden Age. He

Avicenna
```

```
in which he comments on several Quranic verses and holds the Quran in high esteem. Avicenna argued that
↪   the Islamic prophets should be considered higher than philosophers. While he was imprisoned in the
↪   castle of Fardajan near Hamadhan, Avicenna wrote his famous ""Floating Man""  literally falling man
↪   thought experiment to demonstrate human self-awareness and the substantiality and immateriality of
↪   the soul. Avicenna believed his ""Floating Man"" thought experiment demonstrated that the soul is a
↪   substance, and claimed humans cannot doubt their own consciousness, even in a situation that prevents
↪   all sensory data input. The thought experiment told its

Avicenna
a ""Latin Avicennism"" as flourishing for some time, paralleling the more influential Latin Averroism,
↪   but suppressed by the Parisian decrees of 1210 and 1215. Avicenna's psychology and theory of
↪   knowledge influenced William of Auvergne, Bishop of Paris and Albertus Magnus, while his metaphysics
↪   influenced the thought of Thomas Aquinas. Early Islamic philosophy and Islamic metaphysics, imbued as
↪   it is with Islamic theology, distinguishes more clearly than Aristotelianism between essence and
↪   existence. Whereas existence is the domain of the contingent and the accidental, essence endures
↪   within a being beyond the accidental. The philosophy of Ibn Sn, particularly that part relating
↪   to</tool_response><|im_end|>
<|im_start|>assistant
<tool_call>
{"name": "wikipedia_search", "arguments": {"query": "who said that Ibn Sina was one of the greatest
↪   thinkers in history"}}
</tool_call><|im_end|>
<|im_start|>user
<tool_response>Avicenna
non-Christian thinkers in his ""Divine Comedy"" such as Virgil, Averroes, Homer, Horace, Ovid, Lucan,
↪   Socrates, Plato, and Saladin, Avicenna has been recognized by both East and West, as one of the great
↪   figures in intellectual history. George Sarton, the author of ""The History of Science"", described
↪   Ibn Sn as ""one of the greatest thinkers and medical scholars in history"" and called him ""the most
↪   famous scientist of Islam and one of the most famous of all races, places, and times."" He was one of
↪   the Islamic world's leading writers in the field of medicine. Along with Rhazes, Abulcasis, Ibn
↪   al-Nafis,

Avicenna
Avicenna Avicenna (; also Ibn Sn or Abu Ali Sina; ;  June 1037) was a Persian polymath who is regarded as
↪   one of the most significant physicians, astronomers, thinkers and writers of the Islamic Golden Age.
↪   He has been described as the father of early modern medicine. Of the 450 works he is known to have
↪   written, around 240 have survived, including 150 on philosophy and 40 on medicine. His most famous
↪   works are ""The Book of Healing"", a philosophical and scientific encyclopedia, and ""The Canon of
↪   Medicine"", a medical encyclopedia which became a standard medical text at many

"History of science"
Sina (Avicenna, c. 9801037) is regarded as the most influential philosopher of Islam. He pioneered the
↪   science of experimental medicine and was the first physician to conduct clinical trials. His two most
↪   notable works in medicine are the ""Kitb al-shif"" (""Book of Healing"") and The Canon of Medicine,
↪   both of which were used as standard medicinal texts in both the Muslim world and in Europe well into
↪   the 17th century. Amongst his many contributions are the discovery of the contagious nature of
↪   infectious diseases, and the introduction of clinical pharmacology. Scientists from the Islamic world
↪   include al-Farabi (polymath), Abu al-Qasim

Avicenna
has been translated into English by Parwiz Morewedge in 1977. The book is also important in respect to
↪   Persian scientific works. ""Andar Danesh-e Rag"" (, ""On the Science of the Pulse"") contains nine
↪   chapters on the science of the pulse and is a condensed synopsis. Persian poetry from Ibn Sina is
↪   recorded in various manuscripts and later anthologies such as ""Nozhat al-Majales"". Avicenna
↪   Avicenna (; also Ibn Sn or Abu Ali Sina; ;  June 1037) was a Persian polymath who is regarded as one
↪   of the most significant physicians, astronomers, thinkers and writers of the Islamic Golden Age. He

Avicenna
however, stating that: """"I prefer a short life with width to a narrow one with length"""". On his
↪   deathbed remorse seized him; he bestowed his goods on the poor, restored unjust gains, freed his
↪   slaves, and read through the Quran every three days until his death. He died in June 1037, in his
↪   fifty-eighth year, in the month of Ramadan and was buried in Hamadan, Iran. Ibn Sn wrote extensively
↪   on early Islamic philosophy, especially the subjects logic, ethics, and metaphysics, including
↪   treatises named ""Logic"" and ""Metaphysics"". Most of his works were written in Arabic  then the
↪   language of</tool_response><|im_end|>
<|im_start|>assistant
<tool_call>
{"name": "wikipedia_search", "arguments": {"query": "who said that the most influential figure in Islamic
↪   philosophy was one of the greatest thinkers in history"}}
</tool_call><|im_end|>
<|im_start|>user
<tool_response>"Mulla Sadra"
Mulla Sadra adr ad-Dn Muammad Shrz, also called Mulla Sadr (; also spelled Molla Sadra, Mollasadra or
↪   Sadr-ol-Mote'allehin; ) (c. 1571/2  1640), was an Iranian Twelver Shi'a Islamic philosopher,
↪   theologian and lim who led the Iranian cultural renaissance in the 17th century. According to Oliver
↪   Leaman, Mulla Sadra is arguably the single most important and influential philosopher in the Muslim
↪   world in the last four hundred years. Though not its founder, he is considered the master of the
↪   Illuminationist (or, Ishraghi or Ishraqi) school of Philosophy, a seminal figure who synthesized the
↪   many tracts of the Islamic Golden Age
```

```
Al-Kindi
and Roger Bacon. The Italian Renaissance scholar Geralomo Cardano (15011575) considered him as one of the
↪  twelve greatest minds. Al-Kindi Abu Ysuf Yaqb ibn Isq a-abb al-Kind (; ; ; c. 801873 AD) was an Arab
↪  Muslim philosopher, polymath, mathematician, physician and musician. Al-Kindi was the first of the
↪  Muslim peripatetic philosophers, and is unanimously hailed as the ""father of Arab philosophy"" for
↪  his synthesis, adaptation and promotion of Greek and Hellenistic philosophy in the Muslim world.
↪  Al-Kindi was born in Kufa and educated in Baghdad. He became a prominent figure in the House of
↪  Wisdom, and a number of

Al-Tafsir
Al-Tafsir Tafasir Al Quran (Persian:  ) are collections of books as commentaries on Quran written by
↪  Mulla Sadra. adr ad-Dn Muammad Shrz, also called Mulla Sadr (; also spelled Molla Sadra, Mollasadra
↪  or Sadr-ol-Mote'allehin; ) (c. 1571/2  1640), was an Iranian Shia Islamic philosopher, theologian and
↪  lim who led the Iranian cultural renaissance in the 17th century. According to Oliver Leaman, Mulla
↪  Sadra is arguably the single most important and influential philosopher in the Muslim world in the
↪  last four hundred years. Mulla Sadra brought ""a new philosophical insight in dealing with the nature
↪  of reality"" and created

"Mulla Sadra"
Mulla Sadra. In fact, the metalanguage approach could be considered in latter not the former. Mulla Sadra
↪  adr ad-Dn Muammad Shrz, also called Mulla Sadr (; also spelled Molla Sadra, Mollasadra or
↪  Sadr-ol-Mote'allehin; ) (c. 1571/2  1640), was an Iranian Twelver Shi'a Islamic philosopher,
↪  theologian and lim who led the Iranian cultural renaissance in the 17th century. According to Oliver
↪  Leaman, Mulla Sadra is arguably the single most important and influential philosopher in the Muslim
↪  world in the last four hundred years. Though not its founder, he is considered the master of the
↪  Illuminationist (or, Ishraghi or Ishraqi) school

"Islamic philosophy"
the 9th century and ended with Averroes (Ibn Rushd) at the end of 12th century. The death of Averroes
↪  effectively marked the end of a particular discipline of Islamic philosophy usually called the
↪  Peripatetic Arabic School, and philosophical activity declined significantly in Western Islamic
↪  countries such as Islamic Iberia and North Africa. Islamic philosophy persisted for much longer in
↪  Muslim Eastern countries, in particular Safavid Persia, Ottoman and Mughal Empires, where several
↪  schools of philosophy continued to flourish: Avicennism, Averroism, Illuminationist philosophy,
↪  Mystical philosophy, Transcendent theosophy, and Isfahan philosophy. Ibn Khaldun, in his
↪  ""Muqaddimah"", made important contributions to the philosophy</tool_response><|im_end|>
<|im_start|>assistant
<think>From the provided responses, it is clear that figures like George Sarton, Roger Bacon, and others
↪  have referred to Avicenna and other philosophers as one of the greatest thinkers.

- Oliver Leaman described Mulla Sadra as "arguably the single most important and influential philosopher
↪  in the Muslim world in the last four hundred years."
- George Sarton described Avicenna as "one of the greatest thinkers and medical scholars in history."

So, it appears that scholars like George Sarton and others have referred to Islamic philosophers as one
↪  of the greatest thinkers.<think>

<answer>
```json
{
  "rationale": "Mulla Sadra is considered the most influential figure in Islamic philosophy, and George
  ↪  Sarton described Avicenna as one of the greatest thinkers in history."
,
  "answers": [
    "Oliver Leaman",
    "George Sarton"
  ]
}
</answer><|im_end|>
```

---

***Who is the owner of the record label of the performer of What Kind of Love?*** *(from MuSiQue)*

**Reference Answer:** *Warner Music*

```
<|im_start|>assistant
<think>To find the owner of the record label of the performer of "What Kind of Love," first, we need to
↪  identify the performer of the song "What Kind of Love." </think>

<tool_call>
{"name": "wikipedia_search", "arguments": {"query": "who performed the song What Kind of Love"}}
</tool_call><|im_end|>
<|im_start|>user
<tool_response>"What Kind of Love"
```

```
What Kind of Love ""What Kind of Love"" is a song co-written and recorded by American country music artist
↪   Rodney Crowell. It was released in June 1992 as the second single from the album ""Life Is Messy"".
↪   The song reached number 11 on the U.S. ""Billboard"" Hot Country Singles & Tracks chart and peaked at
↪   number 2 on the ""RPM"" Country Tracks in Canada. Crowell and co-writer Will Jennings wrote the
↪   lyrics to a recording of a melody that Roy Orbison had made before his death. The song features
↪   backing vocals from Don Henley and Linda Ronstadt. Deborah Evans Price,

"What Kind of Love"
covered the song on his 2002 album for DreamWorks Records. It was the B-side to the album's first single,
↪   ""Honey Do"". What Kind of Love ""What Kind of Love"" is a song co-written and recorded by American
↪   country music artist Rodney Crowell. It was released in June 1992 as the second single from the album
↪   ""Life Is Messy"". The song reached number 11 on the U.S. ""Billboard"" Hot Country Singles & Tracks
↪   chart and peaked at number 2 on the ""RPM"" Country Tracks in Canada. Crowell and co-writer Will
↪   Jennings wrote the lyrics to a recording of a melody that

"What Kind of Love Are You On"
What Kind of Love Are You On ""What Kind of Love Are You On"" is a song by American hard rock band
↪   Aerosmith. The song, originally a track left off the ""Nine Lives"" album, was included on """" for
↪   the 1998 film ""Armageddon"" starring lead singer Steven Tyler's daughter Liv Tyler. The song, was
↪   released as a promotional single to rock radio, reaching #4 on the Mainstream Rock Tracks chart. It
↪   was written by Steven Tyler, guitarist Joe Perry and outside songwriters Jack Blades and Tommy Shaw
↪   (both formerly of Damn Yankees). It is the second song written for the

"What Kind of Love"
of ""Billboard"" magazine reviewed the song favorably, calling it the ""first step in a major effort to
↪   immerse this rock-influenced country star into the pop and AC realm. She goes on to say that ""twangy
↪   instrumentation a la Tom Petty and the late Roy Orbison (with whom this song was penned), combined
↪   with an appealing vocal, makes a strong argument for play at these and album rock formats."" From
↪   ""Life Is Messy"" liner notes. ""What Kind of Love"" debuted at number 75 on the U.S. ""Billboard""
↪   Hot Country Singles & Tracks for the week of June 27, 1992. Mike Walker

"What Kinda Love"
What Kinda Love ""What Kinda Love"" is a song recorded by Canadian country rock artist Dallas Smith. It
↪   was released in January 2013 as the fourth single from his debut solo album, ""Jumped Right In"". It
↪   peaked at number 77 on the Canadian Hot 100 in April 2013. Casadie Pederson of ""Top Country"" called
↪   the song ""perhaps his best release yet."" She wrote that ""it can relate to so many people as we go
↪   along this journey of life and love so many things, all of which mean different things to us."" The
↪   music video was directed by Stephano Barberis</tool_response><|im_end|>
<|im_start|>assistant
<think>From the provided responses, it is clear that the performer of the song "What Kind of Love" is
↪   Rodney Crowell.

Next, we need identify the record label of the performer Rodney Crowell.</think>

<tool_call>
{"name": "wikipedia_search", "arguments": {"query": "record label of Rodney Crowell"}}
</tool_call><|im_end|>
<|im_start|>user
<tool_response>"Rodney Crowell"
music career and recorded what was to be a new album for Warner Bros., ""Street Language"". That album, a
↪   pop-sounding effort coproduced by David Malloy, was rejected by Warner Bros, and was rejected by the
↪   label and never released. Warner Bros. requested a more Nashville-friendly record, but Crowell
↪   negotiated a release from his contract and moved to Columbia Records. After producing Rosanne Cash's
↪   ""Rhythm & Romance"", Crowell signed to Columbia Records in 1986. His first album for that label was
↪   reworked ""Street Language"", co-produced with Booker T. Jones and featuring a blend of soul and
↪   country music. The album did

"The Rodney Crowell Collection"
The Rodney Crowell Collection The Rodney Crowell Collection is the title of the first compilation album
↪   by American country music artist Rodney Crowell. It was released in 1989 (see 1989 in country music)
↪   by Crowell's former label, Warner Bros. Records, following the huge success of his album ""Diamonds &
↪   Dirt"". It features selections from his first three albums that were released under the Warner Bros.
↪   label between 1978 and 1981. It charted #65 on the Top Country Albums chart. The album is the first
↪   release of Crowell singing ""I Don't Have to Crawl"" The Crowell-penned tune was recorded by Emmylou

"Rodney Crowell"
charts. In 2013, the album won the Americana Music Awards' Album of the Year award and Crowell and Harris
↪   were named group/duo of the year. On January 26, 2014, Crowell won his second Grammy Award when ""Old
↪   Yellow Moon"" won the Grammy for Best Americana Album. On 11 May 2015, Crowell and Emmylou Harris
↪   released ""The Traveling Kind"" on Nonesuch Records. Crowell released his first album on New West
↪   Records, ""Tarpaper Sky"", on April 15, 2014. Crowell co-produced the record with his long-time
↪   collaborator, Steuart Smith. In the fall of 2014, Crowell was hired as music director for the Hank

"Rodney Crowell (album)"
```

```
Rodney Crowell (album) Rodney Crowell is the third, eponymous studio album by American country music
↪  artist Rodney Crowell. It was released in 1981 (see 1981 in country music) by Warner Bros. Records
↪  and was his last album on that label before switching to Columbia. It was the first album Crowell
↪  produced by himself. It reached #47 on the Top Country Albums chart and #105 on the ""Billboard"" 200
↪  albums chart. The songs, ""Stars on the Water"" and ""Victim or a Fool"" were released as singles.
↪  ""Stars on the Water"" reached #30 on the Hot Country Songs chart, his highest charting

"Duke Records"
Duke Records Duke Records was an American record label, started in Memphis, Tennessee, in 1952 by David
↪  James Mattis (WDIA program director and DJ) and Bill Fitzgerald, owners of Tri-State Recording
↪  Company. Their first release was Roscoe Gordon singing ""Hey Fat Girl"", issued on Duke R-1, later
↪  amended to R-101. After forming a partnership with Mattis in the summer of 1952, Don Robey (founder
↪  of Houston's Peacock Records) took control of Duke. Both labels then headquartered at his Bronze
↪  Peacock club at 2809 Erastus Street in Houston, focusing on R&B and gospel music. Robey started a
↪  subsidiary, Back Beat Records,</tool_response><|im_end|>
<|im_start|>assistant
<think>From the provided responses, it is clear that Rodney Crowell has worked with labels like Warner
↪  Bros. and Columbia Records.

Now, we need to identify the record label of Rodney Crowell and then find the owner of that record
↪  label.</think>

<tool_call>
{"name": "wikipedia_search", "arguments": {"query": "owner of Warner Bros Records"}}
</tool_call><|im_end|>
<|im_start|>user
<tool_response>"Warner Bros. Records"
Warner Bros. Records Warner Bros. Records Inc. is an American major record label owned by Warner Music
↪  Group and headquartered in Burbank, California. It was founded in 1958 as the recorded music division
↪  of the American film studio Warner Bros., and was one of a group of labels owned and operated by
↪  larger parent corporations for much of its existence. The sequence of companies that controlled Warner
↪  Bros. and its allied labels evolved through a convoluted series of corporate mergers and acquisitions
↪  from the early 1960s to the early 2000s. Over this period, Warner Bros. Records grew from a struggling

"Warner Bros. Records"
Records from EMI as part of its sale to Universal Music Group. Most Parlophone artists (excluding
↪  Coldplay and Tinie Tempah, who were placed under Atlantic) were placed under Warner Bros. Records for
↪  US distribution. In December 2013, Dan McCarroll joined the company as president. Warner Bros.
↪  Records Warner Bros. Records Inc. is an American major record label owned by Warner Music Group and
↪  headquartered in Burbank, California. It was founded in 1958 as the recorded music division of the
↪  American film studio Warner Bros., and was one of a group of labels owned and operated by larger for
↪  parent corporations

"Warner Music Group"
multibillion-dollar annual turnover, WMG employs more than 3,500 people and has operations in more than
↪  50 countries throughout the world. The company owns and operates some of the largest and most
↪  successful labels in the world, including Elektra Records, Warner Bros. Records, Parlophone, and
↪  Atlantic Records. WMG also owns Warner/Chappell Music, one of the world's largest music publishers.
↪  Since August 2, 2018, WMG has expanded its business to digital media operation through its
↪  acquisition of Uproxx Media Group. The film company Warner Bros. had no record label division at the
↪  time one of its contracted actors, Tab Hunter, scored a

"Warner Bros. Records"
collections from music users. In 1969, Warner-Seven Arts was taken over by the Kinney National Company,
↪  headed by New York businessman Steve J. Ross, who would successfully lead the Warner group of
↪  companies until his death in 1992. The US\$400 million deal created a new conglomerate that combined
↪  the Warner film, recording and music publishing divisions with Kinney's multi-faceted holdings. Ross
↪  had started the company in the late 1950s while working in his family's funeral businessseeing the
↪  opportunity to use the company's cars, which were idle at night, he founded a successful hire car
↪  operation, which he later merged with

"Warner Bros. Records"
minor player in the music industry to one of the top record labels in the world. In 2004, these music
↪  assets were divested by their then owner Time Warner and purchased by a private equity group. This
↪  independent company traded as the Warner Music Group and was the world's last publicly traded major
↪  music company before being bought and privatized by Access Industries in 2011. WMG is the smallest of
↪  the three major international music conglomerates, and Max Lousada oversees recorded music operations
↪  of the company. Artists signed to Warner Bros. Records have included Prince, Cher, Kylie Minogue,
↪  Kimbra, Goo</tool_response><|im_end|>
<|im_start|>assistant
<think>From the provided responses, it is clear that Warner Bros. Records is owned by Warner Music Group.

Next, we need to identify the owner of the record label Columbia Records.</think>

<tool_call>
{"name": "wikipedia_search", "arguments": {"query": "owner of Columbia Records"}}
</tool_call><|im_end|>
<|im_start|>user
<tool_response>"Columbia Records"
```

```
Williams, Harry Styles, BTS, Little Mix and John Cena. The Columbia Phonograph Company was founded in
↪  1887 by stenographer, lawyer and New Jersey native Edward D. Easton (18561915) and a group of
↪  investors. It derived its name from the District of Columbia, where it was headquartered. At first it
↪  had a local monopoly on sales and service of Edison phonographs and phonograph cylinders in
↪  Washington, D.C., Maryland, and Delaware. As was the custom of some of the regional phonograph
↪  companies, Columbia produced many commercial cylinder recordings of its own, and its catalogue of
↪  musical records in 1891 was 10 pages.

"Columbia Records"
Columbia Records Columbia Records is an American record label owned by Sony Music Entertainment, a
↪  subsidiary of Sony Corporation of America, the North American division of Japanese conglomerate Sony.
↪  It was founded in 1887, evolving from the American Graphophone Company, the successor to the Volta
↪  Graphophone Company. Columbia is the oldest surviving brand name in the recorded sound business, and
↪  the second major company to produce records. From 1961 to 1990, Columbia recordings were released
↪  outside North America under the name CBS Records to avoid confusion with EMI's Columbia Graphophone
↪  Company. Columbia is one of Sony Music's four flagship record

"Columbia Records"
with its Epic Records sister to form the Columbia/Epic Label Group under which it operated as an imprint.
↪  In July 2011, as part of further corporate restructuring, Epic was split from the Columbia/Epic Group
↪  as Epic took in multiple artists from Jive Records. As of March 2013, Columbia Records is home to 90
↪  artists such as Lauren Jauregui, Robbie Williams, Calvin Harris, and Daft Punk. On January 2, 2018,
↪  Ron Perry was named as the CEO of Columbia Records. Sony and Columbia had cooperated earlier. The SQ
↪  Stereo Quadraphonic was developed by engineer Benjamin Bauer of Columbia in cooperation with

"Columbia Records"
four television studios. Columbia Records Columbia Records is an American record label owned by Sony
↪  Music Entertainment, a subsidiary of Sony Corporation of America, the North American division of
↪  Japanese conglomerate Sony. It was founded in 1887, evolving from the American Graphophone Company,
↪  the successor to the Volta Graphophone Company. Columbia is the oldest surviving brand name in the
↪  recorded sound business, and the second major company to produce records. From 1961 to 1990, Columbia
↪  recordings were released outside North America under the name CBS Records to avoid confusion with
↪  EMI's Columbia Graphophone Company. Columbia is one of Sony Music's

"Craig Kallman"
Craig Kallman Craig Kallman is an American businessman and former DJ. He is currently the Chairman and
↪  Chief Executive Officer of Atlantic Records. His music career began in the 1980s as a DJ in
↪  Manhattan. He is the founder of Big Beat Records, an independent label he founded in 1987. The label
↪  was purchased by Atlantic Records, where he has worked since 1991 after the acquisition, becoming its
↪  CEO in 2005. Kallman is considered one of the world's leading investors in rare records. He has a
↪  collection of approximately 750,000 records, 150,000 items of music memorabilia and 100,000 compact
↪  discs.</tool_response><|im_end|>
<|im_start|>assistant
<think>From the provided responses, it is clear that Columbia Records is owned by Sony Music
↪  Entertainment.

So, to summarize:

- The performer of "What Kind of Love" is Rodney Crowell.
- Rodney Crowell has worked with labels like Warner Bros. and Columbia Records.
- The record label of Rodney Crowell includes Warner Bros. and Columbia Records.
- The owner of Warner Bros. Records is Warner Music Group.
- The owner of Columbia Records is Sony Music Entertainment.</think>

<answer>
```json
{
  "rationale": "Rodney Crowell performed the song 'What Kind of Love,' and his record label was Warner
  ↪  Bros. and Columbia Records, which are owned by Warner Music Group and Sony Music Entertainment,
  ↪  respectively."
,
  "answers": [
    "Warner Music Group",
    "Sony Music Entertainment"
  ]
}
</answer><|im_end|>
```

---

**Which country Prince Nikolaus Wilhelm Of Nassau's mother is from?** *(from 2Wiki)*
**Reference Answer: *Germany***

```
<|im_start|>assistant
<think>To find the mother's nationality of Prince Nikolaus Wilhelm Of Nassau, first, we need to identify
↪  who Prince Nikolaus Wilhelm Of Nassau is and then find information about his mother. </think>

<tool_call>
{"name": "wikipedia_search", "arguments": {"query": "who was Prince Nikolaus Wilhelm Of Nassau"}}
```

```
</tool_call><|im_end|>
<|im_start|>user
<tool_response>"Prince Nikolaus Wilhelm of Nassau"
Prince Nikolaus Wilhelm of Nassau Prince Nikolaus Wilhelm of Nassau (20 September 1832  17 September
↪   1905), was the only son of William, Duke of Nassau by his second wife Princess Pauline of Wrttemberg.
↪   He married morganatically in London on 1 July 1868 Natalya Alexandrovna Pushkina (Saint Petersburg, 4
↪   June 1836  Cannes, 23 March 1913). She was the daughter of Alexander Sergeevich Pushkin and wife
↪   Natalya Nikolaevna Goncharova, and a descendant of Abram Petrovich Gannibal and Petro Doroshenko,
↪   Hetman of Ukrainian Cossacks, in turn grandson of Mykhailo Doroshenko. She was divorced from Russian
↪   General Mikhail Leontievich von Dubelt, by

"William, Duke of Nassau"
William, Duke of Nassau Wilhelm (Given names: ""Georg Wilhelm August Heinrich Belgicus""; 14 June 1792,
↪   Kirchheimbolanden  20/30 August 1839, Bad Kissingen) was joint sovereign Duke of Nassau, along with
↪   his cousin Frederick Augustus, reigning from 1816 until 1839. He was also sovereign Prince of
↪   Nassau-Weilburg from 1816 until its incorporation into the duchy of Nassau. Frederick Augustus died
↪   on 24 March 1816 and Wilhelm inherited the Usingen territories and became sole sovereign of the Duchy
↪   of Nassau. He is the father of Adolphe, Grand Duke of Luxembourg, and Queen Sophia of Sweden and
↪   Norway, consort of King Oscar II

"Prince Nikolaus Wilhelm of Nassau"
whom she had a daughter. In 1868, George Victor, Prince of Waldeck and Pyrmont created her Countess von
↪   Merenberg. They had three children: Prince Nikolaus Wilhelm of Nassau Prince Nikolaus Wilhelm of
↪   Nassau (20 September 1832  17 September 1905), was the only son of William, Duke of Nassau by his
↪   second wife Princess Pauline of Wrttemberg. He married morganatically in London on 1 July 1868
↪   Natalya Alexandrovna Pushkina (Saint Petersburg, 4 June 1836  Cannes, 23 March 1913). She was the
↪   daughter of Alexander Sergeevich Pushkin and wife Natalya Nikolaevna Goncharova, and a descendant of
↪   Abram Petrovich Gannibal and

"Frederick Augustus, Duke of Nassau"
Frederick Augustus, Duke of Nassau Friedrich August, Duke of Nassau, Prince of Nassau-Usingen (23 April
↪   1738 in Usingen  24 March 1816 in Wiesbaden) was the last Prince of Nassau-Usingen and, jointly with
↪   his cousin, Friedrich Wilhelm of Nassau-Weilburg, first Duke of Nassau. He died without surviving
↪   male issue and was succeeded by his cousin's son, Wilhelm. Friedrich August, born in Usingen, was a
↪   younger son of Prince Charles of Nassau-Usingen and Princess Christiane Wilhelmine of Saxe-Eisenach
↪   (daughter of John William III, Duke of Saxe-Eisenach). From 1744, he lived with his parents in
↪   Schloss Biebrich in Wiesbaden. He followed a

"William, Duke of Nassau"
and also a 17th cousin of William III of the Netherlands. Wilhelm was the eldest son of Frederick William,
↪   Duke of Nassau, and his wife, Burgravine Louise Isabelle of Kirchberg. With the Nassau troops, he was
↪   involved on the Seventh Coalition's side in the Battle of Waterloo against Napoleon. On 9 January
↪   1816, he succeeded his father, Duke Frederick William, as the Prince of Nassau-Weilburg and joint
↪   Duke of Nassau with his cousin, Frederick Augustus, of the Nassau-Usingen branch of his family. When
↪   his cousin and co-Duke died on 24 March 1816, Wilhelm inherited the Usingen territories and became
↪   sole</tool_response><|im_end|>
<|im_start|>assistant
<think>From the provided responses, it is clear that Prince Nikolaus Wilhelm of Nassau was the son of
↪   William, Duke of Nassau, and his second wife, Princess Pauline of Wrttemberg.

Next, we need to identify the nationality of Prince Nikolaus Wilhelm of Nassau's mother, which is
↪   Princess Pauline of Wrttemberg.</think>

<tool_call>
{"name": "wikipedia_search", "arguments": {"query": "what was the nationality of Princess Pauline of
↪   Wrttemberg"}}
</tool_call><|im_end|>
<|im_start|>user
<tool_response>"Princess Pauline of Wurttemberg (18101856)"
Princess Pauline of Wrttemberg (18101856) Princess ""Pauline"" Friederike Marie of Wrttemberg, full
↪   German name: ""Pauline Friederike Marie, Prinzessin von Wrttemberg"" (25 February 1810, Stuttgart,
↪   Kingdom of Wrttemberg  7 July 1856, Wiesbaden, Duchy of Nassau) was a member of the House of
↪   Wrttemberg and a Princess of Wrttemberg by birth. Through her marriage to William, Duke of Nassau,
↪   Louise was also a Duchess consort of Nassau. Pauline is an ancestress of the present Belgian, Danish,
↪   Dutch, Luxembourg, Norwegian, and Swedish Royal families. Pauline was the fourth child of Prince Paul
↪   of Wrttemberg and his wife Princess Charlotte of Saxe-Hildburghausen. Pauline

"Princess Pauline of Wurttemberg (18771965)"
Princess Pauline of Wrttemberg (18771965) Princess Pauline of Wrttemberg (; 19 December 18777 May 1965)
↪   was the elder daughter of William II of Wrttemberg and wife of William Frederick, Prince of Wied. She
↪   was for many years the regional director of the German Red Cross, in western Germany. Pauline was born
↪   at Stuttgart in the Kingdom of Wrttemberg, the elder daughter of William II of Wrttemberg (18481921)
↪   by his first wife Princess Marie of Waldeck and Pyrmont (18571882). She was indicted for concealing,
↪   since October 1945, a pair of important Nazis by a military court of the United States. She

"Pauline Therese of Wurttemberg"
```

```
excluded from her inheritance in his will. She died at Stuttgart, nine years later, on 10 March 1873,
↪  having lived her last years in Switzerland. Pauline had been very popular, not only for the kindness
↪  she showed to her subjects but also for the devotion she showed to the poor. Upon her death,
↪  Wrttemberg inhabitants gave her name to many roads and places in Stuttgart, Esslingen, and Friolzheim.
↪  Pauline Therese of Wrttemberg Pauline of Wrttemberg (4 September 1800  10 March 1873) was a daughter
↪  of Duke Louis of Wrttemberg and Princess Henriette of Nassau-Weilburg. She married her first cousin

"Princess Pauline of Wurttemberg (18101856)"
married William, Duke of Nassau, eldest son of Frederick William, Prince of Nassau-Weilburg and his wife,
↪  Burgravine Louise Isabelle of Kirchberg, on 23 April 1829 in Stuttgart. Pauline and William had four
↪  children: Princess Pauline of Wrttemberg (18101856) Princess ""Pauline"" Friederike Marie of
↪  Wrttemberg, full German name: ""Pauline Friederike Marie, Prinzessin von Wrttemberg"" (25 February
↪  1810, Stuttgart, Kingdom of Wrttemberg  7 July 1856, Wiesbaden, Duchy of Nassau) was a member of the
↪  House of Wrttemberg and a Princess of Wrttemberg by birth. Through her marriage to William, Duke of
↪  Nassau, Louise was also a Duchess consort of Nassau. Pauline

"Princess Pauline of Wurttemberg (18771965)"
""Herr"" and ""Frau"" Scholtz-Klink informed the French that they asked for Princess Pauline's aid in
↪  1945, Princess Pauline arranged for them to stay inconspicuously in Bebenhausen, where they were
↪  arrested by Allied authorities. Princess Pauline married on 29 October 1898 in Stuttgart to William
↪  Frederick, Prince of Wied (18721945), son of William, Prince of Wied and the spectacularly wealthy
↪  Princess Marie of the Netherlands. Her husband's elder brother was William, Prince of Albania, and
↪  she was a first cousin of the Dutch queen, Wilhelmina of the Netherlands. They had children: Princess
↪  Pauline of Wrttemberg (18771965) Princess Pauline of Wrttemberg (;</tool_response><|im_end|>
<|im_start|>assistant
<think>From the provided information, it is clear that Princess Pauline of Wrttemberg was a Princess of
↪  Wrttemberg, and she was born in the Kingdom of Wrttemberg. Therefore, her nationality was German.

So, the mother of Prince Nikolaus Wilhelm of Nassau, Princess Pauline of Wrttemberg, was from Wrttemberg,
↪  which was a part of the German Confederation.</think>

<answer>
```json
{
  "rationale": "Prince Nikolaus Wilhelm of Nassau's mother was Princess Pauline of Wrttemberg, who was a
  ↪  Princess of Wrttemberg, and thus her nationality was German."
,
  "answers": [
    "Wrttemberg",
    "German"
  ]
}
</answer><|im_end|>
```

---

***Where was the place of death of Hayranidil Kadın's husband?*** *(from 2Wiki)*
***Reference Answer:*** *Constantinople*

```
<|im_start|>assistant
<think>To find the place of death of Hayranidil Kadn's husband, first, we need to identify who Hayranidil
↪  Kadn was and then find out who her husband was. Finally, we need to determine the place of death of
↪  that husband.</think>

<tool_call>
{"name": "wikipedia_search", "arguments": {"query": "who was Hayranidil Kadn"}}
</tool_call><|im_end|>
<|im_start|>user
<tool_response>"Hayranidil Kadn"
Hayrandil Kadn died on 26 November 1895 in the Ortaky Palace, and was buried in the mausoleum of Sultan
↪  Mahmud II, located at Divan Yolu street, Istanbul. Hayranidil Kadn Hayranidil Kadn (; 21 November
↪  1846  26 November 1895), meaning 'The excellent heart', was the second wife of Sultan Abdlaziz of the
↪  Ottoman Empire. Of Circassian origin, Hayrandil Kadn was born on 21 November 1846 in Kars, Caucasus.
↪  Hayrandil married Abdlaziz in 1861 in the Dolmabahe Palace, after his accession to the throne. She
↪  was given the title of ""Ikinci Kadn"". Five years after the marriage, on 25 February 1866,

"Hayranidil Kadn"
Hayranidil Kadn Hayranidil Kadn (; 21 November 1846  26 November 1895), meaning 'The excellent heart',
↪  was the second wife of Sultan Abdlaziz of the Ottoman Empire. Of Circassian origin, Hayrandil Kadn
↪  was born on 21 November 1846 in Kars, Caucasus. Hayrandil married Abdlaziz in 1861 in the Dolmabahe
↪  Palace, after his accession to the throne. She was given the title of ""Ikinci Kadn"". Five years
↪  after the marriage, on 25 February 1866, she gave birth to her first child, a daughter, Nazma Sultan,
↪  and two years later on 29 May 1868, she gave birth to her second child, a

"Reftardil Kadn"
```

```
    son, ehzade Selaheddin Efendi bought a mansion at Ortaky. In 1934 ""Reftardil"" took the surname
↪   ""Hatgil"". She had her niece, Cemile Dilberistan Hanmefendi, married to her son, ehzade Selaheddin
↪   Efendi. She died in Ortaky Palace, Istanbul on 3 March 1936. Reftardil Kadn Reftardil Kadn (5 June
↪   1838  3 March 1936) was the second wife of Sultan Murad V. Reftardil Kadn was born in 1838 to an
↪   Abadzekh notable, Hatkoyuko Hatko. Her family used to live at the coast of Black Sea. She had two
↪   sisters Terandil Hanm and Ceylanmelek Hanm who were also given to the palace along with

"Reftardil Kadn"
Reftardil Kadn Reftardil Kadn (5 June 1838  3 March 1936) was the second wife of Sultan Murad V. Reftardil
↪   Kadn was born in 1838 to an Abadzekh notable, Hatkoyuko Hatko. Her family used to live at the coast of
↪   Black Sea. She had two sisters Terandil Hanm and Ceylanmelek Hanm who were also given to the palace
↪   along with her and were later married to other high officials. She was a lovely lady with a pink skin,
↪   large blue eyes, straight nose, and a round face. She was also honest and goodhearted. However, in
↪   the later years Reftardil was

"Duzdidil Kadn"
Dzdidil Kadn Dzdidil Kadn (; 182518 August 1845) was the third wife of Sultan Abdulmejid I of the Ottoman
↪   Empire. Dzdidil Kadn was born in 1825 in North Caucasus. Born as Aye Dian, she was a member of Ubykh
↪   family, Dian. Her father was hm Bey Dian and her mother was an Abkhazian princess belonging to
↪   Shervashidze. Upon Yahya Bey's decision, Aye had been brought to Istanbul as a young child, where she
↪   entrusted to the imperial harem, along with her nanny Cinan Hanm, and a maid Emine Hanm. Here her name
↪   according to the costum of the Ottoman court</tool_response><|im_end|>
<|im_start|>assistant
<think>From the provided responses, it is clear that Hayranidil Kadn was the second wife of Sultan
↪   Abdlaziz of the Ottoman Empire. So, the next step is to identify the place of death of Sultan
↪   Abdlaziz.</think>

<tool_call>
{"name": "wikipedia_search", "arguments": {"query": "where did Sultan Abdlaziz die"}}
</tool_call><|im_end|>
<|im_start|>user
<tool_response>"Abdulaziz"
Abdlaziz Abdlaziz (Ottoman Turkish:    / ""`Abdl-`Azz"", ; 8 February 18304 June 1876) was the 32nd Sultan
↪   of the Ottoman Empire and reigned between 25 June 1861 and 30 May 1876. He was the son of Sultan Mahmud
↪   II and succeeded his brother Abdulmejid I in 1861. Born at Eyp Palace, Constantinople (present-day
↪   Istanbul), on 8 February 1830, Abdlaziz received an Ottoman education but was nevertheless an ardent
↪   admirer of the material progress that was made in the West. He was the first Ottoman Sultan who
↪   travelled to Western Europe, visiting a number of important European capitals including Paris,

"Sultan bin Abdulaziz Al Saud"
be abroad at a given time. The Saudi Royal court announced on 22 October 2011 that Prince Sultan died at
↪   dawn of an unspecified illness. According to media reports, Prince Sultan had been battling cancer
↪   and had been seeking medical treatment in the United States since mid-June 2011. He had a surgical
↪   operation in New York in July 2011. Unnamed U.S. officials cited by ""The New York Times"" stated
↪   that he died at New York-Presbyterian Hospital in Manhattan. His body was taken from New York City to
↪   Riyadh on 24 October 2011. His funeral was held at the Imam Turki

"Abdulaziz"
having been applied to the throat. Abdlaziz had five consorts: Abdlaziz Abdlaziz (Ottoman Turkish:    /
↪   ""`Abdl-`Azz"", ; 8 February 18304 June 1876) was the 32nd Sultan of the Ottoman Empire and reigned
↪   between 25 June 1861 and 30 May 1876. He was the son of Sultan Mahmud II and succeeded his brother
↪   Abdulmejid I in 1861. Born at Eyp Palace, Constantinople (present-day Istanbul), on 8 February 1830,
↪   Abdlaziz received an Ottoman education but was nevertheless an ardent admirer of the material
↪   progress that was made in the West. He was the first Ottoman Sultan who travelled to Western

"Abdulaziz"
Balkan provinces. In 1876, the April Uprising saw insurrection spreading among the Bulgarians. Ill feeling
↪   mounted against Russia for its encouragement of the rebellions. While no one event led to his being
↪   deposed, the crop failure of 1873 and his lavish expenditures on the Ottoman Navy and on new palaces
↪   which he had built, along with mounting public debt, helped to create an atmosphere conducive to his
↪   being overthrown. Abdlaziz was deposed by his ministers on 30 May 1876. Abdlaziz's death at raan
↪   Palace in Constantinople a few days later was documented as a suicide at the time, but suspicions

"Sultan bin Abdulaziz Al Saud"
Sultan bin Abdulaziz Al Saud Sultan bin Abdulaziz Al Saud (30 December 1925  22 October 2011) (), called
↪   ""Sultan al-Khair"" (, ""Sultan the Good"") in Saudi Arabia, was a Saudi nobleman and government
↪   official. A son of King Abdulaziz (Ibn Saud), he was Saudi Defense minister from 1963 to his death in
↪   2011 and the Crown Prince of Saudi Arabia from 2005 to 2011. Sultan was born in Riyadh, Najd Sultanate
↪   in 1928. He was the 12th son of King Abdulaziz and his mother was Hassa bint Ahmed Al Sudairi. He was
↪   the second of the Sudairi Seven. Prince</tool_response><|im_end|>
<|im_start|>assistant
<think>From the provided responses, it is clear that Sultan Abdlaziz died on 4 June 1876.

Now, we need to find the place of his death.</think>

<tool_call>
{"name": "wikipedia_search", "arguments": {"query": "where did Sultan Abdlaziz died, place"}}
</tool_call><|im_end|>
<|im_start|>user
<tool_response>"Abdulaziz"
```

```
Abdlaziz Abdlaziz (Ottoman Turkish:    / ""`Abdl-`Azz"", ; 8 February 18304 June 1876) was the 32nd Sultan
↪  of the Ottoman Empire and reigned between 25 June 1861 and 30 May 1876. He was the son of Sultan Mahmud
↪  II and succeeded his brother Abdulmejid I in 1861. Born at Eyp Palace, Constantinople (present-day
↪  Istanbul), on 8 February 1830, Abdlaziz received an Ottoman education but was nevertheless an ardent
↪  admirer of the material progress that was made in the West. He was the first Ottoman Sultan who
↪  travelled to Western Europe, visiting a number of important European capitals including Paris,

"Abdulaziz"
having been applied to the throat. Abdlaziz had five consorts: Abdlaziz Abdlaziz (Ottoman Turkish:    /
↪  ""`Abdl-`Azz"", ; 8 February 18304 June 1876) was the 32nd Sultan of the Ottoman Empire and reigned
↪  between 25 June 1861 and 30 May 1876. He was the son of Sultan Mahmud II and succeeded his brother
↪  Abdulmejid I in 1861. Born at Eyp Palace, Constantinople (present-day Istanbul), on 8 February 1830,
↪  Abdlaziz received an Ottoman education but was nevertheless an ardent admirer of the material
↪  progress that was made in the West. He was the first Ottoman Sultan who travelled to Western

"Sultan bin Abdulaziz Al Saud"
be abroad at a given time. The Saudi Royal court announced on 22 October 2011 that Prince Sultan died at
↪  dawn of an unspecified illness. According to media reports, Prince Sultan had been battling cancer
↪  and had been seeking medical treatment in the United States since mid-June 2011. He had a surgical
↪  operation in New York in July 2011. Unnamed U.S. officials cited by ""The New York Times"" stated
↪  that he died at New York-Presbyterian Hospital in Manhattan. His body was taken from New York City to
↪  Riyadh on 24 October 2011. His funeral was held at the Imam Turki

"Abdulaziz"
Balkan provinces. In 1876, the April Uprising saw insurrection spreading among the Bulgarians. Ill feeling
↪  mounted against Russia for its encouragement of the rebellions. While no one event led to his being
↪  deposed, the crop failure of 1873 and his lavish expenditures on the Ottoman Navy and on new palaces
↪  which he had built, along with mounting public debt, helped to create an atmosphere conducive to his
↪  being overthrown. Abdlaziz was deposed by his ministers on 30 May 1876. Abdlaziz's death at raan
↪  Palace in Constantinople a few days later was documented as a suicide at the time, but suspicions

"Sultan bin Abdulaziz Al Saud"
Sultan bin Abdulaziz Al Saud Sultan bin Abdulaziz Al Saud (30 December 1925  22 October 2011) (), called
↪  ""Sultan al-Khair"" (, ""Sultan the Good"") in Saudi Arabia, was a Saudi nobleman and government
↪  official. A son of King Abdulaziz (Ibn Saud), he was Saudi Defense minister from 1963 to his death in
↪  2011 and the Crown Prince of Saudi Arabia from 2005 to 2011. Sultan was born in Riyadh, Najd Sultanate
↪  in 1928. He was the 12th son of King Abdulaziz and his mother was Hassa bint Ahmed Al Sudairi. He was
↪  the second of the Sudairi Seven. Prince</tool_response><|im_end|>
<|im_start|>assistant
<think>From the provided responses, it is clear that Sultan Abdlaziz died at raan Palace in
↪  Constantinople (present-day Istanbul).

So, the place of death of Hayranidil Kadn's husband (Sultan Abdlaziz) was raan Palace in Constantinople
↪  (present-day Istanbul).</think>

<answer>
```json
{
  "rationale": "Hayranidil Kadn was the second wife of Sultan Abdlaziz of the Ottoman Empire, and his
  ↪  place of death was raan Palace in Constantinople."
,
  "answers": [
    "raan Palace",
    "Constantinople"
  ]
}
</answer><|im_end|>
```

**Where does the director of film Wine Of Morning work at?** *(from HotpotQA)*
**Reference Answer:** *Bob Jones University*

```
<|im_start|>assistant
<think>To find where the director of the film "Wine of Morning" works, first, we need to identify who the
↪  director of the film "Wine of Morning" is. </think>

<tool_call>
{"name": "wikipedia_search", "arguments": {"query": "who directed the film Wine Of Morning"}}
</tool_call><|im_end|>
<|im_start|>user
<tool_response>"Wine of Morning"
Wine of Morning Wine of Morning is an American feature film from 1955 and has Barabbas as the subject, who
↪  was pardoned according to the Biblical report in place of Jesus Christ by Pontius Pilate. The film is
↪  based on the novel Wine of Morning by Bob Jones Jr and directed by Katherine Stenholm. On a stormy
↪  voyage Barabbas writes a letter to his friend Stephen and remembers his time in Galilean Nazareth.
↪  There he is friends with the carpenter Josef and his son Jesus . One day he meets Irene, the future
↪  bride of his friend Stephanus, and falls
```

```
"Wine of Morning"
"""Wine of Morning""" of the University President, Bob Jones, Jr., 1950, who had long been planning to
↪  write a novel about Barabbas, but did not find the time to write until a pleurisy forced him into the
↪  hospital bed for two months. Six months later, the novel was completed. The novel was finally filmed
↪  by Unusual Films; Bob Jones Jr. took over the role of Pontius Pilatus. Wine of Morning was featured at
↪  the International Congress of Motion's Picture and Television School Directors at the Cannes
↪  International Film Festival. He was the first movie to win the four major awards

"Wine of Morning"
from the """National Evangelical Film Foundation""". Wine of Morning was Katherine Helmond's movie debut.
↪  Wine of Morning in the Internet Movie Database (English) Wine of Morning on www.unusualfilms.com Wine
↪  of Morning Wine of Morning is an American feature film from 1955 and has Barabbas as the subject, who
↪  was pardoned according to the Biblical report in place of Jesus Christ by Pontius Pilate. The film is
↪  based on the novel Wine of Morning by Bob Jones Jr and directed by Katherine Stenholm. On a stormy
↪  voyage Barabbas writes a letter to his friend Stephen and remembers his time in Galilean

"The Wine of Summer"
The Wine of Summer The Wine of Summer is a 2013 romantic drama film written, directed and produced by
↪  Maria Matteoli, starring Elsa Pataky, Sonia Braga, Ethan Peck, Najwa Nimri, Bob Wells and Marcia Gay
↪  Harden. James (Peck), at the age of 27 quits his law career in pursuit of his childhood dream of
↪  becoming an actor. While studying acting under the tutelage of Shelley (Harden), he becomes engrossed
↪  in Carlo Lucchesi's play, Tinto de Verano, which is set in Spain. James girlfriend Brit (Chow) leaves
↪  him, and he spontaneously flies to Spain, where he encounters the playwright Lucchesi (Wells)

"Bob Jones University"
Jones, Jr. selected a speech teacher, Katherine Stenholm, as the first director. Although she had no
↪  experience in cinema, she took summer courses at the University of Southern California and received
↪  personal instruction from Hollywood specialists, such as Rudolph Sternad. Unusual Films has produced
↪  seven feature-length films, each with an evangelistic emphasis: """Wine of Morning""", """Red Runs the
↪  River""", """Flame in the Wind""", """Sheffey""", """Beyond the Night""", """The Printing""", and """Milltown
↪  Pride""". """Wine of Morning""" (1955), based on a novel by Bob Jones, Jr., represented the United States
↪  at the Cannes Film Festival. The first four films are historical dramas</tool_response><|im_end|>
<|im_start|>assistant
<think>From the provided responses, it is clear that the director of the film "Wine of Morning" is
↪  Katherine Stenholm.

Next, we need to find out where Katherine Stenholm works.</think>

<tool_call>
{"name": "wikipedia_search", "arguments": {"query": "where did director Katherine Stenholm work"}}
</tool_call><|im_end|>
<|im_start|>user
<tool_response>"Katherine Stenholm"
Katherine Stenholm Katherine Corne Stenholm (June 19, 1917  November 3, 2015) was an American film
↪  director and the founding director of Unusual Films, the production company of Bob Jones University.
↪  Katherine Corne was born and reared in Hendersonville, North Carolina. As a high school student
↪  during the Depression, she supplemented her family's income by writing movie reviews for a local
↪  newspaper. Rejecting a college scholarship to Wellesley, Corne attended the fledgling Bob Jones
↪  College in Cleveland, Tennessee, after an evangelist convinced her that a Christian young person
↪  should attend a Christian college. At BJC, she majored in speech and

"Katherine Stenholm"
of American cinema training and the film's frank religious message would """provide a revealing contrast
↪  to the entries from Russia and the other Communist-dominated countries.""" In 1986, Stenholm suffered
↪  a stroke in the Soviet Union while taking scenic footage in preparation for another feature-length
↪  film. She retired as director of Unusual Films but continued to teach at BJU until 2001. Stenholm
↪  died in November 2015 at the age of 98. Katherine Stenholm Katherine Corne Stenholm (June 19, 1917
↪  November 3, 2015) was an American film director and the founding director of Unusual Films, the
↪  production company of Bob Jones

"Katherine Stenholm"
became a private student of Bob Jones, Jr., eventually helping him direct Shakespearean plays. After
↪  earning her undergraduate degree, she served on the BJC speech faculty while attending graduate
↪  school at Northwestern University for twelve summers. During this period she married Gilbert R.
↪  Stenholm (191589), who became an influential administrator at the institution; they had one son. In
↪  1950, after the college moved to Greenville and became Bob Jones University, Bob Jones, Sr. and Jr.
↪  asked Stenholm to head a newly conceived campus film production company, Unusual Films. Stenholm then
↪  attended summer film school at the University of Southern California,

"Katherine Stenholm"
making important professional contacts and serving an internship with Stanley Kramer. Stenholm was a
↪  quick learner and soon """became one of only a handful of women in the United States to direct feature
↪  films.""" Through her career she produced seventy-two films of various types including sermon films,
↪  religious documentaries, promotional films, and multi-image presentations. She directed five
↪  feature-length religious films, all costume dramas: The National Evangelical Film Foundation named
↪  Stenholm Director of the Year in 1953, 1955, and 1963; and her favorite film, """Sheffey""", received a
↪  Silver Medallion award from the International Film and Television Festival of New York. In

"Katherine Stenholm"
```

```
    1958, at the height of the Cold War, the University Film Producers Association selected ""Wine of
↪   Morning"" as its submission to the International Congress of Motion Picture and Television School
↪   Directors at the Cannes Film Festival, and Stenholm was the keynote speaker on the occasion. A U.S.
↪   State Department official who briefed Stenholm told her there had been a round of applause when the
↪   Department discovered that BJU had been chosen to represent the United States because ""Bob Jones
↪   University is one school about which there is no worry!"" The selection committee thought ""Wine of
↪   Morning"" would demonstrate the excellence</tool_response><|im_end|>
<|im_start|>assistant
<tool_call>
{"name": "wikipedia_search", "arguments": {"query": "where did Katherine Stenholm work as a film
↪   director"}}
</tool_call><|im_end|>
<|im_start|>user
<tool_response>"Katherine Stenholm"
Katherine Stenholm Katherine Corne Stenholm (June 19, 1917  November 3, 2015) was an American film
↪   director and the founding director of Unusual Films, the production company of Bob Jones University.
↪   Katherine Corne was born and reared in Hendersonville, North Carolina. As a high school student
↪   during the Depression, she supplemented her family's income by writing movie reviews for a local
↪   newspaper. Rejecting a college scholarship to Wellesley, Corne attended the fledgling Bob Jones
↪   College in Cleveland, Tennessee, after an evangelist convinced her that a Christian young person
↪   should attend a Christian college. At BJC, she majored in speech and

"Katherine Stenholm"
of American cinema training and the film's frank religious message would ""provide a revealing contrast
↪   to the entries from Russia and the other Communist-dominated countries."" In 1986, Stenholm suffered
↪   a stroke in the Soviet Union while taking scenic footage in preparation for another feature-length
↪   film. She retired as director of Unusual Films but continued to teach at BJU until 2001. Stenholm
↪   died in November 2015 at the age of 98. Katherine Stenholm Katherine Corne Stenholm (June 19, 1917
↪   November 3, 2015) was an American film director and the founding director of Unusual Films, the
↪   production company of Bob Jones

"Katherine Stenholm"
became a private student of Bob Jones, Jr., eventually helping him direct Shakespearean plays. After
↪   earning her undergraduate degree, she served on the BJC speech faculty while attending graduate
↪   school at Northwestern University for twelve summers. During this period she married Gilbert R.
↪   Stenholm (191589), who became an influential administrator at the institution; they had one son. In
↪   1950, after the college moved to Greenville and became Bob Jones University, Bob Jones, Sr. and Jr.
↪   asked Stenholm to head a newly conceived campus film production company, Unusual Films. Stenholm then
↪   attended summer film school at the University of Southern California,

"Katherine Stenholm"
making important professional contacts and serving an internship with Stanley Kramer. Stenholm was a
↪   quick learner and soon ""became one of only a handful of women in the United States to direct feature
↪   films."" Through her career she produced seventy-two films of various types including sermon films,
↪   religious documentaries, promotional films, and multi-image presentations. She directed five
↪   feature-length religious films, all costume dramas: The National Evangelical Film Foundation named
↪   Stenholm Director of the Year in 1953, 1955, and 1963; and her favorite film, ""Sheffey"", received a
↪   Silver Medallion award from the International Film and Television Festival of New York. In

"Katherine Stenholm"
1958, at the height of the Cold War, the University Film Producers Association selected ""Wine of
↪   Morning"" as its submission to the International Congress of Motion Picture and Television School
↪   Directors at the Cannes Film Festival, and Stenholm was the keynote speaker on the occasion. A U.S.
↪   State Department official who briefed Stenholm told her there had been a round of applause when the
↪   Department discovered that BJU had been chosen to represent the United States because ""Bob Jones
↪   University is one school about which there is no worry!"" The selection committee thought ""Wine of
↪   Morning"" would demonstrate the excellence</tool_response><|im_end|>
<|im_start|>assistant
<think>From the provided responses, it is clear that Katherine Stenholm worked at Bob Jones University as
↪   the founding director of Unusual Films, which is the production company of Bob Jones University.

Additionally, she also taught at Bob Jones University until 2001.

So, where Katherine Stenholm worked includes Bob Jones University.</think>

<answer>
```json
{
  "rationale": "Katherine Stenholm directed the film 'Wine of Morning,' and she worked at Bob Jones
  ↪   University."
  ,
  "answers": [
    "Bob Jones University",
    "Unusual Films"
  ]
}
</answer><|im_end|>
```

> **How are Ceephax Acid Crew and Squarepusher's music similar?** *(from HotpotQA)*
> **Reference Answer:** *drum and bass electronic musician*

```
<|im_start|>assistant
<think>To find how Ceephax Acid Crew and Squarepusher's music are similar, first, we need to search who
↪  Ceephax Acid Crew and Squarepusher are.</think>

<tool_call>
{"name": "wikipedia_search", "arguments": {"query": "who is Ceephax Acid Crew"}}
</tool_call><|im_end|>
<|im_start|>user
<tool_response>"Ceephax Acid Crew"
Ceephax Acid Crew Ceephax Acid Crew (real name Andy Jenkinson) is a British acid techno and drum and bass
↪  electronic musician. Jenkinson is also known simply by the pseudonym Ceephax, which is a reference to
↪  the BBC teletext service Ceefax. He is the brother of Squarepusher (Tom Jenkinson). Ceephax's music
↪  from 1997 to 2002 and beyond used mainly vintage (especially Roland) drum machines and synthesizers
↪  such as the TB-303. This music was often recorded onto a cassette tape deck. The music was released on
↪  vinyl records and cassette tape on underground labels such as Breakin' Records, Lo Recordings and
↪  Firstcask.

"Ceephax Acid Crew"
old school house, acid house, techno, drum and bass and gabber all in Ceephax's distinctive style. He has
↪  also produced several music videos predominantly using old video equipment and early computer
↪  animation. Ceephax Acid Crew Ceephax Acid Crew (real name Andy Jenkinson) is a British acid techno
↪  and drum and bass electronic musician. Jenkinson is also known simply by the pseudonym Ceephax, which
↪  is a reference to the BBC teletext service Ceefax. He is the brother of Squarepusher (Tom Jenkinson).
↪  Ceephax's music from 1997 to 2002 and beyond used mainly vintage (especially Roland) drum machines
↪  and synthesizers such as the

Squarepusher
Squarepusher Squarepusher is the principal pseudonym adopted by Tom Jenkinson (born 17 January 1975), an
↪  English electronic musician. His compositions draw on a number of influences including drum and bass,
↪  acid house, jazz and electroacoustic music. His recordings are typified by a combination of
↪  electronic sound sources, live instrumental playing and digital signal processing. He is the brother
↪  of Ceephax Acid Crew (Andrew Jenkinson). Tom Jenkinson grew up in Chelmsford, Essex. The first school
↪  he attended was affiliated with Chelmsford Cathedral giving him exposure to organ music, which he has
↪  subsequently acknowledged as an influence on his work. He took

"Ceephax Acid Crew"
This primitive Acid House aesthetic and methodology countered the growing popularity of the computer
↪  music and compact disc releases of the time. Rephlex Records and Warp Records also released remixes
↪  of Squarepusher by Ceephax around this time. From 2003 onwards as well as his more typical Acid style
↪  he has also released drum and bass songs made on old samplers, a Commodore Amiga and various early
↪  digital synthesizers, again displaying an interest in late 1980s and early 1990s dance music. His set
↪  on Mary Anne Hobb's show Breezeblock in 2003 and more recent tracks such as ""Castilian"" and
↪  ""Arcadian"" also

"Ceephax Acid Crew"
indicate an interest in Chiptune music. From 2007 he has had full releases on the Rephlex and Planet Mu
↪  record labels. He also continues to release on Firstcask and other small independent labels such as
↪  WM, Bugklinik and his own label Waltzer. Ceephax is popular with some electronic music fans for his
↪  live performances. Ceephax's shows are set apart from the popular Laptop style of ""live"" electronic
↪  performance by exclusively using only analogue and early digital equipment and occasionally an Amiga
↪  computer. Hardware frequently used include a TB-303, TR-909, TR-707, SH-101, Kenton Pro-2000, and
↪  Yamaha RS7000. These sets range from</tool_response><|im_end|>
<|im_start|>assistant
<think>From the provided responses, it is clear that Ceephax Acid Crew and Squarepusher are both
↪  electronic musicians, and they are brothers (Ceephax Acid Crew being the pseudonym of Andy Jenkinson
↪  and Squarepusher being the pseudonym of Tom Jenkinson).

Next, we need to identify the characteristics of their music to find out how their music is
↪  similar.</think>

<tool_call>
{"name": "wikipedia_search", "arguments": {"query": "how is Squarepusher's music similar to electronic
↪  music"}}
</tool_call><|im_end|>
<|im_start|>user
<tool_response>Squarepusher
Squarepusher Squarepusher is the principal pseudonym adopted by Tom Jenkinson (born 17 January 1975), an
↪  English electronic musician. His compositions draw on a number of influences including drum and bass,
↪  acid house, jazz and electroacoustic music. His recordings are typified by a combination of
↪  electronic sound sources, live instrumental playing and digital signal processing. He is the brother
↪  of Ceephax Acid Crew (Andrew Jenkinson). Tom Jenkinson grew up in Chelmsford, Essex. The first school
↪  he attended was affiliated with Chelmsford Cathedral giving him exposure to organ music, which he has
↪  subsequently acknowledged as an influence on his work. He took

Squarepusher
```

```
synth bassline in ""The Modern Bass Guitar"" was triggered from an electric bass using the midi bass
↪  system. January 2005 saw the re-establishment of the studio at Tom's residence in Essex. The pieces
↪  ""Theme From Sprite"", ""Bubble Life"", ""Vacuum Garden"", ""Circlewave 2"" and ""Orient Orange""
↪  were all made in early 2005. All of these tracks are based on live drumming tracks, which Tom had
↪  planned out quite meticulously beforehand, in contrast to similarly realised tracks on ""Music Is
↪  Rotted One Note"". ""Hello Meow"", ""Planetarium"", ""Rotate Electrolyte"" and ""Plotinus"" were made
↪  over the Summer and Autumn of 2005. The track ""Hello

"Nothing Records"
aboard Nothing via a licensing deal with Warp. Squarepusher is Tom Jenkinson of Chelmsford, Essex,
↪  England. After learning bass guitar and playing in local bands in his youth, Tom began taking an
↪  interest in techno and house music in the early 1990s. Amongst his influences were Aphex Twin and
↪  future Nothing Records label-mate Luke Vibert, particularly with his work as Plug. Jenkinson said of
↪  Plug's track ""Military Jazz,"" ""This track came on and, amongst the road noise and chatter, I heard
↪  what I thought was some sort of hip-hop track being played by a band. As the track progressed, I

"Intelligent dance music"
Intelligent dance music Intelligent dance music (commonly abbreviated as IDM) is a form of electronic
↪  music originating in the early 1990s. Relative to other forms of electronic and rave music, IDM came
↪  to be characterized as better suited to home listening than dancing. Inspired by a variety of
↪  sources, including Detroit techno, acid house, ambient music, and breakbeat, IDM tended to rely upon
↪  individualistic experimentation rather than adhering to characteristics associated with specific
↪  genres. Prominent artists associated with the genre include Aphex Twin, -Ziq, the Black Dog, the Orb,
↪  the Future Sound of London, Autechre, Luke Vibert, Squarepusher, Venetian Snares

Squarepusher
""apparently one of Bjrk's favourite songs"". This period also produced the ""Maximum Priest"" e.p. ""Our
↪  Underwater Torch"" was partially inspired by a developing obsession Tom had for the sounds of water.
↪  Tom states that this piece was obliquely inspired by the films ""Solaris"" and ""Stalker"" by the
↪  Russian director Andrei Tarkovsky. The organ in this piece is triggered directly from an electric
↪  bass. ""Decathlon Oxide"" carried on the ideas initiated in ""Fly Street"" and ""Varkatope"" from
↪  ""Budakhan Mindphone"" and features a Gamelan gong. The record also contains remixes by Luke Vibert,
↪  Autechre and Matthew Yee King. Very few live shows</tool_response><|im_end|>
<|im_start|>assistant
<think>From the provided responses, it is clear that Squarepusher's music draws on influences including
↪  drum and bass, acid house, jazz, and electroacoustic music, and it is characterized by a combination
↪  of electronic sound sources, live instrumental playing, and digital signal processing.

Now, let's identify the characteristics of Ceephax Acid Crew's music and compare it with
↪  Squarepusher's.</think>

<tool_call>
{"name": "wikipedia_search", "arguments": {"query": "how is Ceephax Acid Crew's music similar to
↪  electronic music"}}
</tool_call><|im_end|>
<|im_start|>user
<tool_response>"Ceephax Acid Crew"
Ceephax Acid Crew Ceephax Acid Crew (real name Andy Jenkinson) is a British acid techno and drum and bass
↪  electronic musician. Jenkinson is also known simply by the pseudonym Ceephax, which is a reference to
↪  the BBC teletext service Ceefax. He is the brother of Squarepusher (Tom Jenkinson). Ceephax's music
↪  from 1997 to 2002 and beyond used mainly vintage (especially Roland) drum machines and synthesizers
↪  such as the TB-303. This music was often recorded onto a cassette tape deck. The music was released on
↪  vinyl records and cassette tape on underground labels such as Breakin' Records, Lo Recordings and
↪  Firstcask.

"Ceephax Acid Crew"
old school house, acid house, techno, drum and bass and gabber all in Ceephax's distinctive style. He has
↪  also produced several music videos predominantly using old video equipment and early computer
↪  animation. Ceephax Acid Crew Ceephax Acid Crew (real name Andy Jenkinson) is a British acid techno
↪  and drum and bass electronic musician. Jenkinson is also known simply by the pseudonym Ceephax, which
↪  is a reference to the BBC teletext service Ceefax. He is the brother of Squarepusher (Tom Jenkinson).
↪  Ceephax's music from 1997 to 2002 and beyond used mainly vintage (especially Roland) drum machines
↪  and synthesizers such as the

"Acid jazz"
these recordings and who emphasised a groove-based approach to music. Acid jazz uses elements of jazz,
↪  funk and hip-hop. Because of its existence as a percussion-heavy, primarily live music, it was closer
↪  to jazz than any other dance style, but its focus on maintaining a groove allied it with funk, hip-hop,
↪  and dance music. The style is characterized by danceable grooves and long, repetitive compositions.
↪  Typical ensembles include horns, a full rhythm section (often with a drum set and additional
↪  percussion), a vocalist that may sing and rap and often a DJ. Acid jazz has its origins in the 1960s,

"Acid techno"
```

```
Acid techno Acid techno (also known as simply ""acid"") is a genre of techno that developed out of late
↪   1980s Chicago acid house, particularly its ""squelching"" synthesizer sound obtained by using very
↪   distinctive instruments created mainly by Roland, such as the SH-101 and TB-303 for bass and lead
↪   sounds, and the TR-707, TR-808, and TR-909 for percussion. ""Acid"" specifically refers to the use of
↪   the Roland TB-303, or any other synthesizer designed to emulate its unique sound. While modern
↪   electronic instruments have memory banks of different sounds or ""patches"", these machines had to be
↪   manually set by adjusting control knobs.

"Ceephax Acid Crew"
indicate an interest in Chiptune music. From 2007 he has had full releases on the Rephlex and Planet Mu
↪   record labels. He also continues to release on Firstcask and other small independent labels such as
↪   WM, Bugklinik and his own label Waltzer. Ceephax is popular with some electronic music fans for his
↪   live performances. Ceephax's shows are set apart from the popular Laptop style of ""live"" electronic
↪   performance by exclusively using only analogue and early digital equipment and occasionally an Amiga
↪   computer. Hardware frequently used include a TB-303, TR-909, TR-707, SH-101, Kenton Pro-2000, and
↪   Yamaha RS7000. These sets range from</tool_response><|im_end|>
<|im_start|>assistant
<think>From the provided responses, it is clear that Ceephax Acid Crew's music is characterized by acid
↪   techno and drum and bass, often using vintage synthesizers and drum machines like the TB-303.

Now, let's identify the similarities between the music of Ceephax Acid Crew and Squarepusher.</think>

<tool_call>
{"name": "wikipedia_search", "arguments": {"query": "what is similar about the music of Ceephax Acid Crew
↪   and Squarepusher"}}
</tool_call><|im_end|>
<|im_start|>user
<tool_response>"Ceephax Acid Crew"
Ceephax Acid Crew Ceephax Acid Crew (real name Andy Jenkinson) is a British acid techno and drum and bass
↪   electronic musician. Jenkinson is also known simply by the pseudonym Ceephax, which is a reference to
↪   the BBC teletext service Ceefax. He is the brother of Squarepusher (Tom Jenkinson). Ceephax's music
↪   from 1997 to 2002 and beyond used mainly vintage (especially Roland) drum machines and synthesizers
↪   such as the TB-303. This music was often recorded onto a cassette tape deck. The music was released on
↪   vinyl records and cassette tape on underground labels such as Breakin' Records, Lo Recordings and
↪   Firstcask.

Squarepusher
Squarepusher Squarepusher is the principal pseudonym adopted by Tom Jenkinson (born 17 January 1975), an
↪   English electronic musician. His compositions draw on a number of influences including drum and bass,
↪   acid house, jazz and electroacoustic music. His recordings are typified by a combination of
↪   electronic sound sources, live instrumental playing and digital signal processing. He is the brother
↪   of Ceephax Acid Crew (Andrew Jenkinson). Tom Jenkinson grew up in Chelmsford, Essex. The first school
↪   he attended was affiliated with Chelmsford Cathedral giving him exposure to organ music, which he has
↪   subsequently acknowledged as an influence on his work. He took

"Ceephax Acid Crew"
old school house, acid house, techno, drum and bass and gabber all in Ceephax's distinctive style. He has
↪   also produced several music videos predominantly using old video equipment and early computer
↪   animation. Ceephax Acid Crew Ceephax Acid Crew (real name Andy Jenkinson) is a British acid techno
↪   and drum and bass electronic musician. Jenkinson is also known simply by the pseudonym Ceephax, which
↪   is a reference to the BBC teletext service Ceefax. He is the brother of Squarepusher (Tom Jenkinson).
↪   Ceephax's music from 1997 to 2002 and beyond used mainly vintage (especially Roland) drum machines
↪   and synthesizers such as the

"Ceephax Acid Crew"
This primitive Acid House aesthetic and methodology countered the growing popularity of the computer
↪   music and compact disc releases of the time. Rephlex Records and Warp Records also released remixes
↪   of Squarepusher by Ceephax around this time. From 2003 onwards as well as his more typical Acid style
↪   he has also released drum and bass songs made on old samplers, a Commodore Amiga and various early
↪   digital synthesizers, again displaying an interest in late 1980s and early 1990s dance music. His set
↪   on Mary Anne Hobb's show Breezeblock in 2003 and more recent tracks such as ""Castilian"" and
↪   ""Arcadian"" also

"Selection Sixteen"
Selection Sixteen Selection Sixteen is a 1999 album by Squarepusher, released on Warp Records. According
↪   to the CD, the cover-art is from a film called ""Acid Trayners 4"", which seems to be different
↪   pictures of an oscilloscope. Throughout the record, the bass, hooked to an audio/midi converter,
↪   plays an important part in the composition of synth and drum parts. Most releases of ""Selection
↪   Sixteen"" include ""Anti-Greylord Protection Scheme Prelude"", which was originally released as a
↪   separate EP with a slightly different track order. ""Ceephax Mix"" was created by Squarepusher's
↪   brother, Andy Jenkinson, aka Ceephax Acid Crew. On the original release,</tool_response><|im_end|>
<|im_start|>assistant
<think>From the provided responses, it is evident that both Ceephax Acid Crew and Squarepusher are
↪   involved with electronic music genres like acid house and drum and bass, and they use vintage
↪   synthesizers and drum machines.

Specifically, both artists use elements of acid house and drum and bass, and they utilize vintage
↪   electronic instruments such as the TB-303.</think>
```

```
<answer>
```json
{
  "rationale": "Ceephax Acid Crew and Squarepusher are both electronic musicians who produce music in
  ↪  genres such as acid house and drum and bass, and they use vintage synthesizers and drum machines."
  ,
  "answers": [
    "electronic music",
    "acid house",
    "drum and bass"
  ]
}
</answer><|im_end|>
```

> **What is the primary male hormone derived from?** *(from Bamboogle)*
> **Reference Answer:** *cholesterol*

```
<|im_start|>assistant
<think>To find the primary male hormone derived from, first, we need to identify what the primary male
↪  hormone is and then find its derivation. Let's start by identifying the primary male hormone.</think>

<tool_call>
{"name": "wikipedia_search", "arguments": {"query": "what is the primary male hormone"}}
</tool_call><|im_end|>
<|im_start|>user
<tool_response>Testosterone
Testosterone Testosterone is the primary male sex hormone and an anabolic steroid. In male humans,
↪  testosterone plays a key role in the development of male reproductive tissues such as testes and
↪  prostate, as well as promoting secondary sexual characteristics such as increased muscle and bone
↪  mass, and the growth of body hair. In addition, testosterone is involved in health and well-being,
↪  and the prevention of osteoporosis. Insufficient levels of testosterone in men may lead to
↪  abnormalities including frailty and bone loss. Testosterone is a steroid from the androstane class
↪  containing a keto and hydroxyl groups at the three and seventeen

Testosterone
Testosterone is observed in most vertebrates. Testosterone and the classical nuclear androgen receptor
↪  first appeared in gnathostomes (jawed vertebrates). Agnathans (jawless vertebrates) such as lampreys
↪  do not produce testosterone but instead use androstenedione as a male sex hormone. Fish make a
↪  slightly different form called 11-ketotestosterone. Its counterpart in insects is ecdysone. The
↪  presence of these ubiquitous steroids in a wide range of animals suggest that sex hormones have an
↪  ancient evolutionary history. Testosterone Testosterone is the primary male sex hormone and an
↪  anabolic steroid. In male humans, testosterone plays a key role in the development of male
↪  reproductive tissues

Androgen
Androgen An androgen (from Greek ""andr-"", the stem of the word meaning ""man"") is any natural or
↪  synthetic steroid hormone which regulates the development and maintenance of male characteristics in
↪  vertebrates by binding to androgen receptors. This includes the embryological development of the
↪  primary male sex organs, and the development of male secondary sex characteristics at puberty.
↪  Androgens are synthesized in the testes, the ovaries, and the adrenal glands. Androgens increase in
↪  both boys and girls during puberty. The major androgen in males is testosterone. Dihydrotestosterone
↪  (DHT) and androstenedione are of equal importance in male development. DHT ""in utero"" causes

Prostate
sex characteristics. The main male hormone is testosterone, which is produced mainly by the testicles. It
↪  is dihydrotestosterone (DHT), a metabolite of testosterone, that predominantly regulates the
↪  prostate. About 20,000 protein coding genes are expressed in human cells and almost 75\% of these
↪  genes are expressed in the normal prostate. About 150 of these genes are more specifically expressed
↪  in the prostate with about 20 genes being highly prostate specific. The corresponding specific
↪  proteins are expressed in the glandular and secretory cells of the prostatic gland and have functions
↪  that are important for the characteristics of semen. Examples of some

Testosterone
The male generative glands also contain Sertoli cells, which require testosterone for spermatogenesis.
↪  Like most hormones, testosterone is supplied to target tissues in the blood where much of it is
↪  transported bound to a specific plasma protein, sex hormone-binding globulin (SHBG). In males,
↪  testosterone is synthesized primarily in Leydig cells. The number of Leydig cells in turn is
↪  regulated by luteinizing hormone (LH) and follicle-stimulating hormone (FSH). In addition, the amount
↪  of testosterone produced by existing Leydig cells is under the control of LH, which regulates the
↪  expression of 17-hydroxysteroid dehydrogenase. The amount of testosterone synthesized is regulated by
↪  the</tool_response><|im_end|>
<|im_start|>assistant
<think>From the provided responses, it is clear that the primary male hormone is testosterone.

Next, we need to find out what testosterone is derived from.</think>

<tool_call>
```

```
{"name": "wikipedia_search", "arguments": {"query": "what is testosterone synthesized from"}}
</tool_call><|im_end|>
<|im_start|>user
<tool_response>Testosterone
The male generative glands also contain Sertoli cells, which require testosterone for spermatogenesis.
↪  Like most hormones, testosterone is supplied to target tissues in the blood where much of it is
↪  transported bound to a specific plasma protein, sex hormone-binding globulin (SHBG). In males,
↪  testosterone is synthesized primarily in Leydig cells. The number of Leydig cells in turn is
↪  regulated by luteinizing hormone (LH) and follicle-stimulating hormone (FSH). In addition, the amount
↪  of testosterone produced by existing Leydig cells is under the control of LH, which regulates the
↪  expression of 17-hydroxysteroid dehydrogenase. The amount of testosterone synthesized is regulated by
↪  the

Testosterone
reticulum to yield a variety of C steroids. In addition, the 3-hydroxyl group is oxidized by
↪  3-hydroxysteroid dehydrogenase to produce androstenedione. In the final and rate limiting step, the
↪  C17 keto group androstenedione is reduced by 17-hydroxysteroid dehydrogenase to yield testosterone.
↪  The largest amounts of testosterone (>95\%) are produced by the testes in men, while the adrenal
↪  glands account for most of the remainder. Testosterone is also synthesized in far smaller total
↪  quantities in women by the adrenal glands, thecal cells of the ovaries, and, during pregnancy, by the
↪  placenta. In the testes, testosterone is produced by the Leydig cells.

Androgen
testosterone at maintaining normal prostate weight and duct lumen mass (this is a measure of epithelial
↪  cell function stimulation). Whereas DHT was equally potent as testosterone at preventing prostate
↪  cell death after castration. Androgens have also been found to signal through membrane androgen
↪  receptors, which are distinct from the classical nuclear androgen receptor. Androgens are synthesized
↪  from cholesterol and are produced primarily in the gonads (testicles and ovaries) and also in the
↪  adrenal glands. The testicles produce a much higher quantity than the ovaries. Conversion of
↪  testosterone to the more potent DHT occurs the prostate gland, liver, brain and skin.

Androgen
Androgen An androgen (from Greek ""andr-"", the stem of the word meaning ""man"") is any natural or
↪  synthetic steroid hormone which regulates the development and maintenance of male characteristics in
↪  vertebrates by binding to androgen receptors. This includes the embryological development of the
↪  primary male sex organs, and the development of male secondary sex characteristics at puberty.
↪  Androgens are synthesized in the testes, the ovaries, and the adrenal glands. Androgens increase in
↪  both boys and girls during puberty. The major androgen in males is testosterone. Dihydrotestosterone
↪  (DHT) and androstenedione are of equal importance in male development. DHT ""in utero"" causes

Testosterone
Testosterone Testosterone is the primary male sex hormone and an anabolic steroid. In male humans,
↪  testosterone plays a key role in the development of male reproductive tissues such as testes and
↪  prostate, as well as promoting secondary sexual characteristics such as increased muscle and bone
↪  mass, and the growth of body hair. In addition, testosterone is involved in health and well-being,
↪  and the prevention of osteoporosis. Insufficient levels of testosterone in men may lead to
↪  abnormalities including frailty and bone loss. Testosterone is a steroid from the androstane class
↪  containing a keto and hydroxyl groups at the three and seventeen</tool_response><|im_end|>
<|im_start|>assistant
<tool_call>
{"name": "wikipedia_search", "arguments": {"query": "what is the primary source of testosterone"}}
</tool_call><|im_end|>
<|im_start|>user
<tool_response>Testosterone
Testosterone Testosterone is the primary male sex hormone and an anabolic steroid. In male humans,
↪  testosterone plays a key role in the development of male reproductive tissues such as testes and
↪  prostate, as well as promoting secondary sexual characteristics such as increased muscle and bone
↪  mass, and the growth of body hair. In addition, testosterone is involved in health and well-being,
↪  and the prevention of osteoporosis. Insufficient levels of testosterone in men may lead to
↪  abnormalities including frailty and bone loss. Testosterone is a steroid from the androstane class
↪  containing a keto and hydroxyl groups at the three and seventeen

Testosterone
The male generative glands also contain Sertoli cells, which require testosterone for spermatogenesis.
↪  Like most hormones, testosterone is supplied to target tissues in the blood where much of it is
↪  transported bound to a specific plasma protein, sex hormone-binding globulin (SHBG). In males,
↪  testosterone is synthesized primarily in Leydig cells. The number of Leydig cells in turn is
↪  regulated by luteinizing hormone (LH) and follicle-stimulating hormone (FSH). In addition, the amount
↪  of testosterone produced by existing Leydig cells is under the control of LH, which regulates the
↪  expression of 17-hydroxysteroid dehydrogenase. The amount of testosterone synthesized is regulated by
↪  the

Testosterone
```

Testosterone is observed in most vertebrates. Testosterone and the classical nuclear androgen receptor
↪   first appeared in gnathostomes (jawed vertebrates). Agnathans (jawless vertebrates) such as lampreys
↪   do not produce testosterone but instead use androstenedione as a male sex hormone. Fish make a
↪   slightly different form called 11-ketotestosterone. Its counterpart in insects is ecdysone. The
↪   presence of these ubiquitous steroids in a wide range of animals suggest that sex hormones have an
↪   ancient evolutionary history. Testosterone Testosterone is the primary male sex hormone and an
↪   anabolic steroid. In male humans, testosterone plays a key role in the development of male
↪   reproductive tissues

Androgen
Androgens are metabolized mainly in the liver. A low testosterone level (hypogonadism) in men may be
↪   treated with testosterone administration. Prostate cancer may be treated by removing the major source
↪   of testosterone: testicle removal (orchiectomy); or agents which block androgens from accessing their
↪   receptor: antiandrogens. Androgen An androgen (from Greek ""andr-"", the stem of the word meaning
↪   ""man"") is any natural or synthetic steroid hormone which regulates the development and maintenance
↪   of male characteristics in vertebrates by binding to androgen receptors. This includes the
↪   embryological development of the primary male sex organs, and the development of male secondary sex
↪   characteristics

Testosterone
reticulum to yield a variety of C steroids. In addition, the 3-hydroxyl group is oxidized by
↪   3-hydroxysteroid dehydrogenase to produce androstenedione. In the final and rate limiting step, the
↪   C17 keto group androstenedione is reduced by 17-hydroxysteroid dehydrogenase to yield testosterone.
↪   The largest amounts of testosterone (>95\%) are produced by the testes in men, while the adrenal
↪   glands account for most of the remainder. Testosterone is also synthesized in far smaller total
↪   quantities in women by the adrenal glands, thecal cells of the ovaries, and, during pregnancy, by the
↪   placenta. In the testes, testosterone is produced by the Leydig cells.</tool_response><|im_end|>
<|im_start|>assistant
<think>From the provided responses, it is clear that testosterone is synthesized primarily in Leydig
↪   cells of the testes and it is derived from androstenedione or cholesterol.</think>

<tool_call>
{"name": "wikipedia_search", "arguments": {"query": "what is the primary source of androstenedione"}}
</tool_call><|im_end|>
<|im_start|>user
<tool_response>Androstenedione
Androstenedione is released into the blood by theca cells. Conversion of androstenedione to estrone
↪   requires the enzyme aromatase. Androstenedione is a substrate for estrogen production in granulosa
↪   cells which produce aromatase. Thus, theca cells and granulosa cells work together to form estrogens.
↪   Androstanedione is a 5-reduced metabolite of 4-androstenedione which serves as an intermediate in the
↪   biosynthesis of the androgen and neurosteroid androsterone. Levels are normally 30200 ng/dL (1.07.0
↪   nmol/L) in females and 40150 ng/dL (1.45.2 nmol/L) in males. Androstenedione has been shown to
↪   increase serum testosterone levels over an eight-hour period in men when taken as a single oral

Androstenedione
detection. Androstenedione is the common precursor of the androgen and estrogen sex hormones.
↪   Androstenedione can be biosynthesized in one of two ways. The primary pathway involves conversion of
↪   17-hydroxypregnenolone to DHEA by way of 17,20-lyase, with subsequent conversion of DHEA to
↪   androstenedione via the enzyme 3-hydroxysteroid dehydrogenase. The secondary pathway involves
↪   conversion of 17-hydroxyprogesterone, most often a precursor to cortisol, to androstenedione directly
↪   by way of 17,20-lyase. Thus, 17,20-lyase is required for the synthesis of androstenedione, whether
↪   immediately or one step removed. Androstenedione is produced in the adrenal glands and the gonads.
↪   The production of adrenal androstenedione is governed by

Androstenedione
adrenocorticotrophic hormone (ACTH), whereas production of gonadal androstenedione is under control by
↪   the gonadotropins. In premenopausal women, the adrenal glands and ovaries each produce about half of
↪   the total androstenedione (about 3 mg/day). After menopause, androstenedione production is about
↪   halved, due primarily to the reduction of the steroid secreted by the ovary. Nevertheless,
↪   androstenedione is the principal steroid produced by the postmenopausal ovary. Some androstenedione
↪   is also secreted into the plasma, and may be converted in peripheral tissues to testosterone and
↪   estrogens. Androstenedione is converted to either testosterone or estrone. Conversion of
↪   androstenedione to testosterone requires the enzyme 17-hydroxysteroid dehydrogenase.

Androstenedione
Androstenedione Androstenedione, or 4-androstenedione (abbreviated as A4 or -dione), also known as
↪   androst-4-ene-3,17-dione, is an endogenous weak androgen steroid hormone and intermediate in the
↪   biosynthesis of estrone and of testosterone from dehydroepiandrosterone (DHEA). It is closely related
↪   to androstenediol (androst-5-ene-3,17-diol). Androstenedione is a precursor of testosterone and other
↪   androgens, as well as of estrogens like estrone, in the body. In addition to functioning as an
↪   endogenous prohormone, androstenedione also has weak androgenic activity in its own right.
↪   Androstenedione has been found to possess some estrogenic activity, similarly to other DHEA
↪   metabolites. However, in contrast to androstenediol, its affinity for the

Androstenedione

```
     the brand name Metharmon-F and in combination with sex steroids (pregnenolone, testosterone, estrone,
↪    androstenediol) and thyroid hormone (desiccated thyroid), androstenedione is or has been marketed for
↪    medical use in Thailand. Androstenedione, also known as androst-4-ene-3,17-dione, is a naturally
↪    occurring androstane steroid and a 17-ketosteroid. It is closely related structurally to
↪    androstenediol (A5; androst-5-ene-3,17-diol), dehydroepiandrosterone (DHEA;
↪    androst-5-en-3-ol-17-one), and testosterone (androst-4-en-17-ol-3-one), as well as to
↪    5-androstanedione (5-androstane-3,17-dione) and estrone (estra-1,3,5(10)-triene-3-ol-17-one or
↪    19-norandrost-1,3,5(10)-triene-3-ol-17-one). Androstenedione was manufactured as a dietary
↪    supplement, often called ""andro"" or ""andros"" for short. Sports Illustrated credits Patrick Arnold
↪    for introducing androstenedione to the North American market. Androstenedione was
↪    legal</tool_response><|im_end|>
<|im_start|>assistant
<think>From the provided responses, it is clear that testosterone is synthesized from androstenedione and
↪    cholesterol.

So, the primary male hormone testosterone is derived from androstenedione and cholesterol.

Therefore, the primary male hormone derived from is cholesterol and androstenedione.</think>

<answer>
```json
{
  "rationale": "Testosterone, the primary male hormone, is synthesized from cholesterol and
  ↪    androstenedione."
,
  "answers": [
    "cholesterol",
    "Androstenedione"
  ]
}
</answer><|im_end|>
```

