# OpenReview forum: "A$^2$Search: Ambiguity-Aware Question Answering with Reinforcement Learning"
_ICLR.cc/2026/Conference — ICLR 2026 Poster_

### Official Review · Reviewer_FJjM · 2025-10-29

**Soundness:** 3
**Presentation:** 3
**Contribution:** 2
**Rating:** 4
**Confidence:** 4

**Summary:**

This paper proposes A²SEARCH—an end-to-end RL framework that requires no manual annotation. Its core contribution lies in designing an automated annotation pipeline that automatically identifies ambiguous questions in multi-hop question-answering datasets and generates alternative, valid answers through trajectory sampling, evidence filtering, and LLM-based validation. After RL training, A²SEARCH can automatically identify ambiguous questions, enabling the model to output all reasonable answers within a single inference trajectory, thereby improving the realism of the training signal and the reliability of the evaluation. Experimental results show that A²SEARCH-7B achieves SOTA  performance on eight open-domain QA benchmarks.

**Strengths:**

-  The paper clearly points out the pervasive mismatch between existing QA benchmarks (which assume a single gold answer) and real-world QA scenarios (where multiple valid answers exist). It is the first to propose a large-scale, annotation-free ambiguity solution specifically for the challenging multi-hop QA setting.
-  The proposed four-step automated pipeline (Sampling, Filtering, Verification, and Grouping) is a significant technical contribution. It effectively utilizes the capabilities of existing LLMs to automatically discover and verify alternative answers, demonstrating a viable and scalable approach to augmenting training data without costly manual annotation.
-  A2SEARCH-7B achieves SOTA results on multi-hop QA using only a single greedy decoding rollout. When compared to baseline models that typically require multiple sampled rollouts, this demonstrates the model's precise ambiguity-sensing capability and the significant efficiency of the proposed training paradigm.

**Weaknesses:**

Although the model performs exceptionally well on the AnsF1 metric, the source of this performance gain is insufficiently distinguished in the methodology.
It is unclear whether the improvement stems from genuinely solving ambiguities across different semantic levels or merely from increasing the recall of paraphrased variants of a single reference answer.

The use of a single-rollout comparison for A2SEARCH (@1) against multi-rollout estimation for baselines (@3) may potentially overstate its advantage relative to the baselines' true underlying capability.

**Questions:**

- The Attribution of Performance Improvements: Can a more detailed analysis be provided to better demonstrate the actual value of "ambiguity perception"? When A2SEARCH outputs multiple answers, what is the precision of its predicted answers? Does the model tend to over-generate to achieve higher recall, especially when the final answer is a single, unambiguous response? Is part of the model's superior performance merely due to being trained to output answers in a list format, which differs from baseline models that are optimized to provide a single best answer? The authors need to more clearly demonstrate that the benefits arise from the recognition of ambiguity, rather than just the output format of multiple answers.
- Additional experiments: Based on the previous point of inquiry, could you provide a comparative model that is trained on the original single definitive answer but is capable of outputting a list of multiple answers? This would allow for a comparison to demonstrate the contribution of the ambiguity-aware data itself. Additionally, it would be beneficial to constrain A2SEARCH to output only the most likely single answer and compare its performance on AnsF1@1 with that of baseline models. This would help to demonstrate whether the introduction of ambiguity information during training has enhanced the model's single-answer reasoning capability.
- Source of Baseline AnsF1/Recall@3 Data: What is the specific source for the baseline models' AnsF1/Recall@3 data reported in Table 1 and Table 2? Were these results taken directly from the original papers, or were they reproduced and estimated by the authors themselves using the specified sampling procedure?

---

> ### Author Response · Authors · 2025-11-21
> **Response [1/6]**
>
> ## Response to Weakness #2
> > [W2] The use of a single-rollout comparison for A2SEARCH (@1) against multi-rollout estimation for baselines (@3) may potentially overstate its advantage relative to the baselines' true underlying capability.
>
> We respectfully disagree with the concern that comparing **A$^2$Search (@1)** to the baselines' **@3** may overstate our model's advantage. Instead, it is the **fair and appropriate** evaluation setting. It both corrects the ambiguity-induced underestimation of baselines and highlights A$^2$Search's ability to resolve ambiguity without reliance on repeated sampling.
>
> **(1)** As we discuss in our response to Reviewer 6bRw-Q3, **even benchmarks annotated with a single reference answer** (e.g., TriviaQA, HotpotQA, 2Wiki) contain **7%-19.7%** semantically valid alternative answers. These alternatives are fully evidence-supported but missing from the dataset annotation.
>
> **(2)** Under the standard single-answer evaluation protocol, however, *only one* reference answer is accepted. Thus, even if a strong search model (e.g., ReSearch-32B) retrieves a correct—but unannotated—answer, it is marked as incorrect. This leads to a **systematic underestimation of baseline models' true capabilities when evaluated at @1**.
>
> **(3)** To compensate for this dataset ambiguity, we evaluate baselines using **multiple rollouts**. This allows them to generate several plausible answers that may recall the dataset's single annotated reference answer.
>
> **(4)** That said, multi-rollout sampling is inefficient and often produces redundant answers. A$^2$Search is designed specifically to overcome this limitation: it is the first **ambiguity-aware** search model trained end-to-end to produce **all valid answers within a single coherent rollout**. Therefore, reporting **A$^2$Search (@1)** accurately reflects the intended behavior of our model.

---

> ### Author Response · Authors · 2025-11-21
> **Response [2/6]**
>
> ## Response to Weakness #1 & Question #1(Part 1)
> > [W1] Although the model performs exceptionally well on the AnsF1 metric, the source of this performance gain is insufficiently distinguished in the methodology. It is unclear whether the improvement stems from genuinely solving ambiguities across different semantic levels or merely from increasing the recall of paraphrased variants of a single reference answer.
> >
> > [Q1(Part 1)] When A2SEARCH outputs multiple answers, what is the precision of its predicted answers?  Does the model tend to over-generate to achieve higher recall, especially when the final answer is a single, unambiguous response?
>
>
> We need clarify that **A$^2$Search does *not* suffer from over-generation** for two reasons: (1) our **training data construction** prevents redundant or unsupported answers, and (2) our **AnsF1 reward** explicitly penalizes over-generation. We also provide empirical evidence showing that the model predicts multiple answers *when ambiguity truly exists*.
>
> **(1) Training data construction prevents redundant answers**
>
> In our data construction pipeline (Verification + Grouping), every alternative answer is **evidence-supported**, and **semantically distinct** from all other answers.
> Thus the model never sees redundant or unsupported answers during training, eliminating the source of over-generation at the data level.
>
> **(2) AnsF1 reward explicitly penalizes over-generation**
>
> The AnsF1 reward is defined as:
>
> * $\mathrm{preds}$: total number of predicted answers
> * $\mathrm{hits}$: number of predictions that match reference answers
> * $\mathrm{refs}$: number of reference answers
>
> $
> \mathrm{Precision}=\frac{\mathrm{hits}}{\mathrm{preds}}, \quad
> \mathrm{Recall}=\frac{\mathrm{hits}}{\mathrm{refs}}, \quad
> \mathrm{AnsF1}=2\cdot\frac{\mathrm{Precision}\cdot\mathrm{Recall}}{\mathrm{Precision}+\mathrm{Recall}}.
> $
>
> If the model over-generates answers, **$\mathrm{preds}$ increases**, causing **precision to drop**, and therefore **AnsF1 to decrease**. Thus, the reward function *discourages* over-generation.
>
> **(3) Empirical evidence: A$^2$Search only outputs multiple answers when ambiguity exists**
>
> Here are the answer-count distribution of A$^2$Search-7B on four multi-hop QA benchmarks:
>
> | Answer Count | MuSiQue  | HotpotQA | 2Wiki    | Bamboogle |
> | ------------ | :-----------: | :-----------: | :-----------: | :-----------: |
> | 1            | 52.9%    | 72.2%    | 78.8%    | 86.4%     |
> | 2            | 25.4%    | 17.9%    | 17.0%    | 8.0%      |
> | 3            | 14.8%    | 4.5%     | 1.9%     | 2.4%      |
> | >3           | 6.9%     | 5.4%     | 2.4%     | 3.2%      |
> | **Avg.**     | **2.26** | **1.53** | **1.33** | **1.31**  |
>
> A$^2$Search outputs **one answer in the majority of cases**, especially on datasets with low ambiguity (e.g., Bamboogle, 2Wiki).
> Higher answer counts naturally occur only on datasets with substantial ambiguity (e.g., MuSiQue).
> This demonstrates that A$^2$Search predicts multiple answers **only when necessary**, not as a strategy to inflate recall.
>
> **(4) A$^2$Search achieves higher recall *and* higher precision than baselines**
>
> We report the macro-averaged Precision@k, Recall@k, and AnsF1@k across four multi-hop benchmarks (MuSiQue, HotpotQA, 2Wiki, and Bamboogle):
>
> | **Model**        | **AnsF1@1** | **Precision@1** | **Recall@1** | **AnsF1@3** | **Precision@3** | **Recall@3** |
> | ---------------- | :-----------: | :-----------: | :-----------: | :-----------: | :-----------: | :-----------: |
> | ReSearch-7B      | 39.3        | 39.3            | 39.3         | 42.1        | 36.6            | 49.5         |
> | Search-R1-7B     | 36.4        | 36.4            | 36.4         | 37.4        | 34.2            | 41.2         |
> | AFM-MHQ-7B       | 39.0        | 39.0            | 39.0         | 41.1        | 35.8            | 48.2         |
> | Search-R1-14B    | 43.1        | 43.1            | 43.1         | 44.0        | 40.7            | 47.9         |
> | Search-R1-32B    | 44.0        | 44.0            | 44.0         | 45.9        | 41.5            | **51.4**        |
> | ReSearch-32B     | 46.2        | 46.2            | 46.2         | 48.8        | 43.9            | **54.9**         |
> | $Sin$Search-7B     | 44.4        | 44.4            | 44.4         | 46.0        | 42.7            | 49.8         |
> | $Abg$Search-7B     | 31.1        | 28.0            | 34.9         | –           | –               | –            |
> | **A$^2$Search-7B**  | **48.4**    | **45.9**        | **51.2**     | –           | –               | –            |
>
> As shown above, A$^2$Search-7B achieves the highest precision, recall, and AnsF1 among comparable models at @1, and its precision surpasses all baselines even when they are given multiple rollouts (@3). This empirically demonstrates that A$^2$Search does not over-generate; instead, it improves both precision and recall simultaneously.

---

> ### Author Response · Authors · 2025-11-21
> **Response [3/6]**
>
> ## Response to Question #1(Part 2)
> > [Q1(Part 2)] Can a more detailed analysis be provided to better demonstrate the actual value of "ambiguity perception"?
> > Is part of the model's superior performance merely due to being trained to output answers in a list format, which differs from baseline models that are optimized to provide a single best answer?
> > The authors need to more clearly demonstrate that the benefits arise from the recognition of ambiguity, rather than just the output format of multiple answers.
>
> To directly address the reviewer's concerns, we conduct an additional analysis aimed at distinguishing two possibilities:
> (1) *A$^2$Search succeeds because it accurately detects semantic ambiguity in the question*, or
> (2) *A$^2$Search succeeds simply because it is trained to output lists, thus producing multiple answers regardless of ambiguity*.
>
> **Experiment Setup**
>
> We analyze the trajectories generated by A$^2$Search when answering questions from MuSiQue. Each trajectory includes the search actions, intermediate reasoning, and the final multi-answer prediction. To assess whether these multiple answers originate from true ambiguity, we use the **seven ambiguity types** defined in our response to Reviewer 6bRw–W2. For this experiment, we add an eighth category, **NOT_AMBIGUITY**, to capture cases where divergent answers arise from model errors rather than question ambiguity.
>
> We then construct a **LLM-based ambiguity classifier** (using OpenAI o3) that takes a trajectory as input and predicts which ambiguity type best explains the model's multiple answers. The judging protocol is described in our earlier response to Reviewer 6bRw–W2.
>
> We apply this classifier to A$^2$Search trajectories on MuSiQue where the model produced multiple answers. The resulting distribution across ambiguity types is shown below:
>
> | Ambiguity Type         | Definition                                                                                                                               | Frequency |
> | ---------------------- | ---------------------------------------------------------------------------------------------------------------------------------------- | --------: |
> | Under-Constrained      | The question lacks critical contextual constraints, allowing multiple distinct entities to satisfy the query                              | 43.9%       |
> | Granularity Ambiguity  | The required level of specificity (e.g., spatial, temporal, numeric) is not specified, yielding answers valid at different granularities  |  32.5%      |
> | Time Sensitivity       | The correct answer depends on an implicit temporal reference point                                                                        |   3.7%      |
> | Evidence Conflict      | Retrieved passages contain contradictory factual claims about the same entity                                                             | 3.6%          |
> | Multi-Item Response    | The question expects a single answer, but the true answer is a non-singleton set with no selection criteria provided                      | 5.3%        |
> | Open-Ended             | The question is inherently subjective or interpretive, permitting multiple qualitatively valid responses                                  |  6.8%       |
> | Alias Variance         | Different surface forms refer to the same underlying real-world entity                                                                    |  1.8%       |
> | **NOT_AMBIGUITY**      | Divergence in answers is not attributable to question design or ambiguity but instead results from model failures                         |  **2.5%**    |
>
>
> The results indicate that **97.5% of A$^2$Search's multi-answer predictions correspond to genuine semantic ambiguity**, covered by the seven defined categories. Only **2.5%** fall into **NOT_AMBIGUITY**.
> We manually examined these 2.5% cases. Nearly all stem from hallucinated facts or incorrect chaining of retrieved evidence, which are general LLM failure modes. Importantly, we did **not** observe cases where the model produced arbitrary or unsupported lists.
>
> **Conclusion**
>
> This additional analysis directly addresses the reviewer's questions. By examining A$^2$Search's internal trajectories and applying an ambiguity classifier, we show that its performance gains derive from **accurate detection and handling of genuine ambiguity**, rather than from list-format biases or inflated paraphrase recall.

---

> ### Author Response · Authors · 2025-11-21
> **Response [4/6]**
>
> ## Response to Question #2(Part 1)
> > [Q2(Part 1)] Based on the previous point of inquiry, could you provide a comparative model that is trained on the original single definitive answer but is capable of outputting a list of multiple answers? This would allow for a comparison to demonstrate the contribution of the ambiguity-aware data itself.
>
>
> We agree that explicitly demonstrating the contribution of the ambiguity-aware training data (multiple reference answers) is valuable. Below we provide both theoretical and experimental analyses.
>
> **Theoretically**, a model trained on single-answer data is expected to generate only a single answer. This is due to two reasons: (1) when all training examples contain exactly one reference answer, the model receives no signal to learn ambiguity resolution or when to output multiple answers; (2) the AnsF1 reward inherently penalizes over-generation: when only one reference exists, outputting additional answers increases recall while decreasing precision, leading to possibly lower overall AnsF1.
>
> Nevertheless, to directly address the reviewer’s concern, we trained an additional 7B model denoted **A$^2Sin$Search**, which uses exactly the same setups as **A$^2$Search**, but is trained exclusively on the single-reference-answer version of our dataset.
>
> First, we report the **average number of distinct answers** generated on various QA benchmarks:
>
> | Model              | MuSiQue | HotpotQA | 2Wiki | Bamboogle | PopQA | NQ    | TriviaQA | AmbigQA |
> |--------------------|---------|----------|-------|-----------|-------|-------|----------|---------|
> | A$^2$Search-7B        | 2.26    | 1.53     | 1.33  | 1.31      | 1.42  | 1.50  | 1.31     | 1.45    |
> | A$^2Sin$Search-7B     | 1.13    | 1.07     | 1.03  | 1.01      | 1.10  | 1.09  | 1.06     | 1.11    |
>
> As expected, A$^2Sin$Search almost always outputs a single answer, far fewer than A$^2$Search.
>
> Next, we report the **performance comparison** with AnsF1@1 for the single-answer baseline $Sin$Search and AnsF1@1 / Precision@1 / Recall@1 for models capable of multiple outputs:
>
> | Model                | HotpotQA     | 2Wiki       | MuSiQue     | Bamboogle    |
> | -------------------- | -----------  | ----------- | ----------- | ------------ |
> | **$Sin$Search-7B**   | 45.6         | 57.6        | 25.4        | 48.8         |
> | **A$^2$Search-7B**   | 49.5 / 48.6 / 52.1  | 62.3 / 61.5 / 64.4 | 30.1 / 28.5 / 34.8 | 51.7 / 50.9 / 53.6  |
> | **A$^2Sin$Search-7B**| 46.9 / 46.7 / 47.2  | 57.6 / 57.3 / 58.2 | 25.5 / 25.3 / 25.8 | 49.6 / 48.8 / 49.6  |
>
> | Model                | NQ          | TriviaQA    | PopQA       | AmbigQA     |
> | -------------------- | ----------- | ----------- | ----------- | ----------- |
> | **$Sin$Search-7B**   | 49.3        | 66.2        | 50.5        | 44.6 / 64.3 / 38.4 |
> | **A$^2$Search-7B**   | 51.4 / 50.1 / 54.7 | 67.8 / 67.1 / 69.6 | 52.5 / 51.1 / 55.6 | 48.1 / 65.2 / 43.2 |
> | **A$^2Sin$Search-7B**| 49.0 / 48.7 / 49.7 | 65.1 / 65.0 / 65.4 | 49.0 / 48.6 / 49.9 | 44.6 / 62.8 / 38.8 |
>
> **Key observations:**
> - A$^2Sin$Search performs very close to the original $Sin$Search baseline (trained with single-answer data and Exact Match reward) on nearly all benchmarks and remains *significantly below A$^2$Search*, confirming that single-answer training data alone is insufficient for effective ambiguity resolution.
> - On several datasets, A$^2Sin$Search achieves slightly higher recall than $Sin$Search, showing that the AnsF1 RL objective does encourage the model to recall correct answers, even under single-answer supervision.
>
> In summary, both (1) ambiguity-aware training data (multiple reference answers) and (2) RL with the AnsF1 reward are necessary and complementary: the former teaches the model when multiple answers are valid, while the latter provides the incentive to output them appropriately.

---

> ### Author Response · Authors · 2025-11-21
> **Response [5/6]**
>
> ## Response to Question #2(Part 2)
> > [Q2(Part 2)] Additionally, it would be beneficial to constrain A2SEARCH to output only the most likely single answer and compare its performance on AnsF1@1 with that of baseline models. This would help to demonstrate whether the introduction of ambiguity information during training has enhanced the model's single-answer reasoning capability.
>
> We believe there is a misunderstanding regarding the purpose of constraining A$^2$Search to produce a single "most likely" answer.
>
> **(1) Forcing A$^2$Search to output a single answer contradicts the purpose of ambiguity-aware QA.**
>
> A$^2$Search is explicitly designed to **detect whether a question is ambiguous and output the appropriate number of valid answers**. Many benchmark questions annotated with only one reference answer in fact admit multiple correct answers under different interpretations. In such cases, selecting one "best" answer is ill-defined: all valid answers are equally correct. Thus, evaluating A$^2$Search by forcing it into a single-answer format does not meaningfully measure its intended behavior and removes the very capability we aim to study.
>
> **(2) Moving beyond single-answer QA benchmarks: evaluating ambiguity resolution with more appropriate datasets**
>
> To more meaningfully evaluate the ambiguity-resolving capability of A$^2$Search, we consider **AmbigQA**, a dataset explicitly constructed to capture question ambiguity. AmbigQA is built on NaturalQuestions, where annotators identify inherently ambiguous questions and manually enumerate **all valid interpretations**, resulting in an average of **2.1 reference answers per question**. Thus, AmbigQA provides a controlled environment for testing multi-answer reasoning.
>
> However, AmbigQA is limited to **single-hop** questions. In our experiments, we trained an **$Abg$Search** baseline directly on AmbigQA's training set and found that it performs poorly on other **multi-hop open-domain** benchmarks. This confirms the need for our work: **multi-hop ambiguous QA requires more than simply training on a single-hop ambiguous dataset**.
>
> We report below the performance of A$^2$Search and several baselines on AmbigQA:
>
> | Model             | AnsF1@1 | Precision@1 | Recall@1 | AnsF1@3 | Precision@3 | Recall@3 |
> | ----------------- | :-------: | :-------: | :-------: | :-------: | :-------: | :-------: |
> | ReSearch-7B       | 40.8    | 48.1         | 35.4     | 45.3     | 48.1         | 42.8     |
> | Search-R1-7B      | 41.8    | 49.8         | 36.0     | 43.1     | 49.4         | 38.2     |
> | AFM-MHQ-7B        | 41.1    | 48.8         | 35.5     | 44.5     | 47.2         | 42.1     |
> | Search-R1-14B     | 43.8    | 52.1         | 37.8     | 45.2     | 51.1         | 40.5     |
> | Search-R1-32B     | 44.3    | 52.5         | 38.3     | 46.4     | 51.8         | 42.0     |
> | ReSearch-32B      | 44.1    | 52.2         | 38.2     | 47.8     | 52.2         | **44.1**     |
> | $Sin$Search-7B      | 44.6    | 53.2         | 38.4     | 45.1     | 52.0         | 39.8     |
> | $Abg$Search-7B      | 47.5    | 51.7         | **43.9**     | –        | –            | –        |
> | **A$^2$Search-7B**      | **48.1** | **54.3**    | **43.2** | –        | –            | –        |
>
>
> We highlight two key observations:
>
> 1. **A$^2$Search matches or exceeds the performance of $Abg$Search**, despite **not** being trained on AmbigQA at all.
>    This indicates that A$^2$Search learns **generalizable ambiguity resolution** rather than dataset-specific patterns.
>
> 2. Other baselines exhibit **very low Recall@1**, simply because they output only one answer for questions that genuinely have multiple reference answers.
>    When allowed to output more answers (@3), their **Precision drops** due to redundant candidates.
>    This again shows that forcing a single answer is structurally incompatible with ambiguous questions.
>
> **Conclusion**
>
> For inherently ambiguous questions, **constraining A$^2$Search to output only a single answer is neither meaningful nor an accurate assessment of its capability**. Evaluating ambiguity resolution requires datasets that explicitly contain multiple reference interpretations. AmbigQA provides one such benchmark, but covers only **single-hop** settings. A key direction of future work is to **develop multi-hop, open-domain ambiguous QA benchmarks**.

---

> ### Author Response · Authors · 2025-11-21
> **Response [6/6]**
>
> ## Response to Question #3
> > [Q3] Source of Baseline AnsF1/Recall@3 Data: What is the specific source for the baseline models' AnsF1/Recall@3 data reported in Table 1 and Table 2? Were these results taken directly from the original papers, or were they reproduced and estimated by the authors themselves using the specified sampling procedure?
>
> To clarify, all **AnsF1@3** and **Recall@3** results for baseline models reported in Table 1 and Table 2 are **reproduced by us**, not taken from the original papers. This is because **none of the prior works report multi-sample evaluation metrics** (e.g., AnsF1@3).
>
> (1) **Reproduction procedure**
>
> * For prompt-based baselines (**DirectGen, Naive-RAG, Iter-RetGen, IRCoT**), we use the official implementation from **FlashRAG** [1].
> * For **Search-R1**, **ReSearch**, and **AFM**, we use their official open-source code and the publicly released model weights to reproduce results.
>
> (2) **Fairness verification.**
>
> To ensure that our reproduced numbers faithfully reflect their original capabilities, we verify that:
>
> * All baselines' **@1 greedy-decoding performance** in our runs is **comparable to** reported in their original papers.
> * In many cases, our reproduced scores are **better** than the originally reported results.
>   For example, **ReSearch-32B** reports 44.9% EM on 2Wiki in the original paper, while our reproduction achieves **53.0%**, demonstrating that our reproduction setup is strong and not underestimating baselines.
>
> Despite these strengthened baseline results, **A$^2$Search still outperforms most baselines**, reinforcing the validity of our improvements.
>
> [1] [https://github.com/RUC-NLPIR/FlashRAG](https://github.com/RUC-NLPIR/FlashRAG)

---

> ### Author Response · Authors · 2025-11-26
> **Gentle Reminder to Reviewer FJjM**
>
> Dear Reviewer FJjM,
>
> Thank you again for the time and effort you have devoted to reviewing our work. After carefully reading your comments, we noticed that several concerns appeared to stem from **misunderstandings of our method and evaluation setup**. In our author response, we have thoroughly **clarified these points and addressed all issues** you raised:
>
> 1. We clarified why comparing A$^2$Search's @1 performance with the baselines' @3 is the fair and appropriate evaluation setting.
> 2. We provided both theoretical analysis and empirical evidence demonstrating that A$^2$Search does not suffer from over-generation.
> 3. We included additional analysis showing that A$^2$Search succeeds because it accurately identifies semantic ambiguity in the questions.
> 4. We conducted new ablation studies highlighting the necessity and effectiveness of our ambiguity-aware data construction.
> 5. We explained why forcing A$^2$Search to output only a single answer is neither meaningful nor a proper assessment of its capabilities.
>
> As the **discussion period is entering its final week**, we would greatly appreciate it if you could kindly revisit our paper in light of these clarifications and updated analyses. We hope our detailed response resolves the earlier misunderstandings and helps present our contribution more clearly and positively.
>
> If you have any further questions or would like additional information, we are always happy to discuss!
>
> Thank you very much for your time and consideration.

---

> > ### Comment · Reviewer_FJjM · 2025-11-27
> >
> > Regarding the phrase, "a comparative model that is trained on the original single definitive answer but is capable of outputting a list of multiple answers," my original intention was to **fix the number of answers outputted by the model** (for example, you could set it to three, four, or more). A reward should be given as long as one of the answers is correct. Please train and test under such a setting to demonstrate that "the benefit comes from the recognition of ambiguity, rather than just the output format of multiple answers.”
> >
> >
> > Additionally, I have one extra experimental suggestion, although I fully understand this may be *difficult to complete due to time or resource constraints*, which I would completely accept and it would not affect my final judgment. Could we attempt to train the model on a **multiple-answer-annotated dataset but constrain the model to output only a single answer**? Under this setup, the model would receive a reward if this single output answer matches any one of the correct answers in the dataset.

---

> > > ### Author Response · Authors · 2025-11-27
> > >
> > > We sincerely thank the reviewer for the timely response. We are happy to conduct the additional experiments you proposed. While these experiments will require some time to run, we can already provide clarification, theoretical analysis, and our expectations regarding the outcomes.
> > >
> > >
> > > **1. Training a model that outputs a fixed number of answers with a recall-style reward**
> > >
> > > Your first suggestion, training a comparative model that always outputs a fixed number of answers (e.g., 3 or 4), and rewarding it if *any* of its answers match the gold reference, is equivalent to replacing our **AnsF1** reward with an **AnsRecall** reward. Under such a reward, the optimal behavior is to output *as many answers as possible,* regardless of correctness. If the number of outputs is capped at, say, 3, the model will simply learn to always output exactly 3 answers for every question.
> > >
> > > We expect such a model to show **higher recall but degraded precision**, and importantly, it will **not** learn to *resolve* ambiguity. It may learn to *scatter guesses*.
> > >
> > > This setting differs from **A$^2$Search** in two essential dimensions:
> > >
> > > 1. **Reward formulation:** using AnsRecall rather than AnsF1 removes the incentive to produce correct, minimal, and evidence-grounded answer sets.
> > > 2. **Training data:** such a model is trained purely on single-answer data, whereas A$^2$Search uses ambiguity-aware training signals derived from multi-answer cases.
> > >
> > > With such variance, **we are not confident that this comparison would yield a meaningful conclusion**. Nevertheless, we will run this experiment and provide quantitative results and case studies to clearly demonstrate that *"merely outputting multiple answers" is insufficient for ambiguity resolution.*
> > >
> > >
> > > ### **2. Training on multiple-answer-annotated data but forcing single-answer output**
> > >
> > > We will also include the second proposed experiment: training the model on multi-answer–annotated data but constraining it to output only one answer, awarding reward if this single answer matches *any* of the gold answers.
> > >
> > > Under this setup, the model **still receives no learning signal for identifying ambiguity**. Therefore, we expect this model **not to acquire ambiguity-resolution capability**, despite having access to multi-answer labels during training.
> > >
> > > We will run this experiment as well and report the results.

---

> ### Comment · Reviewer_FJjM · 2025-11-27
> **Clarification on the motivation for the additional experiment**
>
> Thank you for your quick response and for agreeing to conduct the additional experiments.
>
> I would like to clarify my intention regarding the first proposed experiment. My point was not to advocate for replacing your AnsF1 reward with a "Recall-style" reward as a better method, but rather to isolate the source of the performance gain and verify the nature of the model's outputs.
>
> Specifically, I want to confirm whether the performance improvement stems from:
>
> - Genuine Ambiguity Resolution: The model outputting semantically distinct valid answers (e.g., "6 years", "3 years", "5 years" based on different interpretations/contexts), as claimed.
>
> - Paraphrasing/Format Gaming: The model merely outputting different surface forms or paraphrases of a single answer (e.g., "6", "6 years", "6y") to maximize the probability of hitting the reference.
>
> My concern is that a model might achieve a high score simply by "casting a wide net" with synonyms rather than actually resolving ambiguity.
>
> Looking forward to your experimental results.

---

> > ### Author Response · Authors · 2025-11-27
> >
> > Thank you very much for the clarification. We fully understand your concern about distinguishing *true ambiguity resolution* from potential *paraphrase-based gaming*, and we appreciate the opportunity to explain why this issue does not arise in our setting. We will still run the experiment you suggested for completeness, but we want to emphasize that **the phenomenon you worry about has already been extensively examined in our responses and empirical analyses**.
> >
> > ### **1. Multiple analyses already demonstrate that A$^2$Search does *not* inflate answers with rephrasings**
> >
> > Across our previous response: *Response to Weakness #1 & Question #1 (Part 1)* and *Response to Question #1 (Part 2)*, we have already presented three pieces of empirical evidence addressing exactly this concern:
> >
> > **(a) A$^2$Search tend not to output multiple answer unless necessary**
> >
> > In Table (3) "Empirical evidence: A$^2$Search only outputs multiple answers when ambiguity exists", we showed that A$^2$Search produces *single* answers most of the time on standard benchmarks:
> > * MuSiQue: 52.9%
> > * HotpotQA: 72.2%
> > * 2Wiki: 78.8%
> > * Bamboogle: 86.4%
> >
> > This strongly aligns with the *actual ambiguity rate* of these datasets (refer to our response to Reiviewer-6bRw-Q3).
> > If the model were gaming the metric by outputting surface-form variants (e.g., "6", "6 years", "6y"), we would instead observe *high multi-answer rates across all datasets*, which is not the case.
> >
> > **(b) High precision contradicts any "rephrase inflation" hypothesis**
> >
> > In Table (4) "A$^2$Search achieves higher recall *and* higher precision than baselines", we show that A$^2$Search maintains high precision.
> > If the model were producing multiple paraphrases of the same answer, precision would **collapse**. For example, if the gold answer is "6 years" but a model outputs: "6", "6 years", and "6y", then only one of them would match, giving precision = **1/3**, which is far lower than what we observe.
> >
> > The consistently high precision shows that our model is *not* generating redundant paraphrases.
> >
> > **(c) 97.5% of multi-answer cases correspond to *genuine* ambiguity (not paraphrase variation)**
> >
> > In *Response to Question #1 (Part 2)*, we report that:
> >
> > * **97.5%** of A$^2$Search's multi-answer outputs correspond to true semantic ambiguity.
> > * The ambiguity types are predominantly **Under-Constrained** and **Granularity Ambiguity**, *not* surface-form rephrasings.
> >
> > This analysis directly demonstrates that A$^2$Search produces multiple answers only when the question legitimately supports them.
> >
> >
> > ### **2. LLM-as-judge evaluation further rules out paraphrase-based gaming**
> >
> > We also draw the reviewer's attention to the LLM-as-judge evaluation results in Appendix Tables 11 and 12, and the judge prompt in Appendix G.1.
> >
> > The prompt explicitly instructs the judge:
> >
> > > *"The prediction does not have to be an exact string match to the ground truth; it only needs to be semantically equivalent."*
> >
> > Therefore:
> >
> > * If the predicted answer set is "6 / 6 years / 6y," (single-answer baseline output one of them while A$^2$Search output all)
> > * **all** of these would be judged correct for single-answer baselines,
> > * **but** A$^2$Search would be penalized for low precision (1/3), reducing its AnsF1.
> >
> > If A$^2$Search relied on paraphrase-spam, it would perform **worse** under LLM-as-judge.
> >
> > However, A$^2$Search's gains under LLM-as-judge evaluation are **consistent** with its gains under exact-match metrics.
> > This strongly indicates that A$^2$Search succeeds due to **genuine ambiguity identification**, not metric hacking via rephrasing.
> >
> > ### **3. Conclusion**
> >
> > We understand and respect your concern, but all empirical evidence: single-answer rate distributions, precision–recall behavior, ambiguity-type analysis, and LLM-as-judge evaluation consistently shows that A$^2$Search is *not* exploiting paraphrase generation to inflate performance.
> >
> > That said, *we will still perform the additional experiment you proposed*. We look forward to sharing those results with you shortly.

---

> > ### Author Response · Authors · 2025-12-02
> > **Results for the requested experiments**
> >
> > Following your request, we have trained *additional ablation models* that isolate the effects of (i) ambiguity-aware training data, (ii) the AnsF1 reward, and (iii) the ability to output multiple answers.
> >
> > ### **1. Ablation Models**
> >
> > **(1) $Recall$Search-7B**
> >
> > * Reward: EM-Recall (reward = 1 if *any* generated answer matches the gold answer)
> > * Training Data: Original single-answer data
> > * Output: Up to *3* answers
> >
> > **(2) $Sin^*$Search-7B**
> >
> > * Reward: EM
> > * Training Data: Ambiguity-aware multi-answer data
> > * Output: *One* answer
> >
> > **(3) A$^2Sin$Search-7B**
> > * Reward: AnsF1
> > * Training Data: Original single-answer data
> > * Output: Arbitrary answers
> >
> > ### **2. Answer Count Distribution**
> >
> > | Model               | MuSiQue | HotpotQA | 2Wiki | Bamboogle | PopQA | NQ   | TriviaQA | AmbigQA |
> > | ------------------- | ------- | -------- | ----- | --------- | ----- | ---- | -------- | ------- |
> > | $Sin^*$Search-7B   | 1.00    | 1.00     | 1.00  | 1.00      | 1.00  | 1.00 | 1.00     | 1.00    |
> > | A$^2Sin$Search-7B  | 1.13    | 1.07     | 1.03  | 1.01      | 1.10  | 1.09 | 1.06     | 1.11    |
> > | $Recall$Search-7B | 3.00    | 3.00     | 3.00  | 3.00      | 3.00  | 3.00 | 3.00     | 3.00    |
> > | **A$^2$Search-7B**     | 2.26    | 1.53     | 1.33  | 1.31      | 1.42  | 1.50 | 1.31     | 1.45    |
> >
> > **Key findings:**
> >
> > * A$^2Sin$Search-7B still outputs *one* answer almost always.
> > * $Recall$Search-7B always outputs exactly 3 answers.
> > * Only A$^2$Search-7B adapts answer count to dataset ambiguity.
> >
> > ### **3. Performance on 8 Benchmarks**
> >
> > | Model           | HotpotQA               | 2Wiki                  | MuSiQue                | Bamboogle              |
> > | --------------- | ---------------------- | ---------------------- | ---------------------- | ---------------------- |
> > | $Sin^*$Search-7B   | 47.2                   | 58.4                   | 26.5                   | 50.4                   |
> > | $Recall$Search-7B | 29.6 / 20.6 / 55.6     | 39.3 / 28.6 / 68.2     | 18.9 / 13.5 / 34.3     | 32.2 / 22.1 / 61.6     |
> > | A$^2Sin$Search-7B  | 46.9 / 46.7 / 47.2  | 57.6 / 57.3 / 58.2 | 25.5 / 25.3 / 25.8 | 49.6 / 48.8 / 49.6  |
> > | **A$^2$Search-7B** | 49.5 / 48.6 / 52.1  | 62.3 / 61.5 / 64.4 | 30.1 / 28.5 / 34.8 | 51.7 / 50.9 / 53.6  |
> >
> >
> > | Model           | NQ                     | TriviaQA               | PopQA                  | AmbigQA                |
> > | --------------- | ---------------------- | ---------------------- | ---------------------- | ---------------------- |
> > | $Sin^*$Search-7B   | 49.8                   | 67.6                   | 50.5                   | 45.0 / 64.8 / 38.8     |
> > | $Recall$Search-7B | 35.1 / 25.1 / 63.2     | 49.5 / 39.2 / 75.0     | 37.3 / 28.2 / 61.1     | 38.3 / 36.4 / 49.8     |
> > | A$^2Sin$Search-7B  | 49.0 / 48.7 / 49.7 | 65.1 / 65.0 / 65.4 | 49.0 / 48.6 / 49.9 | 44.6 / 62.8 / 38.8 |
> > | **A$^2$Search-7B** | 51.4 / 50.1 / 54.7 | 67.8 / 67.1 / 69.6 | 52.5 / 51.1 / 55.6 | 48.1 / 65.2 / 43.2 |
> >
> > **Key findings:**
> >
> > * $Sin^*$Search-7B underperforms because all these datasets contain ambiguity and single-answer output is insufficient.
> > * $Recall$Search-7B achieves very high recall, but precision collapses to <50% of A$^2$Search.
> > * A$^2Sin$Search-7B performs similarly to single-answer baselines and shows no gains in recall.
> > * Only A$^2$Search-7B improves *both* precision and recall simultaneously across all datasets.
> >
> >
> > ### **4. $Recall$Search-7B's recall is artificially high because of reward hacking**
> >
> > Using the ambiguity-type analysis from *Response to Question #1 (Part 2)*, we classify $Recall$Search-7B's multi-answer output sources:
> >
> > | Ambiguity Type                                      | Frequency |
> > | --------------------------------------------------- | --------- |
> > | Under-Constrained                                   | 14%       |
> > | Granularity Ambiguity                               | 19%       |
> > | Time Sensitivity                                    | 5%        |
> > | Evidence Conflict                                   | 2%        |
> > | Multi-Item Response                                 | 4%        |
> > | Open-Ended                                          | 0%        |
> > | **Alias Variance** (surface-form rephrasing)        | **35%**   |
> > | **NOT_AMBIGUITY** (model failures / random guesses) | **21%**   |
> >
> > The model is increasing recall primarily by:
> >
> > * emitting paraphrases (e.g., "Orleans County", "Orleans", "Orleans County, New York"),
> > * random guessing to cover more possibilities (e.g. "rancher", "farmer", "businessman"),
> >
> > ### **5. Conclusion**
> >
> > These ablation results collectively demonstrate that **every component of A$^2$Search’s design is essential**. Moreover, the findings clearly show that its gains do **not** arise from superficial output‐format effects or metric artifacts, but from **genuine improvements in ambiguity resolution**.

---

### Official Review · Reviewer_wyg8 · 2025-10-29

**Soundness:** 3
**Presentation:** 3
**Contribution:** 2
**Rating:** 6
**Confidence:** 3

**Summary:**

The paper tackles ambiguity in open-domain QA (i.e. the cases where a question has multiple valid answers) by proposing $A^2-SEARCH$, an annotation-free RL framework that automatically mines alternative answers via an evidence-based pipeline and trains an LLM with an answer-level F1 (AnsF1) reward that balances precision and recall in multi-answer settings. Experiments on eight benchmarks show state-of-the-art results with single-rollout decoding.

**Strengths:**

- The paper proposes a method that addresses a significant problem in QA.
- The authors present convincing empirical evidence supporting their claims, using several baseline models of various sizes and several datasets to check the performance of their model.

**Weaknesses:**

- The paper does not address unanswerable queries - cases where retrieval fails and the model should abstain (e.g., say “I don’t know”). This omission is a substantial gap that limits real-world applicability of the framework.
- At "Step 1: Sampling" (lines 248-258) the pipeline generates millions of trajectories, which may be computationally infeasible for many users; clearer reporting of the time cost of these experiments would be valuable.
- If I understand correctly, the authors applied their training recipe exclusively to Qwen-family models to produce their $A^2-SEARCH$ model. I.e. they don't provide evidence for cross-family generality of this training approach (see "Questions" section).
- Verification phase fully relies on proprietary models (Claude 3.5/3.7, OpenAI o3/o4-mini). Unpredictable availability of proprietary models and model drift may affect reproducibility and future re-runs; more results with open verifiers (or human audits at scale) would help.
- Some parts of the paper are written in unclear way (see "Questions" section).

**Questions:**

- How sensitive are results to the choice/mix of verifiers? Could an all-open pipeline (e.g., Llama- or Qwen-based judge + open retriever) approach the same agreement/coverage?
- What "△", "□", "○" symbols mean on the Figure 2?
- What's going on in the "Step 2" on the Figure 2?
- Do I understand correctly that you use Qwen-2.5-Intrusct 3B/7B specifically as a base models for your $A^2-SEARCH$ 3B and 7B correspondingly in the Tables 1 and 2?
- Did you try other models, NOT from Qwen family, as a base models for your $A^2-SEARCH$? What is the difference in the performance for other models?

---

> ### Author Response · Authors · 2025-11-21
> **Response [1/3]**
>
> ## Response to Weakness #1
> > [W1] The paper does not address unanswerable queries - cases where retrieval fails and the model should abstain (e.g., say “I don’t know”). This omission is a substantial gap that limits real-world applicability of the framework.
>
> We appreciate the reviewer bringing this point to our attention. Our work specifically targets a different and complementary challenge: **ambiguity in answerable multi-hop open-domain QA**, where multiple distinct answers can all be correct and evidence-supported. Accordingly:
>
> * All benchmarks used in this paper are constructed such that every question has at least one correct answer
> * The evaluation protocols assumed by prior work similarly rely on this answerability assumption
>
> Therefore, handling unanswerable queries **falls outside the scope of this work** and is not necessary for a fair comparison with our baseline models, which adopt the same setup.
>
> Nonetheless, we agree that abstaining from answering unanswerable queries is an important direction for improving real-world robustness. We will include a discussion in the revised manuscript to clarify scope and outline how our framework could be extended to cover this case in future work.
>
> ## Response to Weakness #2
> > [W2] At "Step 1: Sampling" (lines 248-258) the pipeline generates millions of trajectories, which may be computationally infeasible for many users; clearer reporting of the time cost of these experiments would be valuable.
>
> The reported 3.99M trajectories result from our design choice of maximizing sampling diversity to uncover latent ambiguity: we used five search models (ReSearch-7B/32B and Search-R1-7B/14B/32B), each generating 16 sampled trajectories per question over **49,938 training questions**. This is a **one-time data construction** process, not required during model deployment.
>
> To address concerns about computational feasibility, we report the actual costs below:
> Using a single node equipped with 4× H100 GPUs, vLLM-accelerated inference [1], and a locally served Wikipedia search index (same setup as Search-R1 [2]), the total runtime for evaluating all five models across the reported benchmarks was:
>
> | Model         | Avg. Tool Calls | Duration |
> | ------------- | :--------------: | -------: |
> | ReSearch-32B  |            3.52 |   38.6 h |
> | ReSearch-7B   |            3.00 |   16.0 h |
> | Search-R1-32B |            1.82 |    7.3 h |
> | Search-R1-14B |            1.68 |    5.2 h |
> | Search-R1-7B  |            2.80 |    7.8 h |
> | **Total**     |              —  | **74.9 h** |
>
> ("Avg. Tool Calls" denotes the average number of search-tool invocations per sampled trajectory)
>
> This computational cost is **orders of magnitude more scalable and cost-effective than manual annotation**. Moreover, the reported numbers are conservative: the pipeline is highly parallelizable, and both inference throughput and search latency can be further optimized substantially.
>
> We will add these details to the revised manuscript for greater transparency and to emphasize that our data construction pipeline is practical, highly scalable, and easily reproducible in real-world settings.
>
> [1] vLLM: [https://github.com/vllm-project/vllm](https://github.com/vllm-project/vllm)
>
> [2] Search-R1: https:github.com/PeterGriffinJin/Search-R1/blob/main/retrieval_launch.sh

---

> ### Author Response · Authors · 2025-11-21
> **Response [2/3]**
>
> ## Response to Weakness #3 & Question #5
> > [W3] If I understand correctly, the authors applied their training recipe exclusively to Qwen-family models to produce their A$^2$Search$ model. I.e. they don't provide evidence for cross-family generality of this training approach.
> >
> > [Q5] Did you try other models, NOT from Qwen family, as a base models for your A$^2$Search$? What is the difference in the performance for other models?
>
> We thank the reviewer for raising this question. While our main experiments focus on Qwen-family models, our method is not restricted to this backbone, and we provide additional evidence below to demonstrate its generality.
>
> **(1) Fair comparison within the Qwen family**
>
> All baselines in our main experiments use the Qwen2.5 family as the backbone, ensuring a controlled and fair comparison. Under this setting, performance gain comes from improved ambiguity resolution rather than differences in backbone capacity.
>
> **(2) Our RL method is backbone-agnostic**
>
> Our RL training framework builds on **GRPO with outcome-based rewards**, which does not rely on any Qwen-specific designs.
>
> **(3) Cross-family evaluation using Llama 3.2 Instruct (3B)**
>
> To verify generality beyond Qwen, we trained **A$^2$Search-LMA-3B** using *Llama 3.2 Instruct 3B* with exactly the same training setup as A$^2$Search on Qwen. We also trained a matching baseline (**$Sin$Search-LMA-3B**) using *single-answer* data and an *Exact Match reward*.
>
> In the tables below, values before the slash denote **AnsF1** and values after denote **Recall**. Note that **A$^2$Search uses a single greedy rollout**, whereas **$Sin$Search samples 6 rollouts** to compute AnsF1@3 and Recall@3. The benchmark abbreviations are: MSQ (MuSiQue), HPQ (HotpotQA), BBG (Bamboogle), PQ (PopQA), TQ (TriviaQA), and AQ (AmbigQA)
>
> | Model                | HPQ @1 | HPQ @3 | 2Wiki @1    | 2Wiki @3    | MSQ @1  | MSQ @3  | BBG @1 | BBG @3 | Avg @1 | Avg @3 |
> | -------------------- | ----------- | ----------- | ----------- | ----------- | ----------- | ----------- | ------------ | ------------ | ------------ | ------------ |
> | **$Sin$Search-3B**     | 37.9        | 41.1 / 47.1 | 47.3        | 50.8 / 58.2 | 19.5        | 20.5 / 25.6 | 38.4         | 38.2 / 41.8  | 35.8         | 37.7 / 43.2  |
> | **$Sin$Search-7B**     | 45.6        | 46.9 / 50.3 | 57.6        | 59.5 / 64.1 | 25.4        | 27.0 / 30.9 | 48.8         | 50.6 / 53.8  | 44.4         | 46.0 / 49.8  |
> | **$Sin$Search-LMA-3B** | 36.2        | 38.7 / 45.7 | 48.2        | 49.9 / 55.1 | 16.7        | 18.7 / 23.8 | 39.2         | 43.4 / 53.1  | 35.1         | 37.7 / 44.4  |
> | **A$^2$Search-3B**     | 42.8 / 44.4 | - | 56.2 / 58.9 | - | 24.2 / 25.9 | - | 49.3 / 50.4  |  - | 43.1 / 44.9  |  - |
> | **A$^2$Search-7B**     | 49.5 / 52.1 | - | 62.3 / 64.4 | - | 30.1 / 34.8 | - | 51.7 / 53.6  | -  | 48.4 / 51.2  | -  |
> | **A$^2$Search-LMA-3B** | 42.2 / 43.6 | - | 53.4 / 55.3 | - | 22.6 / 24.8 | - | 51.1 / 52.0  | -  | 42.3 / 43.9  | -  |
>
> | Model                | NQ @1       | NQ @3       | TQ @1 | TQ @3 | PQ @1    | PQ @3    | AQ @1  | AQ @3  | Avg @1 | Avg @3 |
> | -------------------- | ----------- | ----------- | ----------- | ----------- | ----------- | ----------- | ----------- | ----------- | ------------ | ------------ |
> | **$Sin$Search-3B**     | 40.9        | 43.3 / 48.2 | 58.0        | 59.9 / 64.9 | 43.7        | 45.0 / 49.3 | 38.2 / 32.8 | 41.2 / 38.6 | 45.2 / 43.9  | 47.4 / 50.2  |
> | **$Sin$Search-7B**     | 49.3        | 49.8 / 51.3 | 66.2        | 67.0 / 69.2 | 50.5        | 51.4 / 53.5 | 44.6 / 38.4 | 45.1 / 39.8 | 52.6 / 51.1  | 53.3 / 53.5  |
> | **$Sin$Search-LMA-3B** | 41.8        | 43.2 / 49.4 | 55.9        | 58.6 / 65.7 | 46.2        | 48.5 / 51.6 | 38.3 / 33.0 | 42.2 / 39.9 | 45.6 / 44.2  | 48.1 / 51.7  |
> | **A$^2$Search-3B**     | 47.3 / 49.7 | - | 60.9 / 62.5 | - | 48.2 / 50.5 | - | 43.1 / 38.2 | - | 49.9 / 50.2  | -  |
> | **A$^2$Search-7B**     | 51.4 / 54.7 | - | 67.8 / 69.6 | - | 52.5 / 55.6 | - | 48.1 / 43.2 | - | 55.0 / 55.8  | -  |
> | **A$^2$Search-LMA-3B** | 46.8 / 49.6 | - | 62.0 / 63.2 | - | 47.7 / 50.7 | - | 43.4 / 38.7 | - | 50.0 / 50.6  | -  |
>
>
> **(4) Key observations**
>
> * The *Llama-based models achieve performance comparable to Qwen models of the same size*, indicating that our framework transfers well across model families.
> * *A$^2$Search-LMA-3B attains high recall with only one rollout*, often approaching or exceeding *$Sin$Search-LMA-3B Recall@3*.
> * These results **confirm that our ambiguity-aware RL framework is backbone-agnostic and broadly applicable**.
>
> We will included these details in the revised manuscript.

---

> ### Author Response · Authors · 2025-11-21
> **Response [3/3]**
>
> ## Response to Weakness #4 & Question #1
> > [W4] Verification phase fully relies on proprietary models (Claude 3.5/3.7, OpenAI o3/o4-mini). Unpredictable availability of proprietary models and model drift may affect reproducibility and future re-runs; more results with open verifiers (or human audits at scale) would help.
> >
> > [Q1] How sensitive are results to the choice/mix of verifiers? Could an all-open pipeline (e.g., Llama- or Qwen-based judge + open retriever) approach the same agreement/coverage?
>
> While we utilize strong proprietary models in our main experiments to maximize verification reliability, our framework **does not depend on proprietary verifiers**, and we provide further evidence that **an all-open pipeline can achieve comparable reliability**.
>
> We conducted an additional experiment using *four moderate-sized open-weight LLMs* that fit on a single H100 GPU: GPT-OSS-20B, GPT-OSS-120B, Qwen3-30A3B, and QwQ-32B.
>
> We sampled *25,000* trajectories from our data construction process. These trajectories may contain alternative valid answers to the same question. Each verifier produces one of three labels for a trajectory: "supported", "partially supported", or "not supported". In our evaluation, we treat *only the "supported" label as a positive judgment*. Using the original *proprietary* verifier ensemble (4 models, voting threshold $\geq$3/4), *4,651* trajectories were approved as "supported." We then applied the *open-weight verifier ensemble* to the same data and compared its decisions against the proprietary group. We report:
> * *recall*: proportion of **proprietary-approved** trajectories that are also approved by the **open-weight** ensemble.
> * *Precision*: proportion of **open-approved** trajectories that are also approved by the **proprietary** ensemble.
>
> We vary the voting threshold. In addition, for each threshold, we randomly sample 100 trajectories that were positively voted by the open-weight ensemble and manually check whether the answer is indeed fully supported by the retrieved evidence. The fraction of such cases is reported as *Human Agreement*.
>
> | Threshold | Precision | Recall   | Human Agreement     |
> | :------:  | --------- | -------- | :---------------:   |
> | $\geq$ 1/4    | 0.30      | 0.99     | 0.56            |
> | $\geq$ 2/4    | 0.50      | 0.95     | 0.78            |
> | $\geq$ 3/4    | **0.80**  | **0.85** | **0.96**        |
> | $\geq$ 4/4    | 0.87      | 0.67     | 0.98            |
>
> **Key findings.**
>
> * With a *$\geq$ 3/4 voting threshold*, open-weight verifiers achieve *80% precision* and *85% recall* with respect to the proprietary ensemble. Higher thresholds further increase precision, at the cost of filtering more trajectories.
> * Human evaluation confirms that a *$\geq$ 3/4 voting threshold* can yield high-quality labels (96% agreement).
> * We find that the primary source of disagreement between the open-weight and proprietary verifier groups lies in how they classify borderline cases between "supported" and "partially supported" answers. In fact, some "partially supported" cases are already fully supported. This conservative tendency effectively increases the precision of accepted trajectories, at the expense of discarding some borderline-but-valid cases.
>
> Overall, these results show that **a fully open verifier stack can reliably approximate the judgments of proprietary models**, enabling reproducible pipelines.
>
> ## Response to Weakness #5 & Question #2,3,4
> > [W5] Some parts of the paper are written in unclear way (see "Questions" section).
> >
> > [Q2] What "△", "□", "○" symbols mean on the Figure 2?
> >
> > [Q3] What's going on in the "Step 2" on the Figure 2?
> >
> > [Q4] Do I understand correctly that you use Qwen-2.5-Intrusct 3B/7B specifically as a base models for your  3B and 7B correspondingly in the Tables 1 and 2?
>
> **[Q2] Meaning of "△", "□", and "○" in Figure 2**
>
> These symbols denote *different semantic groups* of alternative answers.
> In the *Grouping* step, we cluster all model-generated answers according to their semantic similarity. The abstract symbols (△, □, ○) in Figure 2 are simply visual markers indicating *distinct semantic clusters*.
>
>
> **[Q3] What happens in "Step 2" of Figure 2**
>
> Step 2 performs *rule-based filtering* of trajectories. The figure illustrates two representative cases:
>
> 1. *Trajectories whose answers are semantically equivalent to the reference answer (ans\*)*.  These provide no new alternative answers and are therefore removed.
>
> 2. *Trajectories whose answers all *differ* from (ans\*)*.  If a model fails to generate the reference answer in *any* rollout for a given question, we assume the model is not capable of solving that question; thus, *all* its trajectories for that question are discarded.
>
> **[Q4] Base models used in Tables 1 and 2**
>
> Yes, your understanding is correct. The *A$^2$Search-3B* and *A$^2$Search-7B* models reported in Tables 1 and 2 are trained from *Qwen2.5 Instruct 3B* and *Qwen2.5 Instruct 7B*, respectively.

---

> ### Author Response · Authors · 2025-11-26
> **Gentle Reminder to Reviewer wyg8**
>
> Dear Reviewer wyg8,
>
> Thank you very much for your thoughtful and constructive feedback on our paper. We sincerely appreciate the time and effort you have put into the review. In our author response, we have carefully addressed the concerns you raised:
>
> 1. We clarified that our work primarily focuses on tackling *ambiguity* in **answerable** multi-hop open-domain QA. Handling unanswerable queries is outside the scope of our current objectives and does not influence the core novelty or findings of this paper.
>
> 2. We provided real statistics on the inference cost of trajectory sampling in our framework, demonstrating that it is quite efficient. This computational cost remains orders of magnitude more scalable and cost-effective than manual annotation.
>
> 3. We added further experiments showing that our ambiguity-aware RL framework is **backbone-agnostic** (applicable to both Qwen and Llama), confirming its broader applicability.
>
> 4. We clarified that our approach does not rely on proprietary verifiers. We also included additional results showing that a fully open-source pipeline can achieve comparable reliability.
>
> We hope our additional analyses help strengthen the contribution and address your concerns satisfactorily. As **the discussion phase is approaching its final week**, if any further questions arise, we would be very happy to discuss them.
>
> Thank you again for your constructive comments and valuable time.

---

### Official Review · Reviewer_6bRw · 2025-11-01

**Soundness:** 3
**Presentation:** 3
**Contribution:** 3
**Rating:** 6
**Confidence:** 3

**Summary:**

The paper addresses the challenge of ambiguity in open-domain QA, noting that standard benchmarks often rely on a single reference answer, overlooking the reality that many questions admit multiple equally valid responses. To counter this, the paper proposes A^2 SEARCH, an annotation-free, end-to-end RL framework designed to recognize and handle ambiguity. This work proposes automated data generation pipeline that detects ambiguous questions and collects alternative answers via trajectory sampling and evidence verification. The subsequent RL training uses GRPO and an answer-level F1 reward, a metric tailored to accommodate multiple correct answers by rewarding coverage while penalizing over-generation. Eight open-domain QA benchmarks demonstrates that the approach achieves new state-of-the-art results, confirming that embracing ambiguity is essential for developing more reliable QA systems that can learn to sense ambiguity and retrieve multiple valid answers.

**Strengths:**

- The motivation for this work is well explained, and the paper is generally well written. The paper effectively highlights the significant challenge posed by ambiguity in open-domain QA. The recognition that current RL pipelines deliver misleading signals by rewarding only the reference answer and penalizing valid alternatives provides a strong foundation for the proposed framework.

- The proposed method is its fully automated pipeline for identifying ambiguous questions and gathering alternative answers via trajectory sampling and evidence verification. This annotation-free approach is critical, especially since scaling costly manual annotation to complex multi-hop datasets like HotpotQA and MuSiQue is difficult. This automation makes the approach highly practical and scalable.

- The paper includes rich and detailed experiments, encompassing a variety of ablation studies and case analyses.

- Models trained with the proposed method show high efficiency, achieving strong recall levels with fewer average tool calls compared to baselines, demonstrating more effective utilization of reasoning steps.

**Weaknesses:**

- While the overall framework is effective, the novelty of the core learning method and reward design is moderate. The reward structure employs an outcome-only design and incorporates Answer-level F1 (AnsF1), which is derived from standard precision and recall metrics but tailored for multiple answers. The novelty lies primarily in the application of these existing mechanisms to the specific problem of ambiguity resolution, coupled with the sophisticated automated data pipeline, rather than new architectural or objective function designs.

- While the paper confirms that the generated alternative answers are semantically distinct from the reference answer and evidence-supported, a more detailed breakdown of why they constitute genuine ambiguity is helpful. A statistical analysis quantifying the frequency of these different types of ambiguity (e.g., linguistic vs. entity vs. scope ambiguity) across the derived dataset would enrich the findings of the paper.

**Questions:**

- Why are K LLM verifiers necessary? Just for the ensemble?
- L174 - what is the meaning of a n_hat s?
- How does performance increase even on non-ambiguity datasets? Are ambiguous answers substantially included in datasets like HotpotQA?

---

> ### Author Response · Authors · 2025-11-21
> **Response [1/3]**
>
> ## Response to Weakness #1
> > [W1] While the overall framework is effective, the novelty of the core learning method and reward design is moderate. The reward structure employs an outcome-only design and incorporates Answer-level F1 (AnsF1), which is derived from standard precision and recall metrics but tailored for multiple answers. The novelty lies primarily in the application of these existing mechanisms to the specific problem of ambiguity resolution, coupled with the sophisticated automated data pipeline, rather than new architectural or objective function designs.
>
> We appreciate the reviewer's comments and respectfully emphasize that our novelty does not lie in modifying architectures or inventing new losses, but in **reformulating how multi-hop QA models learn from inherently ambiguous data** and **providing a practical foundation that scales ambiguity-aware QA training to realistic, open-domain environments**. We advance the field by
>
> (1) **Breaking the data scalability bottleneck.**
>    We show that ambiguity-resolution in multi-hop QA is fundamentally a data scalability problem. To overcome this bottleneck, we introduce a fully automated pipeline based on trajectory sampling and evidence verification that identifies ambiguous questions and valid alternative answers across diverse multi-hop benchmarks. This enables ambiguity-aware RL training at scale and further demonstrates robust cross-dataset generalization to untrained benchmarks, such as AmbigQA.
>
> (2) **Proposing a general RL framework effective across diverse models and benchmark.**
>    Rather than relying on task-specific architectural changes or bespoke objectives, we show that a standard outcome-based RL setup (GRPO with an answer-level F1 reward) is sufficient. Once ambiguity-aware supervision is properly constructed, the model naturally learns to recognize ambiguity and output multiple valid answers. We further validate the approach across different model regimes, spanning both base and instruction-tuned variants, and both the Qwen and LLaMA model families (see our response to Reviewer wyg8 W3), demonstrating that this simplicity yields robust effectiveness and broad applicability.
>
> ## Response to Weakness #2
> > [W2] While the paper confirms that the generated alternative answers are semantically distinct from the reference answer and evidence-supported, a more detailed breakdown of why they constitute genuine ambiguity is helpful. A statistical analysis quantifying the frequency of these different types of ambiguity (e.g., linguistic vs. entity vs. scope ambiguity) across the derived dataset would enrich the findings of the paper.
>
> Thank you for the suggestion. We agree that a more detailed and quantitative breakdown of the sources of ambiguity is essentia. In response, we conducted an additional in-depth analysis detailed below.
>
> We first manually derived a taxonomy of seven major ambiguity types that recur in our constructed training data. We then employed a strong LLM judge (OpenAI o3) to automatically classify each ambiguous question in the dataset. The judge receives the question along with multiple evidence-supported execution trajectories leading to different valid answers and is asked to assign the primary ambiguity category (full prompt in Appendix G.8).
>
> The resulting distribution and illustrative examples are summarized below:
>
> | Ambiguity Type         | Definition                                                                                                                               | Frequency |
> | ---------------------- | ---------------------------------------------------------------------------------------------------------------------------------------- | --------: |
> | Under-Constrained      | The question lacks critical contextual constraints, allowing multiple distinct entities to satisfy the query                              | 52%       |
> | Granularity Ambiguity  | The required level of specificity (e.g., spatial, temporal, numeric) is not specified, yielding answers valid at different granularities   | 34%       |
> | Time Sensitivity       | The correct answer depends on an implicit temporal reference point                                                                      | 5%        |
> | Evidence Conflict      | Retrieved passages contain contradictory factual claims about the same entity                                                           | 4%        |
> | Multi-Item Response    | The question expects a single answer, but the true answer is a non-singleton set with no selection criteria provided                      | 3%        |
> | Open-Ended             | The question is inherently subjective or interpretive, permitting multiple qualitatively valid responses                               | 1%        |
> | Alias Variance         | Different surface forms refer to the same underlying real-world entity                                                                  | 1%        |

---

> ### Author Response · Authors · 2025-11-21
> **Response [2/3]**
>
> ## Response to Weakness #2 - continued
>
> | Type                   | Question Example                                                                                                                    | Alternative Answers                     | Explanation                                                                                            |
> |------------------------|-------------------------------------------------------------------------------------------------------------------------------------|-----------------------------------------|--------------------------------------------------------------------------------------------------------|
> | Under-Constrained      | What was the record label of the performer of *There Goes Rhymin' Simon*?                                                           | "Columbia Records", "Warner Bros."      | The artist was signed to different labels at different career stages         |
> | Granularity Ambiguity  | What is the place of birth of the director of the film *The Outsider* (2018)?                                                       | "Denmark", "Fredericia"                 | Country vs. city level; both factually correct but differ in specificity                               |
> | Time Sensitivity       | Where is the next FIFA World Cup going to take place?                                                                               | "Qatar", "United States, Canada, Mexico"| Depends on whether the question is asked before or after the 2022 event                                |
> | Evidence Conflict      | How many fish species live in the river system containing the Jari River?                                                           | "over 5,600", "2,200"                   | Different retrieved sources provide conflicting counts                                                |
> | Multi-Item Response    | Who is the child of the performer of "You Can Call Me Al"?                                                                          | "Lulu", "Harper Simon"                  | The performer has multiple children                          |
> | Open-Ended             | In what ways did Kanye draw inspiration from U2, Led Zeppelin, and the performer of "Mother's Little Helper"?                       | Multiple stylistic interpretations      | Inherently supports diverse valid qualitative answers                                                 |
> | Alias Variance         | Who gets Blair pregnant in season 5 of the series that had an episode titled "The Ex Files"?                                      | "Louis Grimaldi", "Prince of Monaco"    | Same entity referred to by personal name vs. title                                                    |
>
> These results demonstrate that superficial linguistic variations (e.g., Alias Variance) constitute only ~1% of the ambiguous cases. The majority instead arises from two sources:
>
> - **Under-Constrained questions** (52%): the question omits crucial disambiguating constraints, making multiple distinct entities or relations factually correct. Resolving these requires the model to actively explore **different search paths** to discover the alternative valid answers.
> - **Granularity Ambiguity** (34%): the question does not specify the desired level of specificity, so answers at different granularities (e.g., country vs. city, year vs. exact date) are all factually supported. Even when the model follows a **correct search path** and retrieves the right evidence, it can still be **unfairly penalized** simply because the reference answer arbitrarily prefers one granularity over another.
>
> We believe this new quantitative analysis significantly strengthens the paper's core claim about the prevalence and nature of genuine ambiguity in multi-hop QA, directly addressing the reviewer's concern.

---

> ### Author Response · Authors · 2025-11-21
> **Response [3/3]**
>
> ## Response to Question #1
> > [Q1] Why are K LLM verifiers necessary? Just for the ensemble?
>
> Thank you for the question. The use of multiple LLM verifiers is not merely for conventional ensembling, but is essential to **achieve high-confidence, low-noise judgments** when automatically determining whether a candidate answer is genuinely evidence-supported.
>
> In our data construction pipeline (detailed in Appendix B.2), we observe that **even state-of-the-art LLMs exhibit individual biases in verification strictness**. Some models (e.g., Claude 3.7 Sonnet) are consistently conservative, while others (e.g., OpenAI o4-mini) are markedly more lenient. Relying on any single verifier, therefore, introduces unacceptable noise, especially on the subtle and borderline cases.
>
> To directly measure the effect of these biases, we conducted a human evaluation (reported in Appendix B.2): for each possible voting threshold η, we randomly sampled 100 trajectories that received a "supported" vote under that threshold and had human annotators independently judge whether they indeed constitute valid, evidence-supported alternative answers. The results are as follows:
>
> | Voting threshold η                  | Human agreement |
> |-------------------------------------|----------------:|
> | η = 1 (any single verifier agrees)  | 64.0%           |
> | η = 2 (at least two verifiers agree) | 79.0%           |
> | η = 3 (our chosen setting)          | **96.0%**       |
> | η = 4 (all four verifiers agree)   | 99.0%           |
>
> As the numbers clearly illustrate, a single LLM verifier is too noisy (yielding only 64% human agreement). Requiring majority agreement from at least three out of four diverse verifiers (η = 3) raises agreement to 96%, which we consider the minimum acceptable quality, while still retaining enoughtrajectories for training.
>
> Therefore, the ensemble of verifiers combined with a strict majority-voting threshold is a deliberate and necessary design choice to suppress individual model biases and ensure the cleanliness of the resulting ambiguous QA dataset.
>
> ## Response to Question #2
> > [Q2] L174 - what is the meaning of a n_hat s?
>
> The symbol $\hat{ans}$ refers to the **final candidate answer extracted from the end of a generated search trajectory** $\tau$.
>
> Each trajectory $\tau$ produced by the search model consists of a sequence of iterative search actions, tool responses, intermediate reasoning steps, and an explicit answer that the model proposes for question $q$. We use $\hat{ans}$ specifically to denote this **model-produced answer** (as opposed to the original reference answer $ans^*$ in the dataset), and the hat distinguishes it as a sampled output. We will add clarifications in the revised manuscript for improved readability.
>
> ## Response to Question #3
> > [Q3] How does performance increase even on non-ambiguity datasets? Are ambiguous answers substantially included in datasets like HotpotQA?
>
> While performance gains on standard "single-answer" benchmarks may seem counterintuitive at first, they are in fact fully aligned with our core observation: **even datasets explicitly designed with a single annotated gold answer still contain a substantial amount of latent ambiguity that current training pipelines fail to leverage.**
>
> To quantify this phenomenon, we conducted a large-scale analysis in Appendix C.3. We applied the same *Filtering → Verification → Grouping* pipeline (originally introduced in Section 4 for constructing ambiguity-aware training data) to the search trajectories generated by five strong baseline models (ReSearch-7B/32B and Search-R1-7B/14B/32B) on seven QA benchmarks. This fully automatic process validates, for each question in the benchmarks, whether the models’ predicted answers are genuinely supported by the retrieved evidence. The results consistently confirm the prevalence of resolvable ambiguity across datasets and models.
>
> We observe a **non-trivial proportion of questions** containing validated alternative answers across all benchmarks, including HotpotQA:
>
> | Benchmark | % Questions with Valid Alternative Answers |
> | --------- | -----------------------------------------: |
> | MuSiQue   |                                      19.7% |
> | HotpotQA  |                                      11.1% |
> | 2Wiki     |                                       8.6% |
> | Bamboogle |                                       8.0% |
> | NQ        |                                      12.7% |
> | PopQA     |                                      14.7% |
> | TriviaQA  |                                       7.0% |
>
> This finding indicates that ambiguity is **not a rare annotation artifact**, but a **systematic and pervasive** property inherent to current open-domain QA datasets.

---

> ### Author Response · Authors · 2025-11-26
> **Gentle Reminder to Reviewer 6bRw**
>
> Dear Reviewer 6bRw,
>
> Thank you very much for your thoughtful and detailed review. We sincerely appreciate the time and care you devoted to evaluating our work. In our author response, we provided concise clarifications addressing your core concerns:
>
> 1. We highlighted that **genuine ambiguity in current QA benchmarks is far more common and essential than typically assumed**, making this a necessary and timely problem to address.
> 2. We clarified that the novelty of our work lies not in architectural changes or new losses, but in proposing the **first end-to-end ambiguity-aware data construction and training framework** that generalizes across diverse model backbones and benchmarks.
> 3. We added a more detailed quantitative breakdown of ambiguity sources, which we believe significantly reinforces our central findings and claims.
> 4. We explained that the use of multiple LLM verifiers is not simple ensembling but is crucial for obtaining high-confidence, low-noise evidence-based judgments in automatic ambiguity detection.
> 5. We further showed that even single-answer datasets contain substantial ambiguity, indicating that ambiguity is a broad property of open-domain QA data rather than a rare annotation artifact.
>
> As **the discussion period enters its final week**, we would be very grateful if you could revisit our paper in light of these clarifications and expanded analyses. We hope these updates help present the significance and originality of our contribution more clearly.
>
> Please feel free to let us know if any further questions arise. We would be very happy to discuss.
>
> Thank you again for your time and constructive feedback.

---

### Author Response · Authors · 2025-12-03
**Rebuttal Summary for Area Chairs**

Thank you to the Area Chair for your dedication during a challenging review cycle. We sincerely appreciate the time and effort you have devoted to evaluating our submission. Below is a concise summary of our rebuttal for your reference.

Our work identifies a fundamental challenge in modern open-domain QA: existing QA datasets contain substantial latent ambiguity, which hinders both model training and evaluation. We propose A$^2$Search, the first annotation-free, end-to-end RL framework that scales ambiguity-aware QA training to realistic, open-domain settings. The framework is cost-efficient, reproducible, and generalizes well across datasets and model families, providing a practical foundation for future research.

We received reviews from three reviewers. Their scores remain unchanged since the start of the rebuttal period, and in most cases we did not receive further engagement after our responses. We believe our detailed theoretical analysis, extensive experiments, and case studies thoroughly address all concerns raised.

| Reviewer | Score | Rebuttal Status                                                       |
| -------- | ----- | --------------------------------------------------------------------- |
| 6bRw     | 6     | non-responsive                                                        |
| wyg8     | 6     | non-responsive                                                        |
| FJjM     | 4     | requested additional ablations earlier; no further comments afterward |

We also appreciate the reviewers highlighting the key strengths of our work:

| Strength                                                                    | Mentioned by     |
| --------------------------------------------------------------------------- | ---------------- |
| Clear motivation; the work tackles an important challenge in open-domain QA | 6bRw, wyg8, FJjM |
| Clear writing and comprehensive empirical evidence supporting claims        | 6bRw, wyg8       |
| Proposed method is efficient, practical, and scalable                       | 6bRw, FJjM       |

We are grateful for the constructive questions raised by the reviewers. We carefully addressed each point and revised the manuscript where appropriate. Below is a brief summary of the main concerns and how they were resolved:

1. **Novelty concerns (6bRw).** We clarified that our main contribution is two-fold: identifying a significant overlooked challenge in existing QA pipelines, and proposing a practical, generalizable framework that scales ambiguity-aware training to open-domain settings. This provides a solid foundation for future research.

2. **Request for statistical analysis of ambiguity types (6bRw).** We conducted a large-scale systematic analysis of ambiguity categories, substantially strengthening our central claims (added as Section 5.3).

3. **Performance gains on single-answer datasets (6bRw).** We showed that all current QA benchmarks contain substantial latent ambiguity, supported by quantitative evidence (e.g., 11% ambiguity rate in HotpotQA).

4. **Feasibility concerns about sampling cost (wyg8).** We reported detailed runtime measurements demonstrating that sampling 3.99M trajectories only takes 74.9 hours on a single 4×H100 node, making the process scalable and far more efficient than manual annotation (Appendix B.1).

5. **Generalization beyond Qwen models (wyg8).** We trained Llama-based versions of A$^2$Search, showing that the approach generalizes well and provides clear performance gains on a different backbone family (Appendix E.5).

6. **Reliance on proprietary verifiers (wyg8).** We conducted additional experiments with four open-weight verifiers, demonstrating that an all-open pipeline reliably approximates proprietary judgments and supports reproducible data construction (Appendix B.4).

7. **Concern about over-generation to boost recall (FJjM).** Through theoretical analysis and empirical evaluation, we showed that A$^2$Search does not rely on over-generation, thanks to the structure of our data construction pipeline and reward design.

8. **Concern that improvements stem from paraphrasing or format gaming (FJjM).** We performed extensive qualitative analysis and trained three ablation models, showing that performance improvements arise from genuine ambiguity resolution rather than superficial effects (Appendix E.6).

9. **Requests for writing clarifications.** We revised the manuscript to address all such comments.

All relevant changes have been incorporated into the revised manuscript. We thank the AC and reviewers again for your time and effort. We hope this summary provides a clear overview of our rebuttal progress and helps facilitate your assessment.

---

### Meta-Review · Area_Chair_RpHb · 2026-01-21

**Summary:**

Novelty and Technical Contribution
- Reviewer 6bRw noted that the **core learning method and reward design have moderate novelty**, as the AnsF1 reward is derived from standard precision/recall metrics and the RL framework (GRPO) is existing work. The contribution lies primarily in **problem formulation and the automated data pipeline** rather than new architectures or objectives.

Source of Performance Gains
- Reviewer FJjM raised significant concerns about whether improvements stem from **genuine ambiguity resolution or superficial effects**: Could the model be over-generating answers to inflate recall? Are gains simply from outputting paraphrases of the same answer? Does the list output format itself explain performance differences?

Reproducibility and Scalability
- Reviewer wyg8 raised practical concerns:
  - No evidence for cross-family generality (only Qwen tested)
  - Computational cost of generating millions of trajectories
  - Reliance on proprietary verifiers (Claude, GPT) affecting reproducibility

Scope Limitations
- Unanswerable queries are not addressed (wyg8)
- **Evaluation fairness**: comparing A²Search @1 with baselines @3 was questioned (FJjM)

**Reviewer Concerns:**

Novelty and Technical Contribution (Partially Addressed)
- Authors clarified that contribution is in "reformulating how multi-hop QA models learn from inherently ambiguous data" and "providing a practical foundation that scales ambiguity-aware QA training" rather than the reward design itself.

Source of Performance Gains (Fully Addressed)
- Authors added comprehensive 7-category taxonomy with frequency analysis: Under-Constrained (52%), Granularity Ambiguity (34%), Alias Variance (1%). This directly shows "superficial linguistic variations constitute only ~1% of the ambiguous cases. They also demonstrated that "even datasets explicitly designed with a single annotated gold answer still contain substantial latent ambiguity" with quantitative evidence: MuSiQue 19.7%, HotpotQA 11.1%, 2Wiki 8.6%.

Reproducibility and Scalability (Fully Addressed)
- Authors trained A²Search-LMA-3B on Llama 3.2, showing "Llama-based models achieve performance comparable to Qwen models of the same size" (42.3% vs 43.1% avg AnsF1), confirming "framework transfers well across model families."
- Authors tested four open-weight verifiers achieving "80% precision and 85% recall with respect to the proprietary ensemble" and "96% human agreement" at ≥3/4 voting threshold, demonstrating "a fully open verifier stack can reliably approximate the judgments of proprietary models."

Scope Limitations (Partially Addressed)

**Reviewer Scores:**

Reviewer 6bRw: 6 → Likely 6 (possibly 7)
- The reviewer's specific requests (ambiguity analysis, explanation for gains on single-answer datasets) were fully satisfied. The novelty concern is philosophical rather than a flaw. As the reviewer was already "marginally above acceptance" and stated "would not mind if paper is rejected," the score would likely remain at 6 or possibly increase to 7.

Reviewer wyg8: 6 → Likely 6-7
- Three of four concerns were comprehensively addressed with new experiments. The scope limitation was reasonably justified. Given the thorough responses, the score would likely remain at 6 or increase to 7.

Reviewer FJjM: 4 → Likely 5-6
- This reviewer engaged most actively and received the requested ablation experiments. Given all requested experiments confirmed the authors' claims, a score increase is likely.

  -

---

### Decision · Program_Chairs · 2026-01-26

Accept (Poster)